# Experimental evidence of seismic ruptures initiated by aseismic slip

Yohann Faure ⓘ & Elsa Bayart ⓘ ✉

Seismic faults release the stress accumulated during tectonic movement through rapid ruptures or slow-slip events. The role of slow-slip events is crucial as they impact earthquakes occurrence. However, the mechanisms by which slow-slip affects the failure of frictionally locked regions remain elusive. Here, building on laboratory experiments, we establish that a slow-slip region acts as a nucleation center for seismic rupture, enhancing earthquakes' frequency. We emulate slow-slip regions by introducing a granular material along part of a laboratory fault. Measuring the fault's response to shear reveals that the heterogeneity serves as an initial rupture, reducing the fault shear resistance. Additionally, the slow-slip region extends beyond the heterogeneity with increasing normal load, demonstrating that fault composition is not the only requirement for slow-slip. Our results show that slow-slip modifies rupture nucleation dynamics, highlighting the importance of accounting for the evolution of the slow-slip region under varying conditions for seismic hazard mitigation.

Seismic faults release accumulated stresses via both rapid ruptures, called seismic events, and slow slip events (SSEs)[1]. SSEs were first discovered in subduction zones, but improved measurement techniques have shown that they also occur within subsurface strike-slip faults[2]. Slow earthquakes can affect a large portion of a fault, resulting in the release of a significant amount of energy. However, they can also be localized to very small portions of faults; these small events are detected indirectly by the seismic events they generate, also known as tremors[3–5]. While their existence is well-established, the role of SSEs in the seismic cycle is still unclear. In some cases, they can be responsible for triggering large earthquakes[6], while in other cases they occur periodically without being earthquake precursors[7]. To describe the slip behavior of a fault, the slip velocity of the fault and the loading velocity are compared through the coupling term. The fault is uncoupled when it slides at the loading velocity and coupled when it is locked during the inter-seismic period. Understanding the interaction mechanisms between uncoupled or partially coupled slipping zones within a fault or fault system, and coupled frictionally locked zones is essential, especially for the design of fault monitoring strategies[8]. A critical question is what fault properties are required for SSEs to occur. Slow slip areas under steady sliding are commonly modeled as velocity strengthening zones[9] that cannot

rupture seismically. However, simple geometric complexity is sufficient to induce slow slip, without introducing any variation of the fault frictional properties over time or space[10]. In addition, observations have shown that seismically slipping zones can exhibit slow slip[11–13]. On the experimental side, slow ruptures have been shown to propagate under quasi-static conditions for low shear loading[14] or low normal loading with high pore fluid pressure[15]. Understanding what sets the boundary between slow slip zones and seismic zones is a key point in improving seismic risk monitoring and prevention[16].

In this study, we consider the case of a laboratory frictional interface along which slow slip is induced by the presence of a heterogeneity in the interface composition. Laboratory experiments provide a unique opportunity to control the local fault composition and loading, and thus to understand the local mechanisms responsible for large-scale phenomena. Interfacial heterogeneities affect the macroscopic dynamics of a frictional system[17,18], for example by promoting confined ruptures and thus increasing the frequency of occurrence of rapid slip events[19–21]. Fault inhomogeneities have, moreover, been shown to be prone to exhibit slow slip[22,23]. Our experiment demonstrates a case where slow slip is responsible for the initiation of rapid slip events by acting as an initial rupture, which

Laboratoire de Physique, Université de Lyon, Ecole Normale Supérieure de Lyon, CNRS, 46 allée d'Italie, Lyon 69007, France.
✉e-mail: elsa.bayart@ens-lyon.fr

eventually propagates and destabilizes the interface. We highlight the interaction between a decoupled, slowly-slipping zone and coupled, seismically ruptured zones, and the role of this interaction in modifying the stick-slip cycle.

## Results

### Laboratory-fault experiment

A quasi-1D frictional interface is formed by two macroscopically flat solid surfaces in contact, with a granular material embedded over 20%

of the total length (Fig. 1a). The system consists of two poly(methylmethacrylate) (PMMA) blocks machined as follows: the contacting surfaces have a 1-μm r.m.s. roughness, while a semi-elliptical shape is hollowed out in the center of each surface, forming an eye-shaped hole when the blocks are in contact, into which grains are inserted (Methods). The embedded granular material is made 2D by inserting nylon cylinders of length $w$, with $w$ the blocks' thickness, parallel to the $z$-axis. Different cylinder diameters, 0.4, 0.7, 0.9, and 1.3 mm in approximate volume ratios of 5, 10, 35, and 50%, are used to

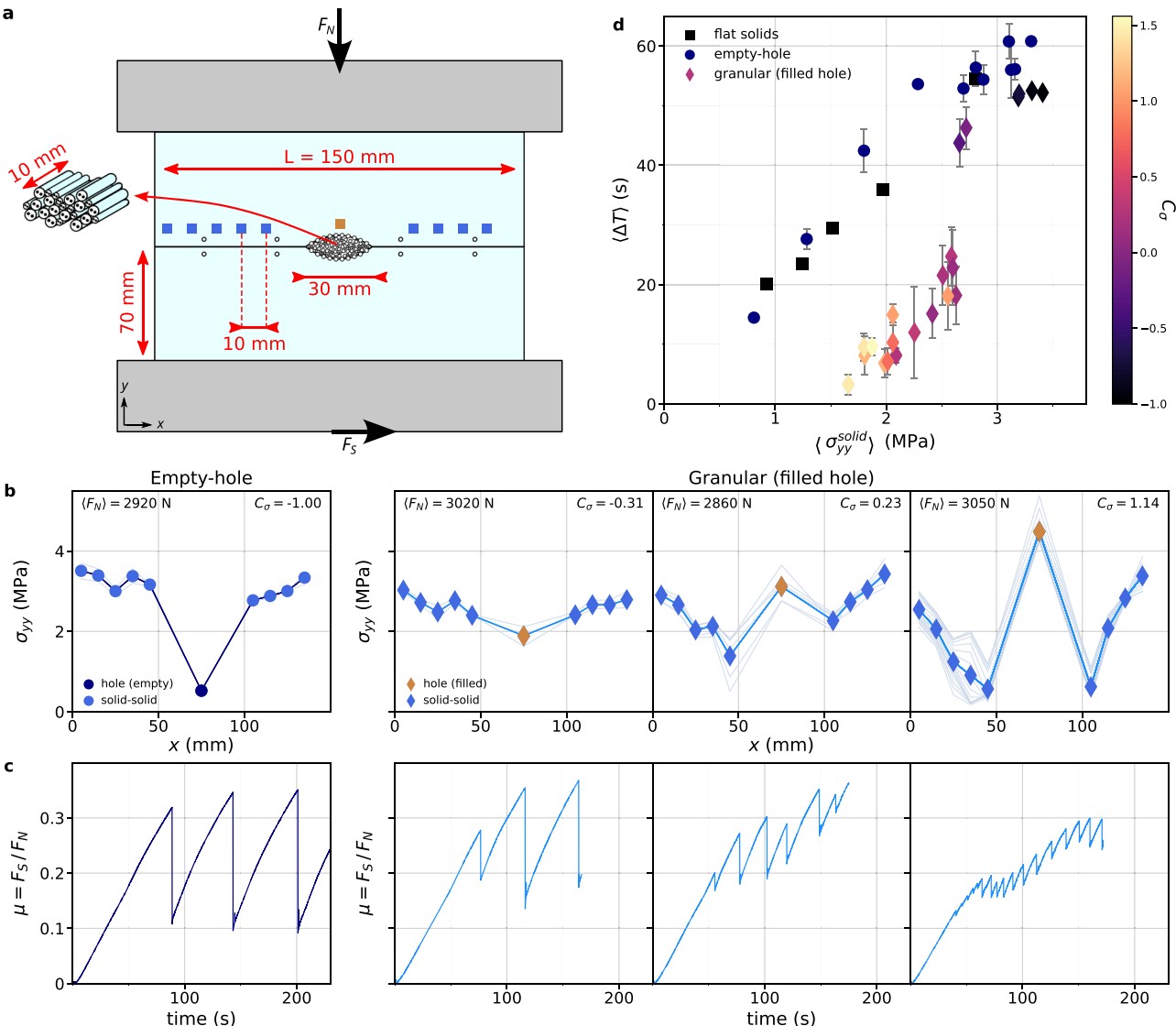

**Fig. 1 | Experimental setup, frictional behavior and stick-slip frequency. a** Two blocks of poly(methylmethacrylate) (PMMA) form a frictional interface with a central eye-shaped hole. The blocks and the hole are (L × H × e) 150 × 90 × 10 mm and 30 × 6 × 10 mm respectively. Shear force $F_S$ is applied by translating the bottom block. Nylon cylinders of diameters 0.4, 0.7, 0.9, and 1.3 mm (volume ratio: 5, 10, 35, and 50%) fill the hole and 1.3 mm diameter cylinders are glued into the hole surfaces for roughening. An array of 10 strain gauges (blue squares) measures the 2D strain tensor components from 2 to 6 mm above the interface. Four pairs of cylinders' patterns are drawn along the solid-solid interface (white circles). Axis $x,y,z$ are respectively the shear loading, normal loading and sample thickness directions. **b** Normal stress distributions $\sigma_{yy}(x)$ along the interface for (left) a reference empty-hole experiment and (right) three granular experiments (hole filled with cylinders), conducted at $F_N \sim 3000$N. For the granular experiments, from left to right, the density of the granular medium is increased. The loading contrast, $C_\sigma = (\sigma_{yy}^{gran} - \langle\sigma_{yy}^{solid}\rangle)/\langle\sigma_{yy}\rangle$, is indicated in each panel. Light blue symbols (circle

for empty-hole experiments, diamond for granular experiments) indicate measurements above the solid-solid sections, dark blue circle above the empty hole and brown diamond above the granular section. Gray lines correspond to measurements before each rapid event, colored lines are averaged over all events of an experiment. **c** Temporal evolution of the friction coefficient $\mu = F_S/F_N$ for the same experiments as in (**b**), demonstrating a stick-slip dynamics. Dark and light blue lines are respectively for empty-hole and granular experiments. **d** Mean stick-slip period as a function of the average normal stress carried by the solid-solid sections $\langle\sigma_{yy}^{solid}\rangle$ for experiments with flat solids (black squares) and empty-hole experiments (circles) performed under varying normal force, 750 N < $F_N$ < 3000 N, and granular experiments (diamonds) performed at $F_N \sim 3000$ N with a varying loading contrast $C_\sigma$. The colorbar codes for $C_\sigma$ for granular experiments. Error bars correspond to the standard deviations of the inter-event times over all the events of each experiment. Source data are provided as a Source Data file.

prevent crystallization. The cylinders are held in place in the eye-shaped hole by friction and jamming. To force interfacial slip within the granular pile, additional cylinders are bonded into grooves machined into the solid surfaces forming the hole. The granular packing fraction is controlled by the number of cylinders inserted into the hole. PMMA has a strain rate dependent Young's modulus $E$ equal to 3 GPa for low strain rate and 5.6 GPa for high strain rate[24], Poisson's ration $v = 0.3$, and a Rayleigh wave speed $c_R = 1237 \pm 10$ m/s (plane stress). For nylon, $E = 1.4$ GPa and $v = 0.4$.

The blocks are pressed together with a normal displacement resulting in a normal force $F_N \sim 3000$ N (2800 N $< F_N < 3200$ N), unless otherwise specified. The shear force $F_S$ is uniformly applied by translating the bottom block at 20 μm/s. $F_N$ and $F_S$ are measured at 315 Hz via load cells. The 2D strain tensor $\varepsilon_{ij}(t)$ is measured at 10 locations, a few millimeters above the interface, and both continuously recorded at 315 Hz and at 4 MHz, when triggered by a rapid event ("Methods" section).

Interfacial slip measurements are performed using cylinder position data obtained with a sub-pixel resolution particle tracking algorithm. Particle tracking involves the correlation of images taken at 100 frames per second during an experiment with a template image. Sub-pixel resolution is achieved using three-point estimator with a Gaussian peak fit[25], resulting in an 8 μm resolution in displacement ("Methods" section). Tracking is performed on the cylinders embedded in the eye-shaped hole boundaries and on similar patterns drawn on the block faces above the interface solid-solid sections, allowing measurements under the same conditions as for the hole. The interfacial slip is measured at 4 locations along the solid-solid sections and 3 locations above the hole, by subtracting the measured position of cylinders or patterns located on the top and bottom blocks, at the same location $x$. Corrections for the rotation of the system during the shear loading are made ("Methods" section).

## Slow slip and stick-slip cycle

We perform reference experiments with an empty hole and granular experiments with cylinders filling the hole. In both cases, the frictional system experiences stick-slip motion (Fig. 1c). The stick-slip frequency is, however, increased in presence of the granular material compared to the reference experiment. The increase is especially noticeable when the density of the granular medium is increased, i.e. an increased number of inserted cylinders (Fig. 1b). We compute the normal stress $\sigma_{yy}$ using the strain gauges measurements, at 9 locations above the solid-solid sections of the interface and one location above the hole, when empty or filled with grains. We define $\sigma_{yy}^{patch}$ as the normal stress measured above the hole, empty or filled, and $\sigma_{yy}^{solid}$ as the normal stress measured above the solid-solid sections of the interface. As $F_N$ is the same for all experiments, $\sigma_{yy}^{patch}$ (resp. $\sigma_{yy}^{solid}$) carried by the granular (resp. solid-solid) section increases (resp. decreases) with density (Fig. 1b). We define the loading contrast $C_\sigma$ as:

$$C_\sigma = \frac{\sigma_{yy}^{patch} - <\sigma_{yy}^{solid}>}{<\sigma_{yy}>} \tag{1}$$

where $<\sigma_{yy}^{solid}>$ and $<\sigma_{yy}>$ are respectively the averaged normal stresses carried by the solid-solid sections and by the total interface. For empty-hole experiments, $C_\sigma \approx -1$, while for granular experiments, $-1 < C_\sigma < 2.5$.

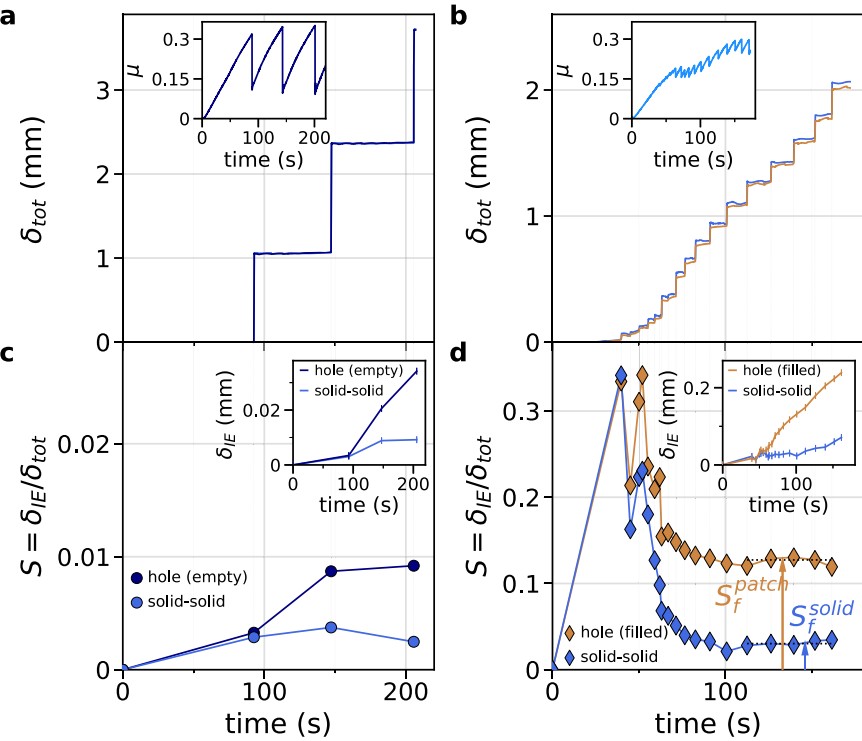

**Fig. 2 | Measurements of the interfacial slip. a, b** Temporal evolution of the total interfacial slip of the solid-solid sections $\delta_{tot}^{solid}(t)$ (light blue line) and of the patch $\delta_{tot}^{patch}(t)$, when empty (dark blue line in (**a**)) or filled with cylinders (brown line in (**b**)), obtained from the particle tracking method for (**a**) an empty-hole experiment and (**b**) a granular experiment ($C_\sigma = 1.2$). Measurements are averaged over 4 locations for the solid-solid sections and 3 locations for the hole, empty or filled. Inset: Temporal evolution of the friction coefficient $\mu$ for the considered experiments. **c, d** Temporal evolution of the normalized cumulated inter-event slip $S^{solid}(t)$ and $S^{patch}(t)$, i.e. the interfacial slip excluding the slip occurring during rapid slip events (**c**) for the empty-hole experiment and (**d**) for the granular experiment. Each dot corresponds to one measurement performed at the end of an inter-event period. Color code and symbols are the same as in (**a, b**). The black dotted lines in (**d**) show the averaged values defined as $S_f^{solid}$ and $S_f^{patch}$, which are defined as the converged value of the normalized cumulated inter-event slip $S^{solid}(t)$ and $S^{patch}(t)$. Inset: the cumulated inter-event slip $\delta_{IE}^{solid}(t)$ (in **c**) and $\delta_{IE}^{patch}(t)$ (in **d**) without normalization (same colors as in main plots). Source data are provided as a Source Data file.

In a classical friction experiment where two flat solid blocks are in contact along the entire interface, i.e. without a hole or a granular patch, a reduced normal load leads to smaller shear force drops and therefore to an increased stick-slip frequency. To verify that the observed frequency increase for an increasing loading contrast $C_\sigma$ is not due to a decrease in the normal loading of the solid-solid sections of the interface, $<\sigma_{yy}^{solid}>$, we compare the averaged stick-slip period $<\Delta T>$ of no-hole experiments, i.e. classical solid-solid friction experiments with flat homogeneous solids in contact, under varying normal force, $750N < F_N < 3000N$, with our reference empty-hole experiments under the same range of normal force, and with granular experiments under $F_N = 3000N$ with a varying density, hence varying $<\sigma_{yy}^{solid}>$ and $C_\sigma$ (Fig. 1d). First, the stick-slip periods of no-hole and empty-hole experiments are comparable for the same $<\sigma_{yy}^{solid}>$, demonstrating that the presence of the hole has no effect on the stick-slip cycle under given loading conditions. Slight differences between the two datasets are due to expected variations in static friction between systems[26]. Second, the period shortening in the presence of the granular section is greater than that observed for a corresponding normal load reduction in no-hole and empty-hole experiments, demonstrating an effect of the compositional heterogeneity on the stick-slip cycle beyond unloading. The high-frequency strain measurements allow us to verify that, in our experiments, each force drop corresponds to a rupture that spans the entire interface, and not to arrested ruptures that may act as precursors to the system-spanning ruptures[27]. In the following, we analyze the underlying mechanisms for this intriguing phenomenon, where the presence of a granular material induces an increase in the stick-slip frequency exhibited by the frictional system.

Interfacial slip measurements are used to determine the total slip of the solid-solid section of the interface, $\delta_{tot}^{solid}(t)$, and of the empty-hole or granular sections, $\delta_{tot}^{patch}(t)$, and the cumulated inter-event slip $\delta_{IE}^{solid}(t)$ and $\delta_{IE}^{patch}(t)$, defined as the slip experienced by the interface *excluding* slip occurring during rapid events (Fig. 2a, b). These quantities are averaged over the measurement locations for solid-solid sections and empty-hole or granular sections. We define the normalized cumulated inter-event slip $S^{\{l\}}(t)$ as:

$$S^{\{l\}}(t) = \frac{\delta_{IE}^{\{l\}}(t)}{\delta_{tot}^{\{l\}}(t)} \qquad (2)$$

where {l} refers to solid-solid (for an empty hole) or patch (for a hole filled with cylinders). $S^{\{l\}}(t)$ quantifies the coupling of the interface as it corresponds to the fraction of slip occurring during the inter-event periods up to a time t. For reference empty-hole experiments, measurements show that the entire interface is locked during the inter-event time, with $S^{\{l\}}(t) < 0.02$ for both sections, corresponding to the ideal stick-slip motion where slip occurs only during rapid events (Fig. 2c). On the contrary, for a larger loading contrast, corresponding to granular experiments, the granular patch experiences inter-event slip while the solid-solid sections remain almost locked between rapid slip events ($S^{solid} \ll S^{patch}$ in Fig. 2d). The measurements of $S^{solid}(t)$ and $S^{patch}(t)$ show large values at early time. For short inter-event periods, i.e. high stick-slip frequency, the quantity $S^{\{l\}}$ consists of the division of two comparable quantities, i.e. the slip over a few inter-event periods and the total slip, both of which are limited at early times. In this case, the uncertainty of the slip measurements becomes of considerable importance. Therefore, we consider the converged value of the cumulated inter-event slip, $S_f^{\{l\}}$, as the average over the last few events of an experiment, as a function of the loading contrast $C_\sigma$ (Fig. 3a). $S_f^{patch}$ and $S_f^{solid}$ are found to increase with $C_\sigma$, but $S_f^{solid}$ always remains below $S_f^{patch}$ (Fig. 3b). This results indicates that differential interfacial

slip, increasing with $C_\sigma$, is induced along the interface between the granular and solid-solid sections. We note that the mean stick-slip frequency of the experiments, $<1/(\Delta T)>$, increases with $S_f^{patch} - S_f^{solid}$, suggesting that this latter quantity, the differential cumulated inter-event slip, is a relevant control parameter for the rapid slip events occurrence. Based on quasi-static and dynamic strain measurements, we will show that slow slip is localized in the granular patch but also in surrounding solid-solid zones.

## Expansion of the slipping zone with the normal load

Rapid sliding events experienced by a frictional interface are mediated by the propagation of a rupture that weakens the microcontacts resisting shear. These ruptures have been observed in many experimental systems with analog materials[28–31], rocks[32–34], or with a gouge layer[19,21]. They are classical shear cracks for a homogeneous system of two flat solids in contact[31,35]. Sometimes linear fracture mechanics must be modified to account for the inhomogeneous residual stress[36–38]. Hence, an increased occurrence of slip events, as observed in our experiment, indicates an increased frequency of interfacial rupture nucleation and propagation. We will show that the slowly slipping area expands when loading contrast is increased, leading to an early destabilization of the frictional system via rapid rupture propagation. The granular patch serves as a trigger for rupture nucleation, possibly due to shear stress buildup at the patch boundary caused by differential slip among interface sections.

In Fig. 4a, b, we present the dynamic measurements of the shear strain drop $\Delta\varepsilon_{xy}(x,t) = \varepsilon_{xy}(x,t) - \varepsilon_{xy}^0(x)$, where $\varepsilon_{xy}^0(x)$ is the shear strain before a rupture event, i.e. the average value of the first 1000 data points of the high-frequency strain record, for two rapid sliding events at a low (Fig. 4a) and high (Fig. 4b) loading contrast. The passage of the frictional rupture is revealed by a sudden drop in $\Delta\varepsilon_{xy}(x,t)$. Picking up the first variation time for each strain gauge provides a measurement of the rupture speed, shown to range from 500 m/s to 2500 m/s for all the observed events, which indicates that both sub-Rayleigh and supershear ruptures occur. In our experiment, each rapid rupture crosses the entire interface. We define the nucleation location of a rupture as the first location where $\Delta\varepsilon_{xy}(x,t)$ starts to depart from 0. Determining this point for all events, we establish histograms of nucleation locations for different ranges of $C_\sigma$ (Fig. 4c). For the reference empty-hole experiments, most of the rupture nucleations occur near one of the patch corners. The rupture nucleation location could be caused by a structural defect along the interface, such as a dip or bump. It is not the case as the nucleation occurs at the same locations, i.e. near the same patch corner, when one of the blocks is flipped, thus inverting the x-positions. Figure 4c clearly shows that the presence of a granular material, even at low density (low $C_\sigma$), affects the nucleation location, which then occurs on both sides of the patch. Moreover, as $C_\sigma$ increases, the nucleation point is shifted away from the granular patch towards the outer corners of the blocks. To demonstrate this effect, we calculate the mean values of the location of rupture nucleation over the different ranges of $C_\sigma$ and then define the mean nucleation distance $<d_{nuc}>$ as the distance from the interface center to the mean nucleation location (taking the absolute value of the nucleation x-coordinate) (Fig. 4d). Indeed, we find that the nucleation distance increases with $C_\sigma$.

Interfacial areas can slip when the frictional microcontacts are already weakened and thus no dynamic rupture can initiate within these areas. Therefore, the fact that rupture nucleation locations move away from the center with the loading contrast indicates that the length of the central slipping zone increases with $C_\sigma$. We interpret this result as the fact that a slowly slipping granular section induces the formation of an initial rupture – a line along which stresses have been released – which destabilizes into a rapid rupture once a critical shear stress is reached[39–41]. According to this scenario, the central zone of the

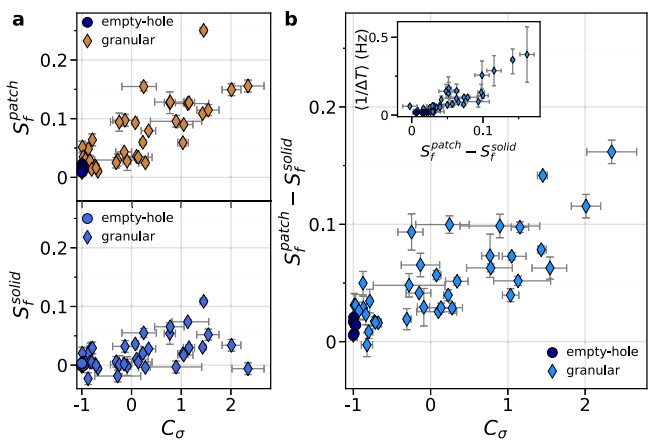

**Fig. 3 | Slow slip measurements. a** Converged values of the normalized cumulated inter-event slip (top) $S_f^{patch}$ for the empty-hole (blue circles) and the granular patch (brown diamonds) and (bottom) $S_f^{solid}$ for the solid-solid sections of the interface, as a function of the loading contrast $C_\sigma$. Horizontal error bars correspond to the standard deviation of the $C_\sigma$ values of each event for an experiment and vertical error bars correspond to the standard deviation of the values of $S^{solid}$ and $S^{patch}$ used to determine $S_f^{solid}$ and $S_f^{patch}$. **b** Differential normalized cumulated inter-event slip $S_f^{patch} - S_f^{solid}$ for empty-hole (dark blue circles) and granular (blue diamonds) experiments as a function of the loading contrast $C_\sigma$. Inset: mean frequency <$1/\Delta T$> of the stick-slip cycle as a function of $S_f^{patch} - S_f^{solid}$. Error bars for the $y$-axis corresponds to the standard deviation of the inter-event times over all the events of each experiments. Source data are provided as a Source Data file.

interface, including the eye-shaped hole filled with cylinders and adjacent solid-solid zones that do not break via dynamic ruptures, must exhibit slow slip during the inter-event periods. This is indeed what our tracking methods revealed (Fig. 3a), but the low spatial resolution of our slip measurements prevents a direct measurement of the slipping zone spatial extent. However, low-frequency strain measurements allow us to overcome this lack of resolution, as we will show below.

For an ideal stick-slip motion, the stick phases correspond to an elastic shear loading, where the shear strain (stress) increases linearly with tangential displacement of the block. In our experiments, this linear loading is indeed observed for some strain gauges, while a sublinear evolution of the shear strain is observed for others (Fig. 5a, b). Sublinear evolution refers to the shear strain signal, $\varepsilon_{xy}(t)$, saturating rather than increasing linearly over an inter-event period (except at very short times after a rapid event, when a rapid increase in shear strain corresponds to interfacial logarithmic aging). Since PMMA has a constant Young's modulus at low frequency, the sublinear shear strain evolution indicates that interfacial slip takes place. Hence, for each inter-event period, the strain gauges exhibiting this sublinear evolution of $\varepsilon_{xy}(x_{SG}, t)$, where $x_{SG}$ are the strain gauges locations along the interface, are detected as a marker of local interfacial slow slip. For all experiments, the fraction of sublinear loading sequences, $N_{SL}/N_{event}$, is counted per strain gauges on the low-frequency measurements of the temporal evolution of shear strain. Measurements are represented for different ranges of $C_\sigma$ (Fig. 5c). Locations with a large number of sublinear evolution of $\varepsilon_{xy}(x_{SG}, t)$ means that slow slip often occurs during the inter-event periods. The spatial extent of the distribution informs us about the most probable length of the slipping patch for a given range of loading contrast. The larger the central width of the distribution, the longer the slipping zone. The slipping patch length, <$l_{slip}$>, is defined as the averaged spatial extent of the region where the sublinear shear strain evolution is detected for each loading phase, i.e., each inter-event period, per range of $C_\sigma$. We first check the consistency of the two previous measurements: rupture nucleation location from high-frequency strain measurements and slipping patch

extension from quasi-static loading curves (Fig. 5d). The two lengths <$d_{nuc}$> and <$l_{slip}$> are indeed comparable, especially when the slipping patch becomes longer than the initial compositional heterogeneity, i.e., the hole length, $l_{hole}$ = 30 mm. Furthermore, as expected from the correlation between <$d_{nuc}$> and <$l_{slip}$>, Fig. 6a shows that the slipping patch length increases with $C_\sigma$. This plot indicates that the larger the normal load carried by the granular patch, the longer the slipping patch, that extends beyond the initial compositional heterogeneity and thus into the solid-solid sections of the interface, which otherwise may rupture seismically under different loading conditions.

### Slowly slipping area acting as a nucleation seed

Our combined measurements of the rupture nucleation locations and the evolution of the shear strain during the inter-event periods show that the introduction of a heterogeneity consisting of a granular material induces localized slow slip during the inter-event periods of the stick-slip cycle. As the normal load applied on the granular section of the interface increases, the slipping zone extends from nearly the size of the patch to almost the entire interface (Fig. 6a). In parallel, the stick-slip frequency increases with the loading contrast, revealing more frequent rupture initiations for a higher loading of the granular section (Figs. 1 and 3b).

Our interpretation is that the central slipping zone acts as a nucleation center, i.e., an initial rupture destabilized into rapid ruptures along the interface. An initiation criterion consists in relating the length of the initial rupture to the critical stress required for rupture destabilization. Many criteria exist, based on different physical quantities, and they generally demonstrate that the critical stress decreases with an increasing length of the initial rupture[39–43]. Linear fracture mechanics is a model, among others, that can be used to introduce an initiation criterion for a rupture. One advantage of this theoretical framework for frictional interfaces is that it relates the applied load to the fracture energy of the interface, a characteristic quantity that is accessible using off-fault strain measurements[31]. In this framework, an initial rupture destabilizes according to the Griffith criterion, $G_S = G_c$, where $G_S$ is the static energy release rate and $G_c$ is the fracture energy of the material. This criterion leads to an expression of the critical stress $\tau_c$ for which an initial rupture of length $l$ is destabilized, $\tau_c \propto \sqrt{G_c E/l}$, which in fact follows the expected behavior of the critical stress with the rupture length. Because we are considering a frictional rupture, where stresses are released to the residual stress value rather than zero[31], with spatially dependent shear loading due to compositional heterogeneity and a normal stress gradient induced by the loading contrast (Fig. 1b), a quantitative estimation of $G_S$ is challenging and cannot be performed in our experiment. In particular, knowledge of the residual stress spatial distribution within the slowly slipping patch is required to evaluate the stress intensity factor at the tip of the initial rupture[44], while in our experiments, only one strain gauge is located above this region. Although our experimental system is not designed to validate a criterion for rupture initiation over another, we can verify that the response of the system to the presence of a slipping patch follows the same trend as in the presence of an initial rupture.

In our experiment, the initial rupture length corresponds to the size of the central slipping zone. The higher the loading contrast, the longer the central slipping zone (Fig. 6a) and the higher the stick-slip frequency (Fig. 1d). This is qualitatively consistent with the Griffith criterion, and more generally with the trend expressed by initiation criteria for frictional ruptures: the longer the initial rupture ($l_{slip}$), the lower the critical stress at which ruptures initiate compared to the minimum stress reached after the interface weakening – the residual stress – resulting in the initiation of rapid slip events after a shorter loading time and thus to an increased stick-slip frequency.

In the fracture mechanics framework, the local value of the fracture energy $G_c$ also affects the stress $\tau_c$ at which a rupture initiates. The fracture energy $G_c$ strongly depends on the local normal load[27] and,

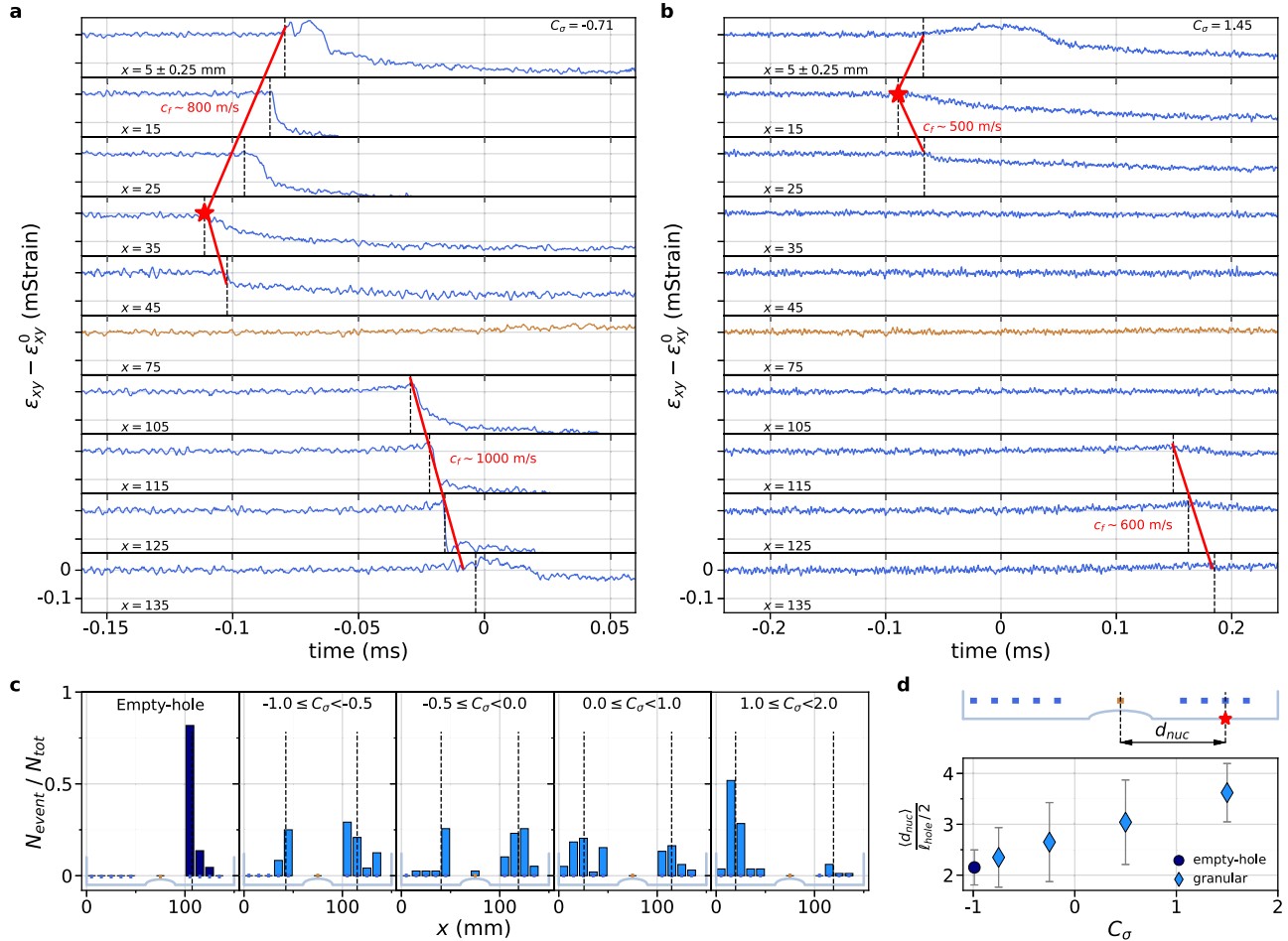

**Fig. 4 | Modification of the rupture nucleation location with an increasing loading contrast. a**, **b** Shear strain variation $\varepsilon_{xy}(x,t) - \varepsilon_{xy}^0(x)$ due to the propagation of a rapid rupture, where $\varepsilon_{xy}^0(x)$ is the initial value of the strain averaged over 1000 points, measured at 4 MHz for granular experiments at (**a**) $C_\sigma = -0.71$ and (**b**) $C_\sigma = 1.45$, at the 10 measurement locations (blue lines for strain gauges above the solid-solid sections, brown line above the granular patch). Red stars indicate the point at which the rupture initiates. The location of the corresponding strain gauge is determined as the nucleation location. Black dotted lines indicate the detection of the initiation of the strain variation for each strain gauge signal and red lines are a guide for the eye to indicate the approximate rupture passage locations. **c** Histograms of the rupture nucleation locations for the empty-hole experiments (left, dark blue) and the granular experiments (right, light blue) for different ranges of $C_\sigma$ mentioned on the plots. Each bin corresponds to the number of ruptures that have been initiated below a given strain gauge normalized by the total number of ruptures for the corresponding range of $C_\sigma$. The black dotted lines show the average value of the distributions used to define the nucleation distance $\langle d_{nuc} \rangle$. **d** Average values of the nucleation distance $\langle d_{nuc} \rangle$ normalized by the central hole half-length, $l_{hole}/2$ where $l_{hole} = 30$ mm, for each range of $C_\sigma$ used for the histograms in (**c**). The nucleation distance $\langle d_{nuc} \rangle$ is the mean value of the distance of the averaged nucleation location from the center of the interface, measured from the data shown in the histograms of (**c**). The circular markers correspond to empty-hole experiments and the diamond markers to granular experiments. Error bars corresponds to the standard deviations of the distributions in (**c**). Source data are provided as a Source Data file.

because of the heterogeneous normal stress distribution along the interface in our experiments, we cannot confidently extract the $G_c$ value from our strain measurements. However, the Griffith criterion must be fulfilled locally at the rupture tip for a rupture to be destabilized. Although $G_c$ cannot be measured, we do measure the normal stress at the nucleation point prior to each event. Since $G_c$ is linearly dependent on normal stress[27,45], this measurement gives an indication of the variation trend of the fracture energy at the rupture tip with the loading contrast. Figure 6b shows the value of $\sigma_{yy}$ at the nucleation point before the rapid events, $\langle \sigma_{yy}^0(x_{nuc}) \rangle$, where $x_{nuc}$ is the nucleation location, averaged over all the events of the empty-hole experiments and granular experiments for the four ranges of loading contrast, $C_\sigma$, shown in Figs. 4c and 5c. This plot demonstrates a decreasing trend of $\langle \sigma_{yy}^0(x_{nuc}) \rangle$ with the loading contrast, $C_\sigma$. Therefore, the variation of the fracture energy $G_c$ at the rupture nucleation location may play a role in reducing the critical stress for the slipping patch destabilization, an effect that is additive to that of the elongation of the slipping patch. The two parameters $l_{slip}$ and $G_c$ cannot be controlled

independently of each other in our experiment, due to the presence of the compositional heterogeneity, i.e., the granular patch.

This mechanism of an extending initial rupture leading to an increased stick-slip frequency, together with a local decrease of the interfacial fracture energy, is not trivial. An obvious scenario would be that the slow slip of the granular patch induces a stress concentration at the junction between the uncoupled zone – the patch – and the coupled zone – the solid-solid sections. Therefore, overloading at the patch corners would be responsible for dynamic rupture propagation by reaching the contacts' shear resistance, as observed in different systems[43,46]. However, this is not what we observe. The slow slip within the granular patch induces creep of the neighboring contacts rather than dynamic rupture. Thus, the slowly slipping area affects the rupture dynamics of the entire interface by modifying the rupture nucleation phase, rather than acting locally on the stress distribution. This result is in contrast with other studies where the frictional properties of interfacial heterogeneities are shown to evolve with the sliding history[22,46,47], or where an increased stick-slip frequency is

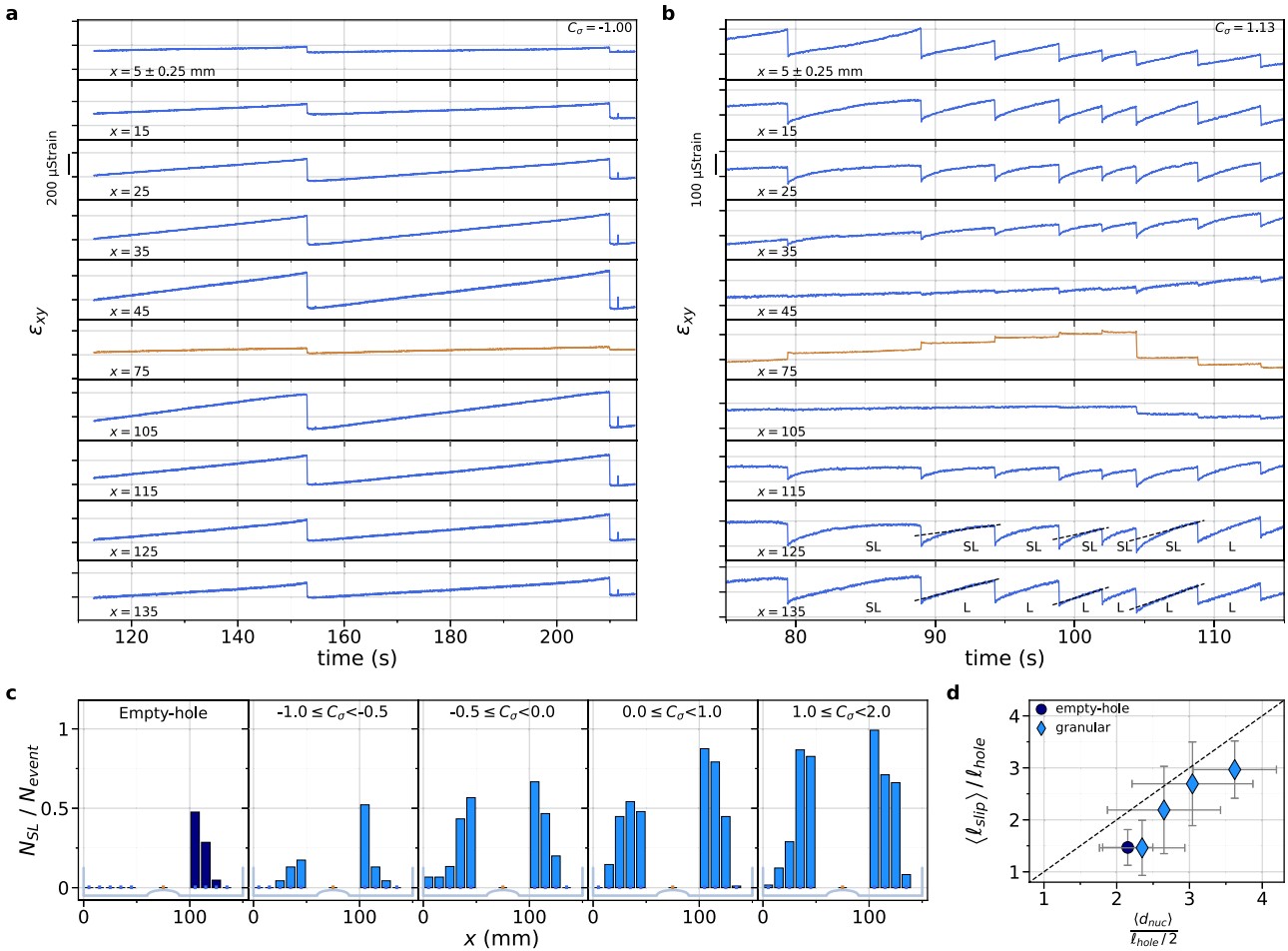

**Fig. 5 | Evolution of the slipping patch length with the loading contrast.**
**a**, **b** Temporal evolution of the shear strain $\varepsilon_{xy}(x,t)$ measured at 315 Hz for the 10 strain gauges locations $x_{SG}$, over (**a**) an empty-hole experiment at $C_\sigma = -1$, and (**b**) a granular experiment at $C_\sigma = 1.13$. For each event and each strain gauge, the evolution of the shear strain during the inter-event period is determined as linear (L) or sublinear (SL). All the events of the empty-hole experiment in (**a**) are linear (L). Black dotted lines in (**b**) are a fit of the shear strain over the last points before a stress drop. **c** Counts of the ratio of the number of events exhibiting a sublinear evolution of the shear strain $N_{SL}$ for each strain gauge normalized by the total number of events $N_{event}$, for empty-hole experiments (left, dark blue) and granular experiments (right, light blue), for the same ranges of $C_\sigma$ as in Fig. 4c, mentioned on the plots. **d** Comparison of the slipping patch length, $<l_{slip}>$, corresponding to the average length of the slipping patch obtained from the data shown in (**c**), normalized by the hole length, $l_{hole} = 30$ mm, and the nucleation distance from the center $<d_{nuc}>$, normalized by the hole half-length $l_{hole}/2$. Circle corresponds to empty-hole experiments and diamonds to granular experiments. Error bars corresponds to the standard deviations of the distributions in (**c**) for the $y$-axis and in Fig. 4c for the $x$-axis. Source data are provided as a Source Data file.

explained by the propagation of ruptures arrested by heterogeneities, which act as a barrier[19–21].

## Discussion

What causes microcontacts to slide instead of breaking dynamically? There may be two possible mechanisms involved. The first mechanism is the creep of the microcontacts loaded by the slipping granular patch at a very low rate, of the order of few μm/s (Fig. 2d). The creep regime has been observed for several experimental systems[48,49]. In fact, frictional contacts age with time, strengthening the interface, while low-rate shear loading induces plastic flow of the contacts[50]. The comparable timescales between these two processes may lead to creep rather than dynamic failure of the microcontacts. The second mechanism is the shear-induced dilatancy of the granular patch. A dense granular material dilates, i.e. expands, when sheared[51], which can locally release the normal stress on the surrounding areas, leading to a transition to stable sliding[52]. In our experiments, we observe a local reduction in normal stress near the edges of the patch as the normal load imposed on the patch increases, although it never disappears (Figs. 1c and 6b). The local reduction in normal stress could explain the transition from stick-slip to steady sliding[52,53]. This last mechanism

emphasizes the importance of considering the mechanical behavior of heterogeneities in models.

This study highlights an interaction mechanism between a partially coupled, slowly-slipping zone and coupled, locked zones. The boundaries of the partially coupled zone are not restricted to the compositional heterogeneity along the fault, i.e. the granular material, but extends through slow slip or contacts creep of surrounding areas. Is the slow sliding a manifestation of a nucleation front, as observed in experiments[22,42,43] and models[54,55]? The temporal resolution achieved in this experiment is not sufficient to characterize the slow dynamics of contacts detachment and further studies should be devoted to the slow slip region extension mechanism, particularly relevant for the prevention of seismic hazard[8,16].

Our experiments provide a way to explore the complex fault dynamics by controlling the ingredients responsible for complexity. For instance, our observations correspond to a scenario of tremors, where slow slip induces repeated microearthquakes[5]. In our study, the event amplitude, i.e. the force drop, is controlled by the length of the slipping patch, rather than the amount of slow slip. Experimental acoustic measurements would allow to relate the seismic wave content to the source properties, which are inaccessible in natural faults. This

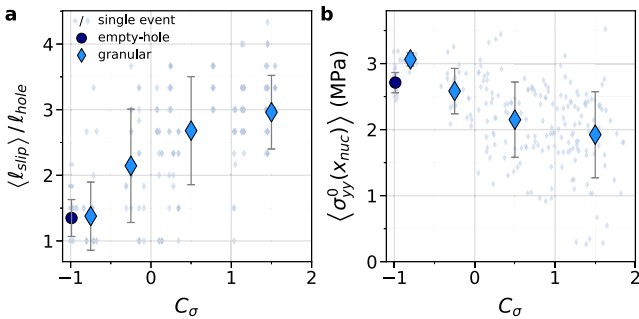

**Fig. 6 | Slipping patch length and local normal stress leading to rupture destabilization. a** Evolution of the slipping patch length, $\langle l_{slip} \rangle$, corresponding to the averaged length of the slipping patch obtained from the data in Fig. 5c, normalized by the hole length, $l_{hole} = 30$ mm, as a function of the loading contrast for the empty-hole experiments (circle) and the granular experiments (diamonds). Error bars corresponds to the standard deviations of the distributions in Fig. 5c. Measurements for each individual event appear in gray, circle or diamond. **b** Averaged values of $\sigma_{yy}^0(x_{nuc})$, the normal stress measured at the nucleation location $x_{nuc}$ of each rapid event, for each range of $C_\sigma$ used for the histograms in Figs. 4c and 5c for the empty-hole experiments (blue circle) and the granular experiments (blue diamonds). Measurements of $\sigma_{yy}^0(x_{nuc})$ for each individual event appear in gray, circle or diamond. Error bars correspond to the standard deviations of the data included in the average values. Source data are provided as a Source Data file.

study also raises the question of which fault zones should be monitored. If the slow slip zone can extend along a fault, as our results suggest, it is important to monitor the zone's evolution during interseismic phases. Finally, a quantitative estimate of a dynamic criterion for rupture propagation would provide information on the destabilization length of the slipping patch, for known fault loading conditions. Modeling the extension of a slow-slip zone may provide insightful results for this purpose.

In summary, we have shown that a slowly slipping area within a frictional interface acts as a nucleation center, which induces an early destabilization of the entire interface by lowering the stress level at which the rupture propagates, following the same trend as a rupture initiation criterion, for instance the Griffith criterion derived from fracture mechanics for pure cracks. This mechanism results in a modification of the stick-slip cycle. Our study shows that the effect of a heterogeneity on the frictional interface dynamics is to modify the rupture nucleation process. Furthermore, the use of a fracture mechanics-based description may allow the use of off-fault measurements to predict the destabilization of a slowly slipping patch. This provides further insight into how to account for fault complexity in models and improve understanding of the diversity of seismic fault behavior.

## Methods
### Experimental setup
Experiments were performed using a manual press to apply normal force via a normal displacement, and a motorized stage with speed control (20 μm/s) to apply shear force. The press consists of a rigid steel frame pressing together two rectangular blocks of poly(methylmethacrylate) (PMMA). The dimensions of the PMMA blocks are 90 mm × 150 mm × 10 mm, clamped along 20 mm, resulting in a free height of 70 mm. The blocks are cut and smoothed on an automated milling machine, then hand-polished (1 μm r.m.s. surface roughness). They are flat to within a 20 μm, checked with a confocal profilometer. A semi-elliptical shape is hollowed out in the center of each surface. The surfaces of the eye-shaped hole are periodically pierced with a 1.3 mm diameter semicircular hole to form grooves into which 1.3 mm diameter cylinders are inserted to artificially roughen the

surface. The cylinders are glued into the grooves. The same two blocks of PMMA are used for all the experiments.

### Materials
PMMA is a viscoelastic material[24] with a strain rate dependent Young's modulus, equal to 3 GPa for low strain rates and 5.6 GPa for high strain rate and Poisson's ratio $v = 0.3$. Because of this effect, the conversion of strains to stresses must be done carefully, using the static Young's modulus for any static loading and the dynamic Young's modulus for any rapid changes. We circumvent this point in this study by calculating only static stresses (Fig. 1) with the static Young's modulus, and using dynamic strain measurements to determine the slipping patch location (Fig. 4).

The granular material positioned at the interface consists of nylon cylinders (Young's modulus $E = 1.4$ GPa and Poisson's ratio $v = 0.4$). It is polydispersed with diameters of 0.4, 0.7, 0.9, and 1.3 mm in approximate volume ratios of respectively 5, 10, 35, and 50%. The cylinders are 10 mm long and are cut from straightened fishing line. Their faces are painted with a pattern that allows optical tracking of their position.

### Particle tracking method
Particle tracking is performed using a laboratory-developed image correlation algorithm with subpixel detection. Each cylinder face is manually patterned with 2 black dots on a white background. We define a template image for each individual tracked cylinder. A correlation matrix $R$ is obtained by convolution of the template with the considered image, around the initial position of the particle. The maximum value $R_{(i,j)}$ of the correlation matrix provides a measurement of the particle position with a pixel resolution. Subpixel resolution is achieved by using a three-point estimator with a Gaussian peak fit [25]. The three-point estimator is applied by considering four points around the position $(i,j)$, i.e. $R_{(i-1,j)}, R_{(i+1,j)}, R_{(i,j-1)}$ and $R_{(i,j+1)}$. The position of the maximum values of the correlation matrix $(x_0, y_0)$ is then refined by using the following relations:

$$x_0 = i + \frac{\ln R_{(i-1,j)} - \ln R_{(i+1,j)}}{2\ln R_{(i-1,j)} - 4\ln R_{(i,j)} + 2\ln R_{(i+1,j)}} \quad (3)$$

$$y_0 = j + \frac{\ln R_{(i,j-1)} - \ln R_{(i,j+1)}}{2\ln R_{(i,j-1)} - 4\ln R_{(i,j)} + 2\ln R_{(i,j+1)}} \quad (4)$$

This correlation method is performed on images of 1.3 mm diameter cylinders, corresponding to 11 pixels, and a template image. We have characterized the resolution of the method by performing a translation of the lower block without load at a constant speed of 10 μm/s (Supplementary Fig. S1a). A comparison between the imposed and measured displacement using the sub-pixel particle tracking method provides a characterization of the spatial resolution of the method (Supplementary Fig. S1b). The standard deviation of the difference between measured and imposed displacement is $\sigma = 2.5$ μm. We use the $3\sigma$ value, i.e. the range that includes 96% of the measured values, as the spatial resolution of the particle tracking. We find that $3\sigma \approx 8$ μm.

We apply the subpixel particle tracking method on the 3 topmost and bottommost embedded cylinders above the hole, positioned at 3.1±0.1 mm above and below the central axis of the interface, to track the displacement of the granular interface section. Eight patterns similar to the cylinder faces are painted directly on each block, with their center at 1.4 to 2 mm above the solid-solid sections of the interface, allowing for displacement tracking at 4 locations along this section. The tracking of the cylinders is performed using image correlation on 112 × 1280 pixels' images, recorded at 100 frames per second throughout the entire experiment.

## Slip measurements

The position recorded for the measurement of the inter-event sliding is calculated by detecting the events, and comparing the average position of the cylinders faces before and after each event, in each section of the interface. The total slip of each section $l$ of the interface $\delta_{tot}^{\{l\}}(t)$ is calculated as:

$$\delta_{tot}^{\{l\}}(t) = \left(x_{bot}^{\{l\}}(t) - x_{bot}^{\{l\}}(t_0)\right) - \left(x_{top}^{\{l\}}(t) - x_{top}^{\{l\}}(t_0)\right) \qquad (5)$$

where $x_{top,bot}^{\{l\}}(t)$ is the average position of the top/bottom grains of the corresponding section of the interface and $t_0$ is the initial time of the experiment. To compute $\delta_{IE}^{\{l\}}(t)$ we define the detected time for the $i^{th}$ event $t_i$, $\tau^-$ and $\tau^+$ such that $\delta_{tot}^{\{l\}}(t_i - \tau^-)$ and $\delta_{tot}^{\{l\}}(t_i + \tau^+)$ correspond to the total displacement before and after the $i^{th}$ event, with $\tau^- = 0.05$ s and $\tau^+$ ranging from 0.15 s to 0.5 s, depending on the duration of the inter-event period, to ensure that the system has stopped shaking. We calculate the slip occurring during the inter-event period following event $i$ as $d_i^{\{l\}} = \delta_{tot}^{\{l\}}(t_{i+1} - \tau^-) - \delta_{tot}^{\{l\}}(t_i + \tau^+)$. The cumulated inter-event slip $\delta_{IE}^{\{l\}}(t_i)$ is defined as:

$$\delta_{IE}^{\{l\}}(t_i) = \sum_{m=1}^{i} d_m \qquad (6)$$

We define the normalized cumulated inter-event slip $S^{\{l\}}(t_i)$ as the ratio of the cumulated inter-event slip to the total slip (occurring during inter-event and rapid events) experienced by the interface just prior to event $i$+1th:

$$S^{\{l\}}(t_i) = \frac{\delta_{IE}^{\{l\}}(t_i)}{\delta_{tot}^{\{l\}}(t_{i+1} - \tau^-)} \qquad (7)$$

The value taken for $S_f^{\{l\}}$ is calculated as the median of the last 5 points of $S^{\{l\}}$ if the experiment includes more than 5 events, or as the median of all the events otherwise. The error bars on $S_f^{\{l\}}$ are calculated as their standard deviation.

To ensure that the spatial resolution of the slip measurements, emanating from the subpixel particle tracking, is sufficient to provide valid cumulated inter-event slip measurements, we have mimicked a stick-slip cycle with inter-event sliding, consisting of 6 repetitions of the following sequence: 100 μm at 20 μm/s, 100 μm at 10 mm/s (quasi-instantaneous step), 5 s stops (Supplementary Fig. S1c). As expected, the cumulated inter-event slip $\delta_{IE}(t)$ increases by increments of 100 μm after each inter-event period and the normalized value $S^{\{l\}} = 0.5$, as the same amount of slip is experienced during the inter-event periods and the rapid events (Supplementary Fig. S1d). Note that the slight difference between the measured value ($S = 0.495$) and the expected value ($S = 0.5$) is due to the fact the inter-event slip $\delta_{IE}$ is measured between 150 to 500 ms ($\tau^+$) after the rapid event and 50 ms before the next rapid event.

## Force, strain, and stress measurements

Force sensors are embedded in the press frame and allow for low-frequency force acquisition at 315 Hz. Strain gauges rosettes are installed on one face of the upper solid block, at 1.5–2 mm from the interface along the solid-solid sections, and at 6 mm from the central axis of the interface above the granular section, i.e. the elliptical hole. We use 10 rosettes, 4 and 5 above each of the 2 solid-solid sections, $10 \pm 0.25$ mm apart, and one above the granular section of the interface. They allow for both a continuous acquisition of the loading profile at 315 Hz and a fast burst acquisition of the deformation tensor during a seismic event at 4 MHz. The strain gauges are 350 Ω resistances with a 1.84 gauge factor, amplified by a factor of 500 using two Anderson loops managing 5 rosettes each[31].

The stresses are calculated from the strains using a plane stress hypothesis, justified by the thinness of the blocks. PMMA is a viscoelastic material, but considering that all our loading phases are quasi-static, we use the low-frequency value of the Young's modulus. An accelerometer, connected to a monostable multivibrator, is used as a trigger to detect the slip events, initiate high speed acquisition of the strain gauges and forces signals, and compute the mean frequency for each experiment.

In this study, 34 experiments have been performed, including 7 empty-hole experiments for 22 rapid sliding events, and 27 granular experiments for 269 rapid sliding events. The loading contrast, $C_\sigma$, defined for each experiment consists on an average of the loading contrast before each sliding events of the experiment. The errorbars are computed as the first and third quartiles of the $C_\sigma$ per event distribution.

The stick-slip frequency is computed as the mean of $\{1/T_i\}_i$ with $T_i$ the time between event $i$ and $i$+1. The errorbars are computed as the first and third quartiles of this distribution.

## Correction for the effect of system rotation on interfacial slip measurements

During the loading phase, due to the relative flexibility of the press frame, rotation occurs between the interface direction and the camera horizontal axis $x_{cam}$ (Supplementary Fig. S2a). It amounts to a maximum of 0.2′ per sec, corresponding to up to 0.2° between two events in the solid-solid experiments. This rotation $\theta$ can add a maximum of 20 μm of apparent relative displacement $\delta x$ between the top and bottom cylinders faces. This rotation applies equally to high frequency and low frequency stick-slip cycles and should therefore not modify our conclusions. However, to ensure that it has no influence on the slip measurements, the rotation angle $\theta(t)$ between the interface and the camera horizontal axis $x_{cam}$ is calculated at any time, as well as the interface axis $x_{int}(t)$. The position of the cylinders' pattern is then projected on this new axis. To compute the angle $\theta(t)$ and $x_{int}(t)$, we compute a linear fit of the top and bottom cylinders patterns relatively to $x_{cam}$, yielding $y_{top}(t) = a_{top}(t) \times x_{cam} + b_{top}(t)$ and $y_{bot}(t) = a_{bot}(t) \times x_{cam} + b_{bot}(t)$. Then $x_{int}(t)$ is calculated as the mean of the two quantities. This solution ensures that the interface is always correctly fitted.

## Elastic deformations at the cylinder's patterns height

The difference in height between cylinder faces above the granular section and the solid-solid sections, respectively 3.1 and 1.5–2 mm, can induce errors in the inter-event sliding computation due to the shear deformation of the blocks (Supplementary Fig. S2b). However, this has been verified to be a negligible correction that does not affect the measurements and conclusions of this study. We evaluate the amplitude of this effect by considering the most unfavorable case, where the shear force attains $F_s^{max} = 1500$ N, corresponding to the maximal shear stress $\tau_{max} = F_s^{max}/A = 1500/1.5 \times 10^{-3} = 10^6$ Pa, where $A$ is the apparent contact area. The induced displacement $\delta x$ between an upper and a bottom cylinder faces is $\delta x = 2h\tau_{max}/G$, where $h$ is the distance between the cylinders faces and $G = E/2(1+\nu)$ the shear modulus, yielding to $\delta x^{gran} = 9$ μm and $\delta x^{solid} = 5$ μm. As the values measured for the inter-event slip range from 0 to 300 μm, with a usual 10 μm error bar, this source of error is negligible.

## Data availability

The experimental data generated in this study have been deposited in the figshare repository under accession code https://doi.org/10.6084/m9.figshare.25488571[56]. Source data are provided with this paper.

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

## Acknowledgements
This work was supported by funding from French National Research Agency (ANR) under Grant No. ANR-20-CE30-0010-01 (project Dis-RuptInt) (E.B.), from the IDEXLYON Project of the University of Lyon as part of the Investissements d'Avenir Program (ANR-16-IDEX-0005) (E.B.), and from the Fédération de Recherche André-Marie Ampère (FRAMA) (E.B.). The authors thank Mokhtar Adda-Bedia and Cécile Lasserre for helpful discussions and comments.

## Author contributions
E.B. conceived the project. Y.F. and E.B. designed the experiment, Y.F. performed the experiments. Y.F. and E.B. analyzed the data and wrote the paper.

## Competing interests
The authors declare no competing interests.
