## [Peer Review File · Nature Communications]

Experimental evidence of seismic ruptures initiated by aseismic slipEditorial note: Parts of this Peer Review File have been redacted as indicated to remove third-party material where no permission to publish could be obtained.

REVIEWER COMMENTS

Reviewer #1 (Remarks to the Author):

I reviewed the manuscript "Experimental evidence of seismic ruptures initiated by aseismic slip" by Fauve and Bayart. The manuscript presents a novel experimental setup to study stick-slip at the frictional interface. It comprises two PMMA blocks with a central region filled with a two-dimensional amorphous solid (comprised of a binary mixture of rods of different diameters). Three different high-frequency measurements are performed: the remote force, the strain close to the interface at discrete locations (using strain Rosetta), and the slip (using digital image correlation).

The authors characterize the amorphous region by the "Load Contrast" which is roughly the fraction of normal load carried by the amorphous region. The authors present that the amorphous region is increasingly slipping slowly with Load Contrast, causing a dramatically changed stick-slip response (Fig. 1c). Furthermore, the authors show that "slip" (the macroscopic drop of remote load) starts by a ballistic rupture that nucleates from a region of extent ℓ , whereby ℓ increases with load contrast (consistent with the decrease of the stick-slip amplitude with increasing Load Contrast). Curiously, ℓ exceed the extent of the central amorphous region.

My take on what we learn from these observations is as follows.

1. A ballistic events nucleates from slowly slipping patch of size $\ell \sim (\Delta \sigma)^{-\nu}$. Such a diverging length-scale has often been suggested theoretically, but not often shown experimentally. Unfortunately, though, the much debated exponent is not quantified. A key question for the authors is why this measurement is not explicitly shown, and if there is hope that their setup could (ever) measure the exponent.
2. The amplitude of slow slip increases with Load Contrast. Qualitatively, this is consistent with a smaller stick-slip amplitude: a higher fraction of the potential energy around the

slowly slipping region is released prior to slipping, increasing the stress on the "locked" region of the frictional interface.

In itself, it is interesting to carefully show both facts. Point 1 is theoretically much debated. Point 2 is a hot topic in the seismological community. However, At the same time, these results are qualitatively consistent with much accepted mechanics, and, unfortunately, apparently too tricky to quantitatively interpret to refine existing theory. Much to the credit of the authors, there is no over-interpretation.

A curious result is that the extent of the nucleation region, ℓ , can exceed the extent of the amorphous region. However, based on Fig. 1c I would say that this might be due to opening of the interface: the region around the amorphous region is simply unloaded by forcing rods in. The explanation is not really addressed by the authors.

In summary, I am a bit mitigated. The setup is novel and timely. However, quantitative answers to key questions are not yet made.

Furthermore, I have quite a number of detailed points:

- * The images are fully rasterized and often not of very high precision.
- * I.29: "localized asperities" = part of a single fault?
- * I.30: "very small events": as far as I understand, the amount of slip can be quite substantial (as the authors show themselves)? The authors probably refer to their slowness?
- * I.34 (and elsewhere). The authors refer to "uncoupled" and "coupled" zones. I have no clue what is meant. Neither before, nor after reading the manuscript.
- * I.38: "no assumptions about the frictional properties": assumptions are *always* made.
- * I.41: what boundaries exactly?
- * I.61: z is depth: I do not find this very logical; I would not introduce another ℓ_{cyl} .
- * I.62: binary mixture of rods: what volume fraction?
- * I.66: the PMMA material response should be better discussed (in a footnote). Listing the range of stiffnesses is too vague.
- * I.75: fps = Hz for consistency?
- * Fig.1d and text: both ΔT and its inverse f_{ss} are used. For simplicity, I would

suggest using only the former. Personally, I would even convert this into a macroscopic friction coefficient $\Delta \mu = \Delta F / N$.

* Fig.1b(left) and text: the term solid-solid confused me a bit, isn't "empty" easier?

* Fig.1b(right): needs an indication of density of rods or Load Contrast.

* Fig.1d: σ_{yy}^s is in the text $\sigma_{yy}^{\text{solid}}$ please unify

* Fig.1d: the line drawn line is not discussed and does not appear to be a fit. I would remove it. If the authors feel strongly about a support line, they could consider a diagonal over the entire figure to avoid the false suggestion of a fit or theory.

* Fig.1d(right): the system is not in a steady-state. However, and it seems hard to extract a uniform response. Undoubtedly, the authors are careful. However, the authors should emphasize what data they use where and why? In addition, the authors should discuss possible reasons for the response.

* Fig.1, caption: "and cylinders ... are inserted into the hole to roughen it". I do not understand what is meant exactly. Are they some glued?

* Fig.1, caption, and text: x_{SG} : why is there a subscript and what does it mean? Could it not be omitted?

* I.90: I am curious to the reasoning behind the definition of the Loading Contrast.

Personally, I would have probably used $\sigma_{yy}^{\text{gran}} / \langle \sigma_{yy} \rangle$ as it is simply the fraction of load carried by the amorphous region, nicely defined between 0 and 1.

* I.96-99: From Fig.1d I would say that $\Delta T = c \sigma_{yy} + f(\text{LC})$: both with and without the amorphous region there is normal force proportionality, it just seems that an offset changed.

* Fig.2: left and right have different limits, I would put labels on both to create clear visual separation.

* Fig.2 bottom: remove the 10^{-2} scale to unify with Fig. 3

* Fig.2 bottom-left: why are there two lines??

* Fig.4a: why does the bottom panel does not show the front.

* Fig.4b: The slopes seem to be drawn a bit arbitrary. Instead, they could be compared to their known macroscopic value.

* Fig.4b,d: not the same measures, but the same label.

* Fig.212-215: I do not get this argument

* I probably missed it in details, but I did not clearly find what the error-bars represent (how are statistics collected: How many experiments? Are samples re-used? Are they reproducible?).

[reviewer name redacted]

Reviewer #2 (Remarks to the Author):

Review of "Experimental Evidence of Seismic Ruptures Initiated by Aseismic Slip" by Faure and Bayart

General Comments:

The paper presents very interesting and novel experimental measurements of fault dynamics in analogue experiments. The novelty lies in the introduction of a granular patch in the middle of the fault setup. The paper deals with the interaction between the slow slip region and the dynamics of initially frictionally locked regions. The text is clear, well-referenced. However, two aspects of the manuscript are somewhat speculative: the location and size of the so-called seed cracks and their extension. I am uncertain whether the displacement measurements have the resolution to adequately inform on these matters, which impacts the discussion. Secondly, the nature of crack propagation and its correspondence to the Griffith criterion, while highly probable, is not explicitly demonstrated in the manuscript as G_c is not computed. Moreover, the discussion regarding the possible impact of the work on understanding the relation between slow slip events and dynamic earthquakes is currently very limited and could be further developed. I recommend major revision.

Specific Comments:

- Line 37: Slow slip areas are also considered as low-stress zones with high pore fluid pressure (Passelegue et al., 2020, for experimental evidence).
- Line 55: Slip displacement measurement should be briefly explained in the main text and not only in the methods section. The resolution of slip measurement should also be

addressed as it significantly impacts what can be observed during the experiments.

- Figure 1: Loading contrast (LC) should be carefully distinguished, as it can be confused with the critical nucleation length noted as L_c in the literature. The definition of LC should be included in the figure caption of Figure 1.

- Line 84: Is the entire fault slipping or just a portion of it?

- Line 99: What is the size of the rupturing patch? Where does the slip occur along the interface?

- Line 102: Please specify how slip measurements are taken.

- Figure 2d: Explain the normalized cumulative inter-event slip variation at early times.

- Figure 3: Clarify the color code in the legend.

- Line 135: Can the granular patch serve as a trigger for rupture nucleation due to different frictional properties between the granular patch and the rest of the sliding interface?

- Line 147: The sentence "the distribution is robust when... is not imposed by the interface" needs clarification.

- Figure 4: Trace the rupture front in dashed lines on Figure 4A and add rupture velocity to the figure. The definition of sublinear evolution is unclear and arises much later in the text.

- Line 150: The statement about the nucleation point shifting as LC increases appears speculative based on the data presented in Figure 4b. There isn't a noticeable change in the nucleation location on the histograms as a function of LC.

- Line 154: Based on previous comments, there is no clear evidence that the length of the central slipping zone increases with LC. Is this visible in the slip measurements? Do you have the resolution to observe it?

- Line 156: The paper discusses a crack governed by the Griffith criterion, but this is not demonstrated in the paper. Did you calculate GC?

- Line 156: Please define "seed crack" as it is not trivial.

- Line 162: "Low strain measurements allow us to overcome this lack of resolution." Please explain further and provide data.

- Line 173: To compare with natural seismicity, it would be interesting to compute and present the fault coupling.

- Line 173: The statement about "the larger the central... the longer the slipping zone" seems speculative based on the data presented in the paper. I'm not sure the authors have the slip resolution to make that claim.

- Line 180: Authors suggest that granular material induces slow slip events. Another hypothesis is that slow slip events are introduced by stress heterogeneity induced by the granular material, not the material itself. Is this plausible?
- Line 187: The definition of the seed crack should precede this point.
- Line 190: Nucleation, as defined in finite fracture mechanics, is not only governed by energy but also by stress. How is this accounted for here?
- Line 216: The authors mention that the granular material dilates. Please show the normal displacement measurements; otherwise, this is speculative.
- Line 211: The authors propose two mechanisms to explain creep: 1) low rate of shear loading and 2) dilatancy. Can the frictional properties of the granular rods alone explain creep?
- Lines 222-223: The statement about "the boundary of the uncoupled zone... extends through slow slip or contacts creep of the surrounding areas" is not fully demonstrated in the manuscript as per previous comments.
- Line 230: The Griffith criterion has not been proved in the manuscript, and G_c was not computed. This statement seems speculative and requires rephrasing.
- The implications of the results for natural cases should be developed further.

Reviewer #3 (Remarks to the Author):

Review of NCOMMS-24-00631:

General Assessment:

The manuscript entitled "Experimental evidence of seismic ruptures initiated by aseismic slip" by Y. Faure and E. Bayart presents an experimental study on the influence of spatially-distributed stable sliding on the initiation of dynamic shear ruptures. The authors use experiments of direct shear performed on a model 1D (thin) fault system located between two PMMA plates. The main novelty lies in the presence of a granular media at the center of the fault, consisting of nylon cylinders. This permits to initiate a slowly-creeping region at the fault center and to investigate its impact on the nucleation of laboratory earthquakes.

The key results presented in this manuscript are:

- The stable sliding region expands quasi-statically along the solid/solid (PMMA/PMMA) interface before the onset of the instability.
- The propensity of the sliding region to penetrate the solid/solid interface strongly depends on the heterogeneities of normal stress along the fault.
- In turns, it modifies the frequency of the seismic (stick-slip) events, promoting low-magnitude tremors over more energetic ruptures.

The research conducted in this work is of notable quality. While the text is generally well-written and structured, there are areas requiring clarification and improvement to enhance readability and precision. Moreover, it is not clear to me whether the impact of stable sliding on the onset of dynamic ruptures is controlled by the fault frictional heterogeneity or rather by the gradient of normal stress generated along the fault. I hope to hear the authors' opinion on the following:

- Abstract – The abstract is overall well-written and engaging, but I have some comments.
 - Comment 0.1: The use of "heterogeneity" lacks specificity. The authors should delineate the types of heterogeneities considered (stress, composition, slip distribution, etc.).
 - Comment 0.2: The author writes at line 18 that the heterogeneity "[reduces] fault shear resistance". I guess that they refer to the decrease in macroscopic friction F_s/F_n observed in their experiments, rather than a microscopic change in friction/shear strength. This concept of "macroscopic stability" may need to be clarified.
 - Comment 0.3: The "fracture concepts" used in the paper are rather limited. If I am right, they consist in the sole Griffith-like criterion in the section entitled "Slowly slipping area acting as a nucleation seed". I am not sure that they permit to "isolate the very origin of earthquake nucleation and slip dynamics in seismic fault", in contrast with previous works performed by the corresponding author.
 - Comment 0.4: The last sentence of the abstract seems to another overstatement regarding the content of the paper. The study conducted by the authors is of high quality and does not need overselling. I am nonetheless totally in line with the tone used in the "Discussion" section of the papier.
- Laboratory-fault experiment

■ Comment 1.1: (line 75) How is calculated the 8 microns precision for the DIC measurements? The pixel size is $\sim 150\text{mm}/1280\text{pix} = 120$ microns. 8 microns is smaller than the error traditionally-associated with DIC measurements (Bornert et al., 2009, doi:10.1007/s11340-008-9204-7).

■ Comment 1.2: (Figure 1) As I am familiar with the work done by the corresponding author, I guess that the diamond shaped points correspond to experiments without center hole. This could be more clear.

■ Comment 1.3: The authors investigate here the inter-event duration. It may be interesting to look at the scalar moment release M , in order to evaluate the potency of the seismic ruptures. This can be done thanks to the displacement measurements made in this study.

• Slow slip and stick-slip cycle

■ Comment 2.1: As in Figure 1, the distinction between solid/solid experiments with (called reference solid/solid experiments) and without (called solid/solid experiments) hole is not clear. For example, I am not sure which kind of experiments is mentioned at lines 92-93 and 97-98.

■ Comment 2.2: In Figure 2c, I don't understand what are the dark blue and light blue lines. Is S^{gran} of the reference solid/solid experiment is defined from the slip displacement measured on the contour of the hole?

■ Comment 2.3: At lines 122-123, the authors argue that the mean stick-slip frequency $\langle f_{ss} \rangle$ may be a relevant control parameter for the occurrence of rapid slip events. However, in their case, the energy content of the seismic events looks almost constant at a given LC. How would this observation fare in realistic conditions, when stick-slip events distributes over multiple time and energy scales?

• Expansion of the slipping zone with the normal load

■ Comment 3.1: For clarity purpose, it would be better to show the nucleation location (defined at lines 144-145) on Figure 4a.

■ Comment 3.2: Can the authors confirm that each F_s/F_n drop observed on Figure 1c correspond to the rupture of the entire frictional interface (as suggested at lines 142-143)? I just want to make sure that the similarities in Figure 1c.iv and Rubinstein et al. (2007, doi:10.1103/PhysRevLett.98.226103)'s Figure 2b cannot be attributed to partial ruptures of

the interface.

■ Comment 3.3: I have worked on a similar experimental setup and I know how difficult it can be to accurately set a time of departure from linearity on strain gages data. For a better understanding of Figure 4d, it would be interesting to show each linear/non-linear time intervals selected in the example of Figure 4c.

■ Comment 3.4: In the case where the pre-slip is large (granular experiments, large LC), is it possible to compare the DIC predictions of the sliding region to that inferred from the strain gages ?

■ Comment 3.5: Would it be possible to superimpose (maybe just in the answer) to Figure 4a the estimated sliding region before nucleation, as predicted from the procedure of Figure 4c? How does a nucleation event look for cases at larger LC? Other examples of Figure 4a could be added as supplemental material.

■ Comment 3.6: How would Figure 4a looked if the portion [75mm, 135mm] was creeping before nucleation (~ -0.1 ms)? Can we be sure that rupturing of the [75mm, 135mm] portion of the fault is not a secondary rupture (Shi et al., 2023, doi:10.1038/s41467-023-44086-1)?

• Expansion of the slipping zone with the normal load

■ Comment 4.1: As shown by the authors, the presence of the granular material introduce two effects: (1) early sliding at the fault center, and (2) the creation of a positive stress gradient. I would hypothesize that (1) without (2) would create early dynamic ruptures with decreased energetic content due to lower applied shear force F_s . But (2) promotes stable sliding (decrease of the crack driving force) which permits to dissipate further the energy injected to the system through aseismic slip. Am I interpreting the authors' results correctly?

■ Comment 4.2: Can the author discuss potential outcomes if the creep patch extends to the fault's ends? (See recent work for some insights on this question: <https://meetingorganizer.copernicus.org/EGU23/EGU23-15559.html>)

• Discussion

■ Comment 5.1: The authors do not mention the positive normal stress gradient observed for large LC (i.e. creeping fault). It looks to me that this effect (which may be related to structural/elastic heterogeneities) may be important to explain stable sliding. Can they

comment on this?

- Methods

- Comment 6.1: There is no much information on the optical tracking method. If DIC was used, what was the subset size? How is the correlation performed (manual/industrial software)?

Additional note: There is no information on data availability. I am advocating for open science and would be glad that the authors put their experimental data in open access (at the very least for the data presented in the manuscript).

Minor formatting comments:

- line 66 "Poisson ration" -> "Poisson's ratio"
- line 189 σ_c -> τ_c
- line 241 $3 < E$ -> $3 < E$

To all our referees

We would like to express our gratitude for the insightful comments and constructive suggestions provided by the three referees. These have significantly strengthened our study and improved the clarity and accuracy of our manuscript. We believe that the modifications made, which were based on the referees' suggestions, have notably enhanced the overall quality of the manuscript.

In response to the reviewers' comments, we have revised the manuscript, marking all changes in red for easy reference. Below, we present a comprehensive point-by-point response to the referees' comments, with their remarks in black, our responses in blue, and the resulting additions to the manuscript highlighted in red. We are confident that all concerns raised by the referees have been effectively addressed.

In case the figures in the revised version of the manuscript do not appear in an acceptable resolution, we have provided them in png format using the following link:

<https://mycore.core-cloud.net/index.php/s/yulcbYpMFxFgWPx>

Reviewer #1 (Remarks to the Author):

Referee: I reviewed the manuscript "Experimental evidence of seismic ruptures initiated by aseismic slip" by Fauve and Bayart. The manuscript presents a novel experimental setup to study stick-slip at the frictional interface. It comprises two PMMA blocks with a central region filled with a two-dimensional amorphous solid (comprised of a binary mixture of rods of different diameters). Three different high-frequency measurements are performed: the remote force, the strain close to the interface at discrete locations (using strain Rosetta), and the slip (using digital image correlation).

The authors characterize the amorphous region by the "Load Contrast" which is roughly the fraction of normal load carried by the amorphous region. The authors present that the amorphous region is increasingly slipping slowly with Load Contrast, causing a dramatically changed stick-slip response (Fig. 1c). Furthermore, the authors show that "slip" (the macroscopic drop of remote load) starts by a ballistic rupture that nucleates from a region of extent ℓ , whereby ℓ increases with load contrast (consistent with the decrease of the stick-slip amplitude with increasing Load Contrast). Curiously, ℓ exceed the extent of the central amorphous region.

My take on what we learn from these observations is as follows.

1. A ballistic events nucleates from slowly slipping patch of size $\ell \sim (\Delta\sigma)^{-\nu}$. Such a diverging length-scale has often been suggested theoretically, but not often shown experimentally. Unfortunately, though, the much debated exponent is not quantified. A key question for the authors is why this measurement is not explicitly shown, and if there is hope that their setup could (ever) measure the exponent.

Response: In our experiments we observe two types of phenomenology, slow sliding in the central part of the interface and rapid ruptures along the locked interfaces. The slow sliding events do not correspond to a rupture event, at least with the temporal resolution of the measurements we are performing. Rapid ruptures, on the other hand, are known to be shear cracks, according to the literature. Therefore, the relation between $\Delta\sigma$ and ℓ during crack propagation is dictated by fracture mechanics (Svetlizky et al 2017 [ref 14]).

In our understanding, the power law $\ell \sim (\Delta\sigma)^{-\nu}$ mentioned by the reviewer is of interest for subcritical ruptures, i.e. ruptures of asperities that propagate for a certain distance below the critical size for macroscopic destabilization into a single rapid rupture. In this study, we are interested in the destabilization criterion, and we do not have access to the microruptures that precede this stage.

In our study, the $\Delta\sigma$ of interest to test in the relation between $\Delta\sigma$ and ℓ is the stress drop required to destabilize the initial seed crack. From a fracture mechanics point of view, it is possible to derive an initiation criterion for our geometry, but it is very challenging to measure due to the lack of spatial resolution in the strain measurements.

What is the correct way to derive such a criterion? It is to determine the static stress intensity factor, K_{II} , at the tip of our seed crack, i.e. the slipping patch. In our geometry this problem was solved by Muskhelishvili in 1953 and the calculation can be found in Leblond (Mécanique de la rupture fragile et ductile, in French). We reproduce the result here to illustrate the complexity of the implementation in our experiment and to state the reasons why we cannot provide quantitative results.

[figure redacted]

Figure for comment 1: Configuration of a crack in a infinite medium, from Leblond (Figure 3.3)

For a mode II crack with a non-uniform loading $q(t)$ where t is the coordinate along the crack, the SIF in $x_1 = \pm a$ writes:

$$K_{II} = \frac{1}{\sqrt{\pi a}} \int_{-a}^{+a} q(t) \sqrt{\frac{a+t}{a-t}} dt$$

As we consider a frictional crack, the residual stresses should be subtracted from the loading. The integral in the K_{II} expression has to be considered on both the granular patch, where the residual stresses are $\tau_g(x)$, and on the neighboring solid-solid areas where the residual stresses are $\tau_r(x)$. We are not able to perform this measurement because of the lack of spatial resolution of the strain/stress measurements. Indeed, τ_g , τ_r and q have a strong spatial dependence in our system.

Following this view, the local stress drop responsible for the seed crack propagation strongly depends on the residual stresses within the slipping patch. As there is a strong dependence between residual stresses and normal stress and that normal stress distribution is our control parameter in the experiments, there is no expectation in this system to find a simple power law between the local stress drop and the length of the slipping patch.

What can be done is to relate the mean macroscopic drop of stress, i.e. the drop of shear force per unit area, and the length of the slipping patch. However, from a fracture mechanics point of view, it is not physically justified. Indeed, the total drop of force is an overall response of the system to the first rupture, the reflections and the subsequent inertial sliding. We provide these data as part of the response, but we do not believe we should learn anything from it, for the reasons stated above.

Figure for comment 1: Macroscopic stress drop as a function of ℓ_{slip} for all events (blue dots), binned by value of stress drop (blue diamond) and by value of ℓ_{slip} (gray diamonds).

However, we believe that the overall message of the study does not suffer from the lack of quantitative verification of the crack initiation criterion, or the quantitative relationship between the nucleation length and the stress drop. In fact, we demonstrate an interaction mechanism between a slipping patch and the neighboring locked frictional interfaces. The slipping patch acts as a nucleation center, and the longer it is, the more frequent the rupture nucleation. This behavior follows the same tendency as a Griffith criterion for a pure crack. We believe that future experiments with our setup will be able to provide insights into the relation $\ell \sim (\Delta\sigma)^{-\nu}$.

We have modified the paragraph discussing the Griffith criterion to make clearer what can be verified based on our experimental data. It now reads:

1.245: “Our interpretation is that the central slipping zone acts as a nucleation center, or a seed crack, along the interface. In Mode II, a seed crack is an area where the shear stress has been released by interfacial slip. The initiation criterion of a crack, the Griffith criterion, is $G_S = G_c$, where G_S is the static energy release rate and G_c is the fracture energy of the material. For a seed crack embedded in an infinite solid under a homogeneous loading, the Griffith criterion leads to an expression of the critical stress τ_c for which a crack of a given length l is destabilized, $\tau_c = \sqrt{G_c E / (1 - \nu^2) l}$. Because we are considering a frictional crack, where stresses are relieved to the residual stress value rather than zero³⁵, with spatially dependent shear loading due to compositional heterogeneity and a normal stress gradient induced by the loading contrast (Fig. 1b), a quantitative estimation of G_S is challenging and cannot be performed in our experiment. However, we can verify that the response of the system to the presence of the slipping patch follows the same trend as in the presence of a pure seed crack.”

Referee: 2. The amplitude of slow slip increases with Load Contrast. Qualitatively, this is consistent with a smaller stick-slip amplitude: a higher fraction of the potential energy around the slowly slipping region is released prior to slipping, increasing the stress on the "locked" region of the frictional interface.

In itself, it is interesting to carefully show both facts. Point 1 is theoretically much debated. Point 2 is a hot topic in the seismological community. However, At the same time, these results are qualitatively consistent with much accepted mechanics, and, unfortunately, apparently too tricky to quantitatively interpret to refine existing theory. Much to the credit of the authors, there is no over-interpretation.

Response: This stress increase is our indirect claim. The description of the slipping patch as a shear crack implies that there is a stress concentration at the crack tip, which facilitates crack propagation. However, as we explain in the manuscript (lines 281-293), in our experiment the stress concentration does not emerge in the most trivial way, i.e. at the edges of the granular patch. The slow slip extends to the neighboring regions and it is at the edges of these regions that the stress concentration occurs.

Why not measure it directly? Because the stress distribution along the interface is heterogeneous. Therefore, each location along the interface has a different background stress (residual stress) and a different shear stress evolution during the inter-event periods. In order to extract a stress divergence that occurs during the inter-event periods, it is necessary to know all of these previous components. Calculating the stress intensity factor is equivalent to measuring the stress divergence. This can be done for a homogeneous flat interface where the homogeneity of the stress distribution can be approximated. This was done, for example, in the study by Gvirtsman and Fineberg, Nature Physics 2021 (ref 38). However, we cannot achieve this with this experimental setup because of the added complexity in the stress distribution due to the granular patch (see our previous answer).

However, we believe that our results showing that the critical stress at rupture initiation decreases with the slipping patch length do demonstrate this stress concentration effect, by construction of the crack initiation criterion. We have revised the manuscript to clarify this statement:

1.257: “In our experiment, the seed crack length corresponds to the size of the central slipping zone. The higher the loading contrast, the longer the central slipping zone (Fig. 6a) and the higher the stick-slip frequency (Fig. 1d). This is qualitatively consistent with the Griffith criterion for an infinite and homogeneous system: the longer the seed crack (ℓ_{slip}), the lower the critical stress at which ruptures initiate compared to the minimum stress reached after the interface weakening – the residual stress – resulting in the initiation of rapid slip events after a shorter loading time and thus to an increased stick-slip frequency.”

Referee: A curious result is that the extent of the nucleation region, ℓ_{nuc} , can exceed the extent of the amorphous region. However, based on Fig. 1c I would say that this might be due to opening of the interface: the region around the amorphous region is simply unloaded by forcing rods in. The explanation is not really addressed by the authors.

Response: We have verified that there is no total unloading of the interface at any location (normal stresses don't vanish) and no cylinders are entering the solid-solid interfaces. However, the partial unloading of the interface at the patch edges might be the reason for which stable sliding is observed. The local value of the normal stress may be low enough that the stick-slip criterion is no longer fulfilled. The other possible reason is a weakening of the solid contacts by a creep phenomenon, which would be induced by a very low shear rate imposed by the slow sliding of the granular patch. The reasons for the extension of the

slipping zone was initially discussed in the original manuscript, lines 211 to 219. This discussion has been now clarified in the revised manuscript:

I. 295: “What causes microcontacts to **slide** instead of breaking dynamically? There may be two possible mechanisms involved. The first **mechanism is the creep of the microcontacts loaded by the slipping granular patch at a very low rate, of the order of few $\mu\text{m/s}$ (Fig. 2d). The creep regime has been observed for several experimental systems^{40,41}. In fact, frictional contacts age with time, strengthening the interface, while low-rate shear loading induces plastic flow of the contacts⁴². The comparable timescales between these two processes may lead to creep rather than dynamic failure of the microcontacts. The second mechanism is the shear-induced dilatancy of the granular patch. A dense granular material dilates, i.e. expands, when sheared⁴³, which can locally release the normal stress on the surrounding areas, leading to a transition to stable sliding³⁵. In our experiments, we observe a local reduction in normal stress near the edges of the patch as the normal load imposed on the patch increases, although it never disappears (Figs. 1c). The local reduction in normal stress could explain the transition from stick-slip to steady sliding^{44,45}. This last mechanism emphasizes the importance of considering the mechanical behavior of heterogeneities in models.”**

Referee: In summary, I am a bit mitigated. The setup is novel and timely. However, quantitative answers to key questions are not yet made.

Response: We hope to have convinced the reviewer that our study does not suffer from a lack of quantitative results and that the interaction mechanism highlighted has never been observed in such detail before.

Furthermore, I have quite a number of detailed points:

* The images are fully rasterized and often not of very high precision.

Response: The figure integration in the word document together with the process of the journal server might be the reasons. We hope that the quality of the figures in the resubmitted version is better. If it is not the case with the revised version, the figures can be downloaded following this link:

<https://mycore.core-cloud.net/index.php/s/yulcbYpMFxFgWPx>

* 1.29: "localized asperities" = part of a single fault?

* 1.30: "very small events": as far as I understand, the amount of slip can be quite substantial (as the authors show themselves)? The authors probably refer to their slowness?

Response: The very small events referred to the SSE occurring on small portions of faults. We have rephrased this sentence, which now reads:

I. 29: “Slow earthquakes can affect a large portion of a fault, resulting in the release of a significant amount of energy. However, they can also be localized to very small portions of faults; these small events are detected indirectly by the seismic events they generate, also known as tremors³⁻⁵.”

* 1.34 (and elsewhere). The authors refer to "uncoupled" and "coupled" zones. I have no clue what is meant. Neither before, nor after reading the manuscript.

Response: We now define it in the introduction:

1.34: “The behavior of fault zones is often described by a coupling term ϕ , where $\phi=0$ when the fault is slipping at the same velocity than the loading rate (uncoupled fault), and $\phi=1$ when the fault is fully locked during the inter-seismic period (coupled fault).”

* 1.38: "no assumptions about the frictional properties": assumptions are *always* made.

Response: Indeed, the formulation is wrong. It now reads:

1.42: “However, simple geometric complexity is sufficient to induce slow slip, without introducing any variation of the fault frictional properties over time or space¹⁰.”

* 1.41: what boundaries exactly?

Response: We agree that the sentence was unclear. It now reads:

1.47: “Understanding what sets the boundary between slow slip zones and seismic zones is a key point in improving seismic risk monitoring and prevention¹⁶.”

* 1.61: ℓ is depth: I do not find this very logical; I would not introduce another ℓ_{cyl} .

Response: The block's thickness is now w and we have removed ℓ_{cyl} .

* 1.62: binary mixture of rods: what volume fraction?

Response: This information was initially in the caption of Fig. 1 and in the methods section. We added it to the main text. It now reads:

1.69: “Different cylinder diameters, 0.4, 0.7, 0.9 and 1.3 mm in approximate volume ratio of 5, 10, 35 and 50%, are used to prevent crystallization.”

* 1.66: the PMMA material response should be better discussed (in a footnote). Listing the range of stiffnesses is too vague.

Response: We added a Materials section in Methods with extensive information about the viscoelastic behavior of PMMA.

1.349: “**Materials.** PMMA is a viscoelastic material²⁴ with a strain rate dependent Young's modulus, equal to 3 GPa for low strain rates and 5.6 GPa for high strain rate and Poisson's ratio $\nu = 0.3$. Because of this effect, the conversion of strains to stresses must be done carefully, using the static Young's modulus for any static loading and the dynamic Young's modulus for any rapid changes. We circumvent this point in this study by calculating only static

stresses (Fig. 1) with the static Young's modulus, and using dynamic strain measurements to determine the slipping patch location (Fig. 4). "

We also added in the main text:

1.74: "PMMA has a strain rate dependent Young's modulus E equal to 3 GPa for low strain rate and 5.6 GPa for high strain rate²⁴,..."

* 1.75: fps = Hz for consistency?

Response: Indeed, fps = Hz. However, it seems to us that it is unusual to use Hz for a camera acquisition rate. We have changed fps in frames per second for clarity.

1.86 and 1.387: "... 100 frames per second..."

* Fig.1d and text: both ΔT and its inverse f_{ss} are used. For simplicity, I would suggest using only the former. Personally, I would even convert this into a macroscopic friction coefficient $\Delta \mu = \Delta F / N$.

Response: We agree that f_{ss} is not necessary. However, we do prefer ΔT instead of $\Delta \mu$ to keep the notations consistent with our direct observations. In fact, we emphasize that the granular patch increases the frequency of the stick-slip cycle, and it seems to us that it is convenient for the reader to provide temporal measurements to quantify this observation.

We now use $\langle \Delta T \rangle$ and $\langle 1/\Delta T \rangle$ in the text and have modified the inset in Fig. 3b.

* Fig.1b(left) and text: the term solid-solid confused me a bit, isn't "empty" easier?

Response: We agree with the reviewer that our definition of experiments was confusing. We now refer to:

- reference empty-hole experiments
- granular experiments with cylinders filling the hole
- classical solid-solid friction experiments with flat homogeneous solids.

We have modified the text and the captions following these new definitions.

* Fig.1b(right): needs an indication of density of rods or Load Contrast.

Response: The value of the loading contrast value now appears in each panel.

* Fig.1d: σ_{yy}^s is in the text $\sigma_{yy}^{\text{solid}}$ please unify

Response: done.

* Fig.1d: the line drawn line is not discussed and does not appear to be a fit. I would remove it. If the authors feel strongly about a support line, they could consider a diagonal over the entire figure to avoid the false suggestion of a fit or theory.

Response: Indeed, the line was a guide. We removed it.

* Fig.1d(right): the system is not in a steady-state. However, and it seems hard to extract a uniform response. Undoubtedly, the authors are careful. However, the authors should emphasize what data they use where and why? In addition, the authors should discuss possible reasons for the response.

Response: In Fig. 1d, $\langle \Delta T \rangle$ is the mean value of the periods of each event of an experiment and the error bar is the standard deviation. This information can be found in the Methods, 1.437. Obviously, solid-solid and reference experiments contain only few events (from 2 to 4), then the standard deviation is not fully significant in these configuration.

We have added data to Fig.1 and described them in the main text:

1.111: “In a classical friction experiment where two flat solid blocks are in contact along the entire interface, i.e. without a hole or a granular patch, a reduced normal load leads to smaller shear force drops and therefore to an increased stick-slip frequency. To verify that the observed frequency increase for an increasing loading contrast C_σ is not due to a decrease in the normal loading of the solid-solid sections of the interface, $\langle \sigma_{yy}^{solid} \rangle$, we compare the averaged stick-slip period $\langle \Delta T \rangle$ of no-hole experiments, i.e. classical solid-solid friction experiments with flat homogeneous solids in contact, under varying normal force, $750 N < F_N < 3000 N$, with our reference empty-hole experiments under the same range of normal force, and with granular experiments under $F_N = 3000 N$ with a varying density, hence varying $\langle \sigma_{yy}^{solid} \rangle$ and C_σ (Fig. 1d). First, the stick-slip periods of no-hole and empty-hole experiments are comparable for the same $\langle \sigma_{yy}^{solid} \rangle$, demonstrating that the presence of the hole has no effect on the stick-slip cycle under given loading conditions. Slight differences between the two datasets are due to expected variations in static friction between systems²⁶. Second, the period shortening in the presence of the granular section is greater than that observed for a corresponding normal load reduction in no-hole and empty-hole experiments, demonstrating an effect of the compositional heterogeneity on the stick-slip cycle beyond unloading.”

* Fig.1, caption: "and cylinders ... are inserted into the hole to roughen it". I do not understand what is meant exactly. Are they some glued?

Response: The solid surfaces within the hole are pierced with semi-circular holes with a diameter of 1.3mm (diameter of the thicker cylinders) in order to insert the cylinders in it. The inserted cylinders are glued in these holes.

We have revised the text:

1.71: “To force interfacial slip within the granular pile, additional cylinders are bonded into grooves machined into the solid surfaces forming the hole.”

and detailed the Methods section:

1.345: “The surfaces of the eye-shaped hole are periodically pierced with a 1.3 mm diameter semicircular hole to form grooves into which 1.3 mm diameter cylinders are inserted to artificially roughen the surface. The cylinders are glued into the grooves.”

* Fig.1, caption, and text: x_{SG} : why is there a subscript and what does it mean? Could it not be omitted?

Response: x_{SG} stands for the location of the strain gauges. The introduction of a new variable was intended to show that the strain measurements locations are discrete. We actually had an inconsistency between the figure and the caption. We have removed the x_{SG} notation in the section about high-frequency strain measurements. However, we kept it to describe our analysis of sublinear/linear evolution of quasi-static strain measurements because in this case, the behavior is “SL” or “L” for each single strain gauge. We have added the definition of x_{SG} :

1.215: “Hence, for each inter-event period, the strain gages exhibiting this sublinear evolution of $\varepsilon_{xy}(x_{SG}, t)$, where x_{SG} is the strain gauges position along the interface, are detected as a marker of local interfacial slow slip.”

* 1.90: I am curious to the reasoning behind the definition of the Loading Contrast. Personally, I would have probably used $\sigma_{yy}^{\text{gran}} / \langle \sigma_{yy} \rangle$ as it is simply the fraction of load carried by the amorphous region, nicely defined between 0 and 1.

Response: Our definition of the loading contrast allows to compare the loading of the different sections: a negative value corresponds to configuration where the granular patch is less loaded than the solid-solid sections, a zero value shows that the loading is approximately homogeneous and a positive value corresponds to a higher loading of the patch than the solid-solid sections. Your proposition would also work, but this variable would not be bounded between 0 and 1, as the loading of the granular patch may exceed the average loading of the interface.

Following reviewer #2 proposition, we have changed the notation for the loading contrast, now C_σ .

* 1.96-99: From Fig.1d I would say that $\Delta T = c \sigma_{yy} + f(LC)$: both with and without the amorphous region there is normal force proportionality, it just seems that an offset changed.

Response: Our study shows the existence of a complex interplay between the amorphous region and the seismic interfaces, and it does not seem obvious to extract a tendency for the stick-slip period. Also $\langle \sigma_{yy}^{\text{solid}} \rangle$ is an averaged quantity and how the normal stress is distributed along the interface impacts the stick-slip period (Ben-David et al, PRL 2010, ref 26).

* Fig.2: left and right have different limits, I would put labels on both to create clear visual separation.

Response: done.

* Fig.2 bottom: remove the 10^{-2} scale to unify with Fig. 3

Response: done.

* Fig.2 bottom-left: why are there two lines??

Response: In both the empty-hole and the granular experiments, we measure the cumulated slow slip separately for the solid-solid interfaces and for the hole (empty or filled). We have chosen the following color code, which is now clarified in the caption: light blue for the solid-solid interfaces, dark blue for the empty hole, and light brown for the hole when filled with cylinders. We have also changed the symbols: circle for empty-hole experiments and diamond for granular experiment.

The figure caption has been modified:

Caption of Fig. 2: “**Fig. 2: Measurements of the interfacial slip.** (a-b) Temporal evolution of the total interfacial slip of the solid-solid sections $\delta_{tot}^{solid}(t)$ (light blue line) and of the patch $\delta_{tot}^{patch}(t)$, when empty (dark blue line in (a)) or filled with cylinders (brown line in (b)), obtained from the particle tracking method for (a) an empty-hole experiment and (b) a granular experiment ($C_\sigma = 1.2$). Measurements are averaged over 4 locations for the solid-solid sections and 3 locations for the hole, empty or filled. (c-d) Temporal evolution of the normalized cumulated inter-event slip $S^{solid}(t)$ and $S^{patch}(t)$, i.e. the interfacial slip excluding the slip occurring during rapid slip events (c) for the empty-hole experiment and (d) for the granular experiment. Each dot corresponds to one measurement performed at the end of an inter-event period. Color code and symbols are the same as in (a-b). Inset: the cumulated inter-event slip $\delta_{IE}^{solid}(t)$ and $\delta_{IE}^{patch}(t)$ without normalization (same colors as in main plots).”

* Fig.4a: why does the bottom panel does not show the front.

Response: The y-axis is at the same scale for all strain gauges. In the bottom panel, there is a strain drop but smaller than for the other locations. This can be explained by a smaller rupture velocity at this specific location, or a non-trivial finite size effect where the rupture front interacts with the reflected waves.

* Fig.4b: The slopes seem to be drawn a bit arbitrary. Instead, they could be compared to their know macroscopic value.

Response: This is a valuable comment, but the answer is more difficult than it seems. The known macroscopic value is based on shear force measurements. However, due to the heterogeneity of the normal stress distribution, each strain gauge shows a specific load

response. This is a general fact in friction experiments, where the macroscopic force evolution cannot simply be related to the local stress evolution.

We have modified Fig. 5b (originally Fig. 4b) to make a clearer distinction between linear and sublinear evolution.

* Fig.4b,d: not the same measures, but the same label.

Response: It has been modified. The count of sublinear events (now in Fig. 5c) is now called N_{SL} / N_{event} and defined in the main text:

1.218: “For all experiments, the fraction of sublinear loading sequences, N_{SL} / N_{event} , is counted per strain gauges on the low frequency measurements of the temporal evolution of shear strain.”

* Fig.212-215: I do not get this argument

Response: In this section, we explore the hypotheses that explain the slow sliding of the solid interfaces surrounding the granular patch. We consider the possibility that microcontacts are creeping, i.e. put in a steady stable regime because of the low rate of the loading, induced by the slowly-slipping granular patch. We hope that the text is now clearer.

1.295: “What causes microcontacts to slide instead of breaking dynamically? There may be two possible mechanisms involved. The first mechanism is the creep of the microcontacts loaded by the slipping granular patch at a very low rate, of the order of few $\mu\text{m/s}$ (Fig. 2d). The creep regime has been observed for several experimental systems^{40,41}. In fact, frictional contacts age with time, strengthening the interface, while low-rate shear loading induces plastic flow of the contacts⁴². The comparable timescales between these two processes may lead to creep rather than dynamic failure of the microcontacts.”

* I probably missed it in details, but I did not clearly find what the error-bars represent (how are statistics collected: How many experiments? Are samples re-used? Are they reproducible?).

Response: The experiments are all performed on the same PMMA blocks. The number of cylinders inserted into the hole varies from one experiment to another. As the normal force applied to the system is approximately constant ($2800 \text{ N} < F_N < 3200 \text{ N}$), our control parameter for the cylinders density is the normal stress carried by the granular patch. The experiments are not reproducible per se, as can be seen on Fig. 3 for example. For a given value of the loading contrast, hence for a given normal stress distribution, measurements of the stick-slip period and the cumulated inter-event slip are spread. Frictional sliding is a mechanical instability, hence sensitive to details of the system, to the mechanical noise, etc.

In details, the data come from 34 experiments: 7 empty-hole experiments (22 rapid events), 27 granular experiments (269 rapid events).

We add this information in the revised manuscript:

1.77: “The blocks are pressed together with a normal displacement resulting in a normal force $F_N \sim 3000 \text{ N}$ ($2800 \text{ N} < F_N < 3200 \text{ N}$), unless otherwise specified.”

1.348: “The same two blocks of PMMA are used for all the experiments.”

1.434: “In this study, 34 experiments have been performed, including 7 empty-hole experiments for 22 rapid sliding events, and 27 granular experiments for 269 rapid sliding events”

[reviewer named redacted]

Reviewer #2 (Remarks to the Author):

Review of "Experimental Evidence of Seismic Ruptures Initiated by Aseismic Slip" by Faure and Bayart

General Comments:

Referee: The paper presents very interesting and novel experimental measurements of fault dynamics in analogue experiments. The novelty lies in the introduction of a granular patch in the middle of the fault setup. The paper deals with the interaction between the slow slip region and the dynamics of initially frictionally locked regions. The text is clear, well-referenced. However, two aspects of the manuscript are somewhat speculative: the location and size of the so-called seed cracks and their extension. I am uncertain whether the displacement measurements have the resolution to adequately inform on these matters, which impacts the discussion. Secondly, the nature of crack propagation and its correspondence to the Griffith criterion, while highly probable, is not explicitly demonstrated in the manuscript as G_c is not computed. Moreover, the discussion regarding the possible impact of the work on understanding the relation between slow slip events and dynamic earthquakes is currently very limited and could be further developed. I recommend major revision.

Response: We thank our referee for their comments on the interest and novelty of our work. Their insights and questions allowed us to significantly improve the quality of our manuscript and, we hope, to better convince them and the reader of the validity of our results. We made significant changes to the manuscript.

The most significant change, thanks to the reviewer's questions, is the introduction of a new figure (Fig. 6), which shows measurements of the evolution of the slipping patch length with the loading contrast, as well as a reduction of the normal stress at the rupture nucleation point with the loading contrast, which induces a rather limited reduction of the fracture energy at the rupture tip. Another significant revision concerns the data presented in the original Fig. 4, which have been reorganized into 2 figures (Figs. 4 and 5) for clarity and completeness. The revised Fig. 4 is devoted to measurements of rupture nucleation based on high-frequency strain signals, while Fig. 5 is devoted to measurements of the slipping patch length based on quasi-static strain measurements. We have added data quantifying the spatial shift of the nucleation location in Fig. 4 with the loading contrast and of the slipping patch length in Fig. 5.

We have also clarified the discussion of the validity of the Griffith criterion: in this study, the aim is to verify that the slipping patch acts as a seed crack and that the destabilization of the slipping patch into a propagative crack follows the same trend as a pure crack. We have made this message clearer in the revised version of the manuscript, and we also present the limitations of a quantitative validation of a crack initiation criterion.

Finally, we provide detailed information on slip measurements and calibration data in a new supplementary figure to characterize the resolution of our measurements.

More specifically, **regarding the location and size of the so-called seed crack**, the new figures (Figs. 4d, 5d and 6a) show that (i) there is a real shift of the nucleation location with the loading contrast (Fig. 4d), (ii) there is a real increase of the slipping patch length with the loading contrast (Fig. 6a) and (iii) that the two independent measurements, i.e. rupture nucleation location based on high-frequency strain measurements and evolution of the quasi-

static shear strain during inter-events, are in agreement (Fig.5d). See answer to the reviewer's comment on Line 150 and Line 173 for further details.

Regarding the displacement measurements, we do have the resolution to measure interfacial slip and we now demonstrate it in the manuscript. We are convinced that our measurements provide a proof that the central part of the interface slides during the inter-event periods and that the length of the sliding section increases with the loading contrast for the following reasons:

- The particle tracking method provides a resolution of 8 μm in displacement. Calibration data and detailed information about the algorithm are provided. We develop our answer in the reply to reviewer's comment on line 55.

- We consider the cumulated inter-event slip, a quantity commonly used in geodetic or experimental data, instead of presenting the slip distance for each single inter-event period. In fact, for high values of the loading contrast, the stick slip frequency becomes high and therefore, the slip distance during each inter-event period becomes small compared to our measurement resolution (up to 12 $\mu\text{m}/\text{event}$). The fact that the cumulated inter-event slip increases along the experiment (see for example the inset of Fig. 2d shows that the measurements are out of the noise.

- Fig. 3 shows an increase in the final value of the cumulated inter-event slip with the loading contrast. The fact that we are able to detect a trend proves that our measurements are out of the noise, otherwise the data points would be in a cloud of points.

Regarding the measurement of G_c and the verification of the Griffith criterion, our answer is threefold:

(i) Confidence in measuring G_c may be limited in our experiment, yet, thanks to the reviewer's comment, we now provide data suggesting that G_c exhibits a variation at the rupture nucleation location leading to an additional effect of reducing the macroscopic shear resistance of the fault. See our response to the reviewer's comment on line 156 for more details.

(ii) Quantitative validation of the correct crack initiation criterion for our system's geometry remains challenging, due to the lack of spatial resolution in strain measurements. See our response to the comment on line 156.

(iii) We believe that the overall message of the study does not suffer from the lack of quantitative verification of the crack initiation criterion. In fact, we demonstrate an interaction mechanism between a slipping patch and locked frictional interfaces. The slipping patch acts as a nucleation center, and the longer it is, the more frequent the rupture nucleation. This behavior follows the same tendency as a Griffith criterion for a pure crack. We have modified the main text to make this message clearer, as we agree with the reviewer that it was not sufficiently emphasized in the first version of the manuscript. Refer to our response to the reviewer's comment on line 230.

Regarding the possible impact of the work, we have added a paragraph discussing the implications of our study. Please refer to reviewer's last comment for further details.

Specific Comments:

- Line 37: Slow slip areas are also considered as low-stress zones with high pore fluid pressure (Passelegue et al., 2020, for experimental evidence).

Response: In the mentioned paper, the authors show frictional ruptures propagating at very low velocities, a result of the equation of motion of a frictional rupture (as shown in Svetlizky et al, Phys. Rev. Lett. 118, 125501). It is rather different than steady sliding. We have modified the text to mention the existence of slow fronts and distinguish them from steady sliding patches:

1.40: “Slow slip areas **under steady sliding** are commonly modeled as velocity strengthening zones⁹ that cannot rupture seismically.”

and

1.44: “**On the experimental side, slow ruptures have been shown to propagate under quasi-static conditions for very low shear loading¹⁴ or low normal loading with high pore fluid pressure¹⁵.**”

- Line 55: Slip displacement measurement should be briefly explained in the main text and not only in the methods section. The resolution of slip measurement should also be addressed as it significantly impacts what can be observed during the experiments.

Response: We have developed a particle tracking algorithm with a sub-pixel detection of the particle position. The sub-pixel resolution is achieved by using a 3-point estimator with a Gaussian peak fit. This is known to give a resolution better than 0.1 pixel, up to 0.01 pixel in the best case (Gaussian intensity distribution of the target) (ref 25, Particle image velocimetry: a practical guide / Markus Raffel et al (2007)). We have determined the spatial resolution of the tracking method by performing a translation of the unloaded lower block at a constant speed (10 $\mu\text{m/s}$). The 8 μm resolution is obtained as follows: the standard deviation of the difference between the particle displacement and the applied displacement is $\sigma = 2.5 \mu\text{m}$. We considered that the resolution is 3σ , i.e. the range that includes 96% of the measured values. These data are now presented in the new Fig. S1 and discussed in “Particle tracking methods” section of the Methods.

In addition, we provide the data of a calibration experiment mimicking a stick-slip cycle with inter-event sliding. The unloaded block experiences 6 repetitions of the following sequence: 100 μm at 20 $\mu\text{m/s}$, 100 μm at 10 mm/s (quasi-instantaneous step), 5 s stops (Fig. S1c). We calculate the cumulated inter-event sliding for this experiment and verify that the measurements are consistent with what is expected ($S^{\{I\}} = 0.5$).

1.361: a section is added to the Methods, entitled “**Particle tracking method.**” and referring to a new supplementary figure, Fig. S1a, b.

More details have been added to the “Slip measurements” section to describe our mimicked stick-slip experiment (Fig. S1c, d):

1.409: “**To ensure that the spatial resolution of the slip measurements, emanating from the subpixel particle tracking, is sufficient to provide valid cumulated inter-event slip measurements, we have mimicked a stick-slip cycle with inter-event sliding, consisting of 6**

repetitions of the following sequence: 100 μm at 20 $\mu\text{m/s}$, 100 μm at 10 mm/s (quasi-instantaneous step), 5 s stops (Fig. S1c). As expected, the cumulated inter-event slip $\delta_{IE}(t)$ increases by increments of 100 μm after each inter-event period and the normalized value $S^{\{t\}} = 0.5$, as the same amount of slip is experienced during the inter-event periods and the rapid events (Fig. 1d). Note that the slight difference between the measured value ($S = 0.495$) and the expected value ($S = 0.5$) is due to the fact the inter-event slip δ_{IE} is measured between 150 to 500 ms (τ^+) after the rapid event and 50 ms before the next rapid event.”

Following the reviewer’s request, we have also added more details within the main text:

1.84: “Interfacial slip measurements are performed using cylinder position data obtained with a sub-pixel resolution particle tracking algorithm. Particle tracking involves the correlation of images taken at 100 frames per second during an experiment with a template image. Sub-pixel resolution is achieved using three-point estimator with a Gaussian peak fit²⁵, resulting in an 8 μm resolution in displacement (Methods). Tracking is performed on the cylinders embedded in the eye-shaped hole boundaries and on similar patterns drawn on the block faces above the interface solid-solid sections, allowing measurements under the same conditions as for the granular section. The interfacial slip is measured at 4 locations along the solid-solid sections and 3 locations above the hole, by subtracting the measured position of cylinders or patterns located on the top and bottom blocks, at the same location x . Corrections for the rotation of the system during the shear loading are made (Methods).”

The new figure in the supplementary (Fig. S1) reads:

Fig. S1: Determination of the spatial resolution of the tracking method. (a) Displacement curves corresponding to a translation of the lower block (without shear resistance) at $10 \mu\text{m/s}$ imposed by the motorized stage. Instruction (blue line), block's displacement measured with an optical confocal profilometer (accuracy $0.4 \mu\text{m}$, orange line), pixel resolution particle tracking (red line) and subpixel resolution particle tracking (green line) are superimposed. Black dotted lines indicate the $\pm 8 \mu\text{m}$ around the instruction curve corresponding to our subpixel tracking resolution. Inset: Zoom on the displacement curves (displacement from the pixel resolution tracking has been removed for clarity). (b) Difference between the displacement determined by the subpixel particle tracking measurement and the instruction (blue line) and the profilometer measurements (orange line). The standard deviation of $x - x_{ref}$ is $\sigma = 2.5 \mu\text{m}$, and $3\sigma \approx 8 \mu\text{m}$. Black dotted lines indicate the $\pm 8 \mu\text{m}$ resolution. (c) Displacement curves corresponding to a series of instructions emulating a stick-slip cycle with inter-event sliding, consisting of 6 repetitions of the following sequence: $100 \mu\text{m}$ at $20 \mu\text{m/s}$, $100 \mu\text{m}$ at 10mm/s (quasi instantaneous step), 5 s stops. Colored lines as in (a). Inset: Zoom on one sequence (displacement from the pixel resolution tracking has been removed for clarity). (d) Temporal evolution of the cumulated inter-event slip without (top) and with (bottom) normalization. The non-normalized value is expected to be incremented by $100 \mu\text{m}$ after each inter-event period, and the normalized value is expected to be 0.5, since the same amount of slip ($100 \mu\text{m}$) occurs during a fast and a slow sliding event. The inter-event slip δ_{IE} is measured between 150 ms and 500 ms after the rapid event and 50 ms before the next rapid event (see Methods), which explains the difference between the expected (0.5) and measured values (0.495) for S .

- Figure 1: Loading contrast (LC) should be carefully distinguished, as it can be confused with the critical nucleation length noted as L_c in the literature. The definition of LC should be included in the figure caption of Figure 1.

Response: We agree with the reviewer that the notation is confusing. We have changed it for the notation C_σ and we have added the definition to the caption of Fig. 1:

Caption of Fig. 1: "The loading contrast, $C_\sigma = (\sigma_{yy}^{gran} - \langle \sigma_{yy}^{solid} \rangle) / \langle \sigma_{yy} \rangle$ (see main text), is indicated in each panel."

- Line 84: Is the entire fault slipping or just a portion of it?
 - Line 99: What is the size of the rupturing patch? Where does the slip occur along the interface?

Response: Here are the answers to the last two questions, as they are related. Each force drop corresponds to an entire rupture of the interface. We can check this based on the high-frequency strain measurements. In all the performed experiments, we have not observed partial ruptures of the interface. We clarify this point in the text, which now reads:

I.126: "The high-frequency strain measurements shown in Fig. 4 allow us to verify that, in our experiments, each force drop corresponds to a rupture that spans the entire interface, and not to arrested ruptures that may act as precursors to the system-spanning ruptures²⁷."

- Line 102: Please specify how slip measurements are taken.

Response: As mentioned in our reply to the reviewer's comment on Line 55, we have added more details about the particle tracking in the main text and in the Methods section, with an

additional Supplementary figure (new Fig. S1). We believe that these parts now provide the information needed to understand slip measurements.

- Figure 2d: Explain the normalized cumulative inter-event slip variation at early times.

Response: Cumulated inter-event slip is the amount of slip that occurred during inter-event periods divided by the total amount of slip experienced by the interface up to a time t . Due to the short duration of the inter-event periods in our experiments (typically less than 5 s for the shortest ones, cf Fig. 1d), the early times consists of the ratio of two comparable amount of slip, therefore very sensitive to the measurements uncertainty.

We clarified this point in the main text by adding the following:

1.148: “The measurements of $S^{solid}(t)$ and $S^{patch}(t)$ show large values at early time. For short inter-event periods, i.e. high stick-slip frequency, the quantity $S^{\{l\}}$ consists of the division of two comparable quantities, i.e. the slip over a few inter-event periods and the total slip, both of which are limited at early times. In this case, the uncertainty of the slip measurements becomes of considerable importance. Therefore, we consider the converged value of the cumulated inter-event slip, $S_f^{\{l\}}$, as the average over the last few events of an experiment, as a function of the loading contrast C_σ (Fig. 3a).”

- Figure 3: Clarify the color code in the legend.

Response: Color code is now added to the legend and caption. For clarity, we have also modified the symbols with consistency in all the figures. Circles are for empty-hole experiments, diamond for granular experiments, light blue for measurements above the solid-solid sections, dark blue above the empty hole and brown above the hole filled with cylinders.

Fig. 3 caption: “**Fig. 3: Slow slip measurements.** (a) Converged values of the normalized cumulated inter-event slip (top) S_f^{patch} for the empty-hole (blue circles) and the granular patch (brown diamonds) and (bottom) S_f^{solid} for the solid-solid sections of the interface, as a function of the loading contrast C_σ . (b) Differential normalized cumulated inter-event slip $S_f^{patch} - S_f^{solid}$ for empty-hole (dark blue circles) and granular (blue diamonds) experiments as a function of the loading contrast C_σ . Inset: mean frequency $\langle 1/\Delta T \rangle$ of the stick-slip cycle as a function of $S_f^{patch} - S_f^{solid}$.”

- Line 135: Can the granular patch serve as a trigger for rupture nucleation due to different frictional properties between the granular patch and the rest of the sliding interface?

Response: The frictional properties of the granular patch, both the intergranular friction and the mechanics of the granular pile, probably explain why the granular patch slides as soon as it is loaded. However, the fact that the frictional properties are different does not explain the mechanism by which the ruptures are nucleated within the neighboring solid-solid frictional interfaces. What is often modeled in numerical simulations is a velocity-strengthening patch in stable sliding embedded in a velocity-weakening matrix. The frictional properties are

different, but the mechanism is the emergence of a stress concentration due to the slip mismatch.

- Line 147: The sentence "the distribution is robust when... is not imposed by the interface" needs clarification.

Response: We have clarified this sentence, which now reads:

1.183: "The rupture nucleation location could be caused by a structural defect along the interface, such as a dip or bump. It is not the case as the nucleation occurs at the same locations, i.e. near the same patch corner, when one of the blocks is flipped, thus inverting the x -positions."

- Figure 4: Trace the rupture front in dashed lines on Figure 4A and add rupture velocity to the figure.

Response: Lines are added. Upon the suggestion of referee #3, we have also added another example of strain measurements for an experiment with a higher loading contrast. The revised Fig. 4a-b is:

Fig. 4: Modification of the rupture nucleation location with an increasing loading contrast. (a-b) Shear strain variation $\varepsilon_{xy}(x, t) - \varepsilon_{xy}^0(x)$ due to the propagation of a rapid rupture, measured at 4 MHz for granular experiments at (a) $C_\sigma = -0.71$ and (b) $C_\sigma = 1.45$, at the 10 measurement locations (blue lines for strain gages above the solid-solid sections, brown line above the granular patch). Red stars indicate the point at which the rupture initiates. The location of the corresponding strain gauge is determined as the nucleation location. Black dotted lines indicate the detection of the initiation of the strain variation for each strain gauge signal and red lines indicate an approximate rupture speed.

- The definition of sublinear evolution is unclear and arises much later in the text.

Response: We have detailed further the definition of sublinear strain evolution, which now appears when the corresponding new Fig. 5 is cited.

1.211: “Sublinear evolution refers to the shear strain signal, $\varepsilon_{xy}(t)$, saturating rather than increasing linearly over an inter-event period (except at very short times after a rapid event, when a rapid increase in shear strain corresponds to interfacial logarithmic aging).”

- Line 150: The statement about the nucleation point shifting as LC increases appears speculative based on the data presented in Figure 4b. There isn't a noticeable change in the nucleation location on the histograms as a function of LC.

Response: We thank the reviewer for their comments, which allowed us to significantly improve Fig. 4 and strengthen its message. Fig. 4 is now dedicated to the high-frequency strain measurements and the determination of the rupture nucleation location. A new figure, Fig. 5, is devoted to the low-frequency measurements and the determination of the length of the slipping patch.

Fig. 4d shows the averaged location of the rupture nucleation as a function of the loading contrast. We calculate the average value of each side of the histograms, and take the absolute value of this quantity as the averaged nucleation location. The nucleation distance $\langle d_{nuc} \rangle$ is the mean value of the distance of the averaged nucleation location from the center of the interface (see schematic of Fig. 4d). We believe that this plot clearly shows the shift in the nucleation point with the loading contrast.

The new Fig. 4d is:

New Fig. 4d: (d) Averaged values of the nucleation distance $\langle d_{nuc} \rangle$ normalized by the central hole half-length, $\ell_{hole}/2$ where $\ell_{hole} = 30$ mm, for each range of C_σ used for the histograms in (c). The nucleation distance $\langle d_{nuc} \rangle$ is the mean value of the distance of the averaged nucleation location from the center of the interface, measured from the data shown in the histograms of (c). Circle corresponds to empty-hole experiments and diamonds to granular experiments. Error bars corresponds to the standard deviations of the distributions in (c).

We have modified the main text and the figure caption to comment this new plot. It now reads:

1.188: “...the nucleation point is shifted away from the granular patch towards the outer corners of the blocks. To demonstrate this effect, we calculate the mean values of the location of rupture nucleation over the different ranges of C_σ and then define the mean nucleation distance $\langle d_{nuc} \rangle$ as the distance from the interface center to the mean nucleation location

(taking the absolute value of the nucleation x-coordinate) (Fig. 4d). Indeed, we find that the nucleation distance increases with C_σ ."

- Line 154: Based on previous comments, there is no clear evidence that the length of the central slipping zone increases with LC. Is this visible in the slip measurements? Do you have the resolution to observe it?

Response: We hope that based on our response to the previous comments, the reviewer is now convinced that the slipping patch expands with an increasing loading contrast. The experiments presented in this manuscript have been performed with 4 tracking locations along the solid-solid sections of the interface for the interfacial slip measurements. We indeed observe a spatial dependence on the measurement of the inter-event cumulated slip $S_f^{\{l\}}$ as shown on the plots below. Figs 2 and 3 in the manuscript show the average of the 4 solid measurements for what is called the solid-solid measurements.

Figure for comment 3.4: Converged values of the normalized cumulated inter-event slip at each tracking location: 4 locations along the solid-solid interfaces, 1 location in the middle of the granular patch.

We have also added data in Figs. 4 and 5, where we compute the averaged values of the nucleation location (Fig. 4d) and of the slipping patch length (Figs. 5d and 6a) per range of loading contrast to extract, which are both obtained from 2 different type of measurements (nucleation location and sublinear evolution of the quasi-static shear strain). In Fig. 5d, we plot one against the other and find that these two quantities agree, confirming the validity of our detection method of the central slipping patch.

New Fig. 5d: (d) Comparison of the slipping patch length, $\langle \ell_{slip} \rangle$, corresponding to the averaged length of the slipping patch obtained from the data shown in (c), normalized by the hole length, $\ell_{hole} = 30$ mm, and the nucleation distance from the center $\langle d_{nuc} \rangle$, normalized by the hole half-

length $\ell_{hole}/2$. Circle corresponds to empty-hole experiments and diamonds to granular experiments.

- Line 156: The paper discusses a crack governed by the Griffith criterion, but this is not demonstrated in the paper. Did you calculate GC?

Response: The paper discusses the role of the central slipping patch on the nucleation of rapid ruptures. We argue that the slipping patch acts as a seed crack. For a crack governed by linear elastic fracture mechanics, Griffith criterion relates the length of the seed crack to the loading that allows it to propagate. In our study, the longer the patch, the higher the frequency of nucleation, and therefore the lower the stress at which nucleation occurs compared to the residual stress of the interface, as expressed by the Griffith criterion.

What's the correct way to verify the validity of the Griffith criterion or, more generally, of a crack initiation criterion? It would consist in determining the static stress intensity factor, K_{II} , at the tip of our seed crack, i.e. the slipping patch. In our geometry this problem has been solved by Muskhelishvili in 1953 and the calculation can be found in the book of Leblond, "Mécanique de la rupture fragile et ductile" (in French, but it might be possible to find another reference in English).

[figure redacted]

Figure for comment on line 156: Configuration of a crack in an infinite medium, from the book of Leblond (Figure 3.3)

For a mode II crack with a non-uniform loading $q(t)$ where t is the coordinate along the crack, the SIF in $x = \pm a$ writes:

$$K_{II} = \frac{1}{\sqrt{\pi a}} \int_{-a}^{+a} q(t) \sqrt{\frac{a+t}{a-t}} dt$$

As we consider a frictional crack, the residual stresses should be subtracted from the loading. The integral in the K_{II} expression has to be considered on both the granular patch, where the residual stresses are $\tau_g(x)$, and on the neighboring solid-solid areas where the residual stresses are $\tau_r(x)$. We are not able to perform this measurement because of the lack of spatial resolution of the strain/stress measurements: τ_g , τ_r and q have a strong spatial dependence in our system. Therefore, the best we can do is to compare the trend of the relation between ℓ_{slip} and the shear loading with the initiation criterion for a pure crack in an infinite system under homogeneous loading.

We clarified the statement about the Griffith criterion in the revised manuscript and discuss the limitations of a quantitative verification:

1. 245: “Our interpretation is that the central slipping zone acts as a nucleation center, or a seed crack, along the interface. In Mode II, a seed crack is an area where the shear stress has been released by interfacial slip. The initiation criterion of a crack, the Griffith criterion, is $G_S = G_c$, where G_S is the static energy release rate and G_c is the fracture energy of the material. For a seed crack embedded in an infinite solid under a homogeneous loading, the Griffith criterion leads to an expression of the critical stress τ_c for which a crack of a given length l is destabilized, $\tau_c = \sqrt{G_c E / (1 - \nu^2) l}$. Because we are considering a frictional crack, where stresses are relieved to the residual stress value rather than zero³⁵, with spatially dependent shear loading due to compositional heterogeneity and a normal stress gradient induced by the loading contrast (Fig. 1b), a quantitative estimation of G_S is challenging and cannot be performed in our experiment. However, we can verify that the response of the system to the presence of the slipping patch follows the same trend as in the presence of a pure seed crack.

In our experiment, the seed crack length corresponds to the size of the central slipping zone. The higher the loading contrast, the longer the central slipping zone (Fig. 6a) and the higher the stick-slip frequency (Fig. 1d). This is qualitatively consistent with the Griffith criterion for an infinite and homogeneous system: the longer the seed crack (l_{slip}), the lower the critical stress at which ruptures initiate compared to the minimum stress reached after the interface weakening – the residual stress – resulting in the initiation of rapid slip events after a shorter loading time and thus to an increased stick-slip frequency.”

About G_c , in this experiment, we cannot measure it confidently along the interface, because to perform this measurement is not an easy task. First of all, G_c cannot be determined based on the data of a single strain gauge, averaging is necessary (see Svetlizky and Fineberg, Nature 2014, ref 35). Second, G_c strongly depends on the local normal stress (see Bayart et al, Nature Physics 2016 for experimental data, ref 27). As the normal stress distribution along the interface is strongly heterogeneous in our experiments, a local determination of G_c , i.e. at each strain gauge location, is necessary. For these two reasons, we are not able to provide G_c measurements.

However, as far as crack nucleation is concerned, the relevant G_c value is the one at the tip of the seed crack. In our experiments, we have access to the local value of the normal stress at the crack nucleation site for each event. Since G_c is proportional to normal stress, this measurement is a proxy for the G_c values at the rupture nucleation site. Thanks to the reviewer’s question, we have performed this measurement and the data are now presented in the revised manuscript, in Fig. 6b. We found that, as the loading contrast C_σ increases, the normal stress at the nucleation location decreases. As the fracture energy is proportional to the normal stress, this reduction leads to a reduction of the fracture energy at the nucleation location. This effect plays the same role as the slipping patch elongation in destabilizing the seed crack at lower shear loading.

Estimating the value of G_c under these normal stress conditions is uncertain without measuring it, but we can use the data presented in Fig. 2b in Bayart et al. Nature Physics 2016 (ref 27), even though this type of data is very sensitive to the details of the system. Normal stress values spread in average between 2 and 3 MPa, which corresponds to G_c values

between 0.5 and 0.75 J/m² in Bayart et al, 2016. Because the system is different, we have chosen not to present these numbers in the revised version of the manuscript, but only to mention the additional effect of the fracture energy variation on the destabilization of the seed crack.

The new Fig. 6 is:

Fig. 6: Slipping patch length and local normal stress leading to rupture destabilization. (a) Evolution of the slipping patch length, $\langle \ell_{slip} \rangle$, corresponding to the averaged length of the slipping patch obtained from the data in Fig. 5c, normalized by the hole length, $\ell_{hole} = 30$ mm, as a function of the loading contrast for the empty-hole experiments (circle) and the granular experiments (diamonds). Error bars corresponds to the standard deviations of the distributions in Fig. 5c. Measurements for each individual event appear in gray, circle or diamond. (b) Averaged values of $\sigma_{yy}^0(x_{nuc})$, the normal stress measured at the nucleation locations x_{nuc} of each rapid event, for each range of C_σ used for the histograms in Fig. 4c and Fig.5c for the empty-hole experiments (blue circle) and the granular experiments (blue diamonds). Measurements of $\sigma_{yy}^0(x_{nuc})$ for each individual event appear in gray, circle or diamond.

We discuss the results of Fig. 6b in the main text:

1.264: “The local value of the fracture energy G_c also affects the stress τ_c at which a rupture initiates. The fracture energy G_c strongly depends on the local normal load²⁷ and, because of the heterogeneous normal stress distribution along the interface in our experiments, we cannot confidently extract the G_c value from our strain measurements. However, the Griffith criterion must be fulfilled locally at the rupture tip for a rupture to be destabilized. Although G_c cannot be measured, we do measure the normal stress at the nucleation point prior to each event. Since G_c is linearly dependent on normal stress^{27,36}, this measurement gives an indication of the variation trend of the fracture energy at the rupture tip with the loading contrast. Fig. 6b shows the value of σ_{yy} at the nucleation point before the rapid events, $\langle \sigma_{yy}^0(x_{nuc}) \rangle$ where x_{nuc} is the nucleation location, averaged over all the events of the empty-hole experiments (circle) and granular experiments (diamonds) for the four ranges of loading contrast, C_σ , shown in Figs. 4c and 5c. This plot demonstrates a decreasing trend of $\langle \sigma_{yy}^0(x_{nuc}) \rangle$ with the loading contrast, C_σ . Therefore, the variation of the fracture energy G_c at the rupture nucleation location may play a role in reducing the critical stress for the seed crack destabilization, an effect that is additive to that of the elongation of the slipping patch. The two parameters ℓ_{slip} and G_c cannot be controlled independently of each other in our experiment, due to the presence of the compositional heterogeneity, i.e. the granular patch.”

- Line 156: Please define "seed crack" as it is not trivial.

Response: We added the following sentence to the main text:

1.197: “We interpret this result as the fact that a slowly slipping granular section induces the formation of a slipping seed crack, destabilized following a **fracture initiation** criterion. **A seed crack is a pre-existing crack, i.e. a line along which stresses have been released, that destabilizes once the shear stress reaches a threshold value, given by the Griffith criterion in elastic fracture mechanics.**”

- Line 162: "Low strain measurements allow us to overcome this lack of resolution." Please explain further and provide data.

Response: This sentence refers to the measurements that are presented in the next paragraph (lines 207 to 243). We clarified the sentence, which now reads:

1.206: “However, low-frequency strain measurements allow us to overcome this lack of resolution, **as we will show below.**”

- Line 173: To compare with natural seismicity, it would be interesting to compute and present the fault coupling.

Response: We agree that it is interesting to extract seismic or geodesic measurements type from experimental data. However, the scope of this paper is to demonstrate the interaction mechanisms between a slipping patch and the neighboring locked frictional interfaces. We feel that it is out of the scope of this specific study and might be the subject of a future work.

- Line 173: The statement about “the larger the central... the longer the slipping zone” seems speculative based on the data presented in the paper. I'm not sure the authors have the slip resolution to make that claim.

Response: Regarding the resolution of slip measurements, please refer to our response to the previous comment on Line 55. However, this statement refers specifically to the interpretation of the events count presented in the new Fig. 5c (originally Fig. 4d), which are not based on slip measurements, but on the evolution of the quasi-static shear strain signal at each strain gauge location during the inter-event periods (linear if the interface is locked at this specific location, sub-linear if the interface is sliding). We have added data in Fig. 5d and Fig 6a showing the evolution of the slip length $\langle \ell_{slip} \rangle$ with the loading contrast, where $\langle \ell_{slip} \rangle$ is obtained from averaging each side of the event counts in Fig. 5c.

Fig. 5d is:

New Fig. 5d: (d) Comparison of the slipping patch length, $\langle \ell_{slip} \rangle$, corresponding to the averaged length of the slipping patch obtained from the data shown in (c), normalized by the hole length, $\ell_{hole} = 30$ mm, and the nucleation distance from the center $\langle d_{nuc} \rangle$, normalized by the hole half-length $\ell_{hole}/2$. Circle corresponds to empty-hole experiments and diamonds to granular experiments.

Fig. 6a is:

New Fig. 6a: (a) Evolution of the slipping patch length, $\langle \ell_{slip} \rangle$, corresponding to the averaged length of the slipping patch obtained from the data in Fig. 5c, normalized by the hole length, $\ell_{hole} = 30$ mm, as a function of the loading contrast for the empty-hole experiments (circle) and the granular experiments (diamonds). Error bars corresponds to the standard deviations of the distributions in Fig. 5c. Measurements for each individual event appear in gray, circle or diamond.

We have revised the text to comment these figures:

1.224: “The spatial extent of the distribution informs us about the most probable length of the slipping patch for a given range of loading contrast. The larger the central width of the distribution, the longer the slipping zone. The slipping patch length, $\langle \ell_{slip} \rangle$, is defined as the averaged spatial extent of the region where the sublinear shear strain evolution is detected for each loading phase, i.e. each inter-event period, per range of C_σ . We first check the consistency of the two previous measurements: rupture nucleation location from high-frequency strain measurements and slipping patch extension from quasi-static loading curves (Fig. 5d). The two lengths $\langle d_{nuc} \rangle$ and $\langle \ell_{slip} \rangle$ are indeed comparable, especially when the slipping patch becomes longer than the initial compositional heterogeneity, i.e., the hole length, $\ell_{hole} = 30$ mm. Furthermore, as expected from the correlation between $\langle d_{nuc} \rangle$ and $\langle \ell_{slip} \rangle$, Fig. 6a shows that the slipping patch length increases with C_σ . This plot indicates that the larger the normal load carried by the granular patch, the longer the slipping patch, that extends beyond the initial compositional heterogeneity and thus into the solid-solid sections of the interface, which otherwise may rupture seismically under different loading conditions.”

- Line 180: Authors suggest that granular material induces slow slip events. Another hypothesis is that slow slip events are introduced by stress heterogeneity induced by the granular material, not the material itself. Is this plausible?

Response: It is an interesting hypothesis. Our empty-hole experiments do have a strong stress heterogeneity, with a zone of low normal stress, and in this case, we do not observe any slow

slip along the interface (see Fig. 3). Therefore, a “negative” defect in the normal stress distribution does not induce slow sliding. The effect of a “positive” defect, i.e. a patch with a stronger normal stress than the averaged stress, has not been checked so far. However, the corresponding author of this manuscript has previously worked with strong gradient of normal stress applied on the interface (see Bayart et al, Nature Physics 2016 for instance) and slow slip has never been observed, even at the locations where the normal stress was maximal. A dedicated study would be needed, using pistons or machining the block surfaces. However, this is not currently possible with our setup.

- Line 187: The definition of the seed crack should precede this point.

Response: The seed crack is now defined Line 197 (see our response to reviewer’s comment on Line 156).

- Line 190: Nucleation, as defined in finite fracture mechanics, is not only governed by energy but also by stress. How is this accounted for here?

Response: Fracture mechanics is an interplay between the elastic energy, expressed by the energy release rate G , and the energy dissipated at the crack tip, the fracture energy G_c . The energy release rate is calculated via the stresses applied to the solid. Nucleation criterion can be expressed in terms of energy, which is the Griffith criterion and writes $G = G_c$, or in term of stresses, which writes $K = K_c$, where K is the stress intensity factor and K_c is the toughness of the material. These two criteria are formally equivalent.

- Line 216: The authors mention that the granular material dilates. Please show the normal displacement measurements; otherwise, this is speculative.

Response: The Reynolds dilatancy is a well-known phenomenon for dense granular materials (Reynolds, 1885, On the dilatancy of media composed of rigid particles in contact, Phil. Mag., 20, 469-481). In our experiment, the density of the granular material can be estimated to be larger than 0.6 (volume of the cylinders divided by the volume of the hole). In this range, it is expected to observe shear-induced dilatancy. In addition, our static measurements of the normal stress before each rapid event (gray lines on Fig. 1b) allows us to follow the evolution of the normal stress from one rapid event to another. We observe that the normal stress above the granular patch increases over the experiment. This is an indication of the dilatant behavior of the granular material.

About measuring the vertical displacement above the granular patch, we do not have the resolution. Let’s consider an interface loaded with a normal stress $\sigma_{yy} = 2$ MPa (normal force $F_N = 3 \cdot 10^3$ N and area $S = 15$ cm x 1 cm). Let’s now consider that normal stress would increase by 50% between events due to dilatancy. The expected displacement of a particle located on the hole boundary would be $\delta = \frac{\sigma_{yy} h}{4E} \approx 2$ μ m, with $h = 1$ cm, the hole’s height and $E = 3$ GPa the Young’s modulus. This quantity, overestimated, is below the resolution of the tracking method.

The discussion on granular dilatancy has been clarified:

l. 301: “The second **mechanism** is the shear-induced dilatancy of the granular patch. A dense granular material dilates, i.e. expands, when sheared⁴³, which can locally release the normal stress on the surrounding areas, leading to a transition to stable sliding³⁵. **In our experiments, we observe a local reduction in normal stress near the edges of the patch as the normal load imposed on the patch increases, although it never disappears (Figs. 1c). The local reduction in normal stress could explain the transition from stick-slip to steady sliding^{44,45}.** This last mechanism emphasizes the importance of considering the mechanical behavior of heterogeneities in models.”

The gray lines are now defined in the caption of Fig. 1b : “Gray lines correspond to measurements before each rapid event, while colored lines are averaged over all events of an experiment.”

- Line 211: The authors propose two mechanisms to explain creep: 1) low rate of shear loading and 2) dilatancy. Can the frictional properties of the granular rods alone explain creep?

Response: In this part of the manuscript, we are discussing why the solid-solid sections neighboring the granular patch are slowly sliding during the inter-event periods. Therefore, we do not directly mention the frictional properties of the cylinders. In fact, the granular patch is always sliding, and this is probably due to its frictional properties. But what is intriguing is that the solid-solid interfaces can be either locked or in stable sliding, without modification of the composition of the interface.

The discussion is clarified in the revised manuscript:

l. 295: “What causes microcontacts to **slide** instead of breaking dynamically? There may be two possible mechanisms involved. The first **mechanism is the creep of the microcontacts loaded by the slipping granular patch at a very low rate, of the order of few $\mu\text{m/s}$ (Fig. 2d). The creep regime has been observed for several experimental systems^{40,41}.** In fact, frictional contacts age with time, strengthening the interface, **while low-rate shear loading induces plastic flow of the contacts⁴².** The comparable timescales between these two processes may lead to creep rather than dynamic failure of the microcontacts.”

- Lines 222-223: The statement about "the boundary of the uncoupled zone... extends through slow slip or contacts creep of the surrounding areas" is not fully demonstrated in the manuscript as per previous comments.

Response: We hope that our answers to previous comments have now convinced the reviewer that the solid-solid interfaces neighboring the granular patch are sliding during the inter-event periods.

- Line 230: The Griffith criterion has not been proved in the manuscript, and G_c was not computed. This statement seems speculative and requires rephrasing.

Response: We agree with our reviewer that the validity of the Griffith criterion has not been formally proved. Please refer to our response to reviewer’s comment on Line 156 for further

details. In addition to the modification from 1.245 to 280 (mentioned in the response to comment Line 156), we have modified the conclusion to better convey the study's message:

1. 329: "In summary, we have shown that a slowly slipping area within a frictional interface acts as a seed crack, which induces an early destabilization of the entire interface by lowering the stress level at which the crack propagates, following the same trend as a Griffith criterion derived from fracture mechanics for pure cracks."

- The implications of the results for natural cases should be developed further.

Response: We develop further on the insights of this study for geosciences.

1. 317: "Our experiments provide a way to explore the complex fault dynamics by controlling the ingredients responsible for complexity. For instance, our observations correspond to a scenario of tremors, where slow slip induces repeated microearthquakes⁵. In our study, the event amplitude, i.e. the force drop, is controlled by the length of the slipping patch, rather than the amount of slow slip. Experimental acoustic measurements would allow to relate the seismic wave content to the source properties, which are inaccessible in natural faults. This study also raises the question of which fault zones should be monitored. If the slow slip zone can extend along a fault, as our results suggest, it is important to monitor the zone's evolution during interseismic phases. Finally, a quantitative estimate of a dynamic criterion for rupture propagation would provide information on the destabilization length of the slipping patch, for known fault loading conditions. Modeling the extension of a slow-slip zone may provide insightful results for this purpose."

Reviewer #3 (Remarks to the Author):

Review of NCOMMS-24-00631:

General Assessment:

The manuscript entitled “Experimental evidence of seismic ruptures initiated by aseismic slip” by Y. Faure and E. Bayart presents an experimental study on the influence of spatially-distributed stable sliding on the initiation of dynamic shear ruptures. The authors use experiments of direct shear performed on a model 1D (thin) fault system located between two PMMA plates. The main novelty lies in the presence of a granular media at the center of the fault, consisting of nylon cylinders. This permits to initiate a slowly-creeping region at the fault center and to investigate its impact on the nucleation of laboratory earthquakes.

The key results presented in this manuscript are:

- The stable sliding region expands quasi-statically along the solid/solid (PMMA/PMMA) interface before the onset of the instability.
- The propensity of the sliding region to penetrate the solid/solid interface strongly depends on the heterogeneities of normal stress along the fault.
- In turns, it modifies the frequency of the seismic (stick-slip) events, promoting low-magnitude tremors over more energetic ruptures.

The research conducted in this work is of notable quality. While the text is generally well-written and structured, there are areas requiring clarification and improvement to enhance readability and precision. Moreover, it is not clear to me whether the impact of stable sliding on the onset of dynamic ruptures is controlled by the fault frictional heterogeneity or rather by the gradient of normal stress generated along the fault.

Response: We sincerely appreciate the valuable comments provided by the reviewer and hope that our responses adequately address their concerns. Regarding the impact of the normal stress gradient, please refer to our answer to reviewer’s comment 5.1.

I hope to hear the authors’ opinion on the following:

- Abstract – The abstract is overall well-written and engaging, but I have some comments.
 - Comment 0.1: The use of "heterogeneity" lacks specificity. The authors should delineate the types of heterogeneities considered (stress, composition, slip distribution, etc.).
 - Comment 0.2: The author writes at line 18 that the heterogeneity “[reduces] fault shear resistance”. I guess that they refer to the decrease in macroscopic friction F_s/F_n observed in their experiments, rather than a microscopic change in friction/shear strength. This concept of “macroscopic stability” may need to be clarified.

Response: We agree with reviewer comments 0.1 and 0.2 and suggest a more specific version of this sentence. However, we are rather limited in the details we can provide due to the 200-word limit for the abstract.

It now reads:

1.16: “By measuring the response of the fault to shear, we show that the role of the **compositional** heterogeneity is to serve as a seed crack for rapid ruptures, reducing **the macroscopic** fault shear resistance.”

■ Comment 0.3: The “fracture concepts” used in the paper are rather limited. If I am right, they consist in the sole Griffith-like criterion in the section entitled “Slowly slipping area acting as a nucleation seed”. I am not sure that they permit to “isolate the very origin of earthquake nucleation and slip dynamics in seismic fault”, in contrast with previous works performed by the corresponding author.

■ Comment 0.4: The last sentence of the abstract seems to another overstatement regarding the content of the paper. The study conducted by the authors is of high quality and does not need overselling. I am nonetheless totally in line with the tone used in the “Discussion” section of the paper.

We have modified the last sentences of the abstract to convey a more realistic message:

1.21: “Our findings underscore the utility of fracture tools in elucidating the dynamics of complex faults. The identification of the interaction mechanisms between slowly-slipping and locked regions paves the way for the construction of novel models, thereby improving fault monitoring capabilities and seismic hazard mitigation strategies. “

• Laboratory-fault experiment

■ Comment 1.1: (line 75) How is calculated the 8 microns precision for the DIC measurements? The pixel size is $\sim 150\text{mm}/1280\text{pix} = 120$ microns. 8 microns is smaller than the error traditionally-associated with DIC measurements (Bornert et al., 2009, doi:10.1007/s11340-008-9204-7).

Response: We have developed a particle tracking algorithm with a sub-pixel detection of the particle position. The sub-pixel resolution is achieved by using a 3-point estimator with a Gaussian peak fit. This is known to give a resolution better than 0.1 pixel, up to 0.01 pixel in the best case (Gaussian intensity distribution of the target, see section 5.4.5 in ref 25 (Particle image velocimetry: a practical guide / Markus Raffel et al (2007))). As the reviewer correctly estimated, our resolution is $1\text{ mm}/8\text{ pix} = 125\text{ }\mu\text{m}/\text{pix}$. We have calibrated the spatial resolution of the tracking by performing a translation of the unloaded lower block at a constant speed ($10\text{ }\mu\text{m}/\text{s}$). The $8\text{ }\mu\text{m}$ resolution is obtained as follows: the standard deviation of the difference between the particle displacement and the applied displacement is $\sigma = 2.5\text{ }\mu\text{m}$. We considered that the resolution is 3σ , i.e. the range that includes 96% of the measured values. These data are now presented in a new Fig. S1 (original Fig. S1 becomes Fig. S2) and discussed in “Particle tracking methods” section of the Methods.

Following the comments of reviewers #2 and #3, we have added extensive information and data on the particle method and resolution in the Methods section (see the "Particle tracking method" and "Slip measurements" sections of the Methods and new Fig. S1). We also provide more details in the main text:

1.361: a section is added to the Methods, entitled “Particle tracking method.” and referring to a new supplementary figure, new Fig. S1a, b.

More details have been added to the “Slip measurements” section to describe our mimicked stick-slip experiment (new Fig. S1c, d):

1.409: “To ensure that the spatial resolution of the slip measurements, emanating from the subpixel particle tracking, is sufficient to provide valid cumulated inter-event slip

measurements, we have mimicked a stick-slip cycle with inter-event sliding, consisting of 6 repetitions of the following sequence: 100 μm at 20 $\mu\text{m/s}$, 100 μm at 10 mm/s (quasi-instantaneous step), 5 s stops (Fig. S1c). As expected, the cumulated inter-event slip $\delta_{IE}(t)$ increases by increments of 100 μm after each inter-event period and the normalized value $S^{\{t\}} = 0.5$, as the same amount of slip is experienced during the inter-event periods and the rapid events (Fig. 1d). Note that the slight difference between the measured value ($S = 0.495$) and the expected value ($S = 0.5$) is due to the fact the inter-event slip δ_{IE} is measured between 150 to 500 ms (τ^+) after the rapid event and 50 ms before the next rapid event.”

We have also added more details within the main text:

1.84: “Interfacial slip measurements are performed using cylinder position data obtained with a sub-pixel resolution particle tracking algorithm. Particle tracking involves the correlation of images taken at 100 frames per second during an experiment with a template image. Sub-pixel resolution is achieved using three-point estimator with a Gaussian peak fit²⁵, resulting in an 8 μm resolution in displacement (Methods). Tracking is performed on the cylinders embedded in the eye-shaped hole boundaries and on similar patterns drawn on the block faces above the interface solid-solid sections, allowing measurements under the same conditions as for the granular section. The interfacial slip is measured at 4 locations along the solid-solid sections and 3 locations above the hole, by subtracting the measured position of cylinders or patterns located on the top and bottom blocks, at the same location x . Corrections for the rotation of the system during the shear loading are made (Methods).”

The new figure in the supplementary (new Fig. S1) reads:

Fig. S1: Determination of the spatial resolution of the tracking method. (a) Displacement curves corresponding to a translation of the lower block (without shear resistance) at $10 \mu\text{m/s}$ imposed by the motorized stage. Instruction (blue line), block's displacement measured with an optical confocal profilometer (accuracy $0.4 \mu\text{m}$, orange line), pixel resolution particle tracking (red line) and subpixel resolution particle tracking (green line) are superimposed. Black dotted lines indicate the $\pm 8 \mu\text{m}$ around the instruction curve corresponding to our subpixel tracking resolution. Inset: Zoom on the displacement curves (displacement from the pixel resolution tracking has been removed for clarity). (b) Difference between the displacement determined by the subpixel particle tracking measurement and the instruction (blue line) and the profilometer measurements (orange line). The standard deviation of $x - x_{ref}$ is $\sigma = 2.5 \mu\text{m}$, and $3\sigma \approx 8 \mu\text{m}$. Black dotted lines indicate the $\pm 8 \mu\text{m}$ resolution. (c) Displacement curves corresponding to a series of instructions emulating a stick-slip cycle with inter-event sliding, consisting of 6 repetitions of the following sequence: $100 \mu\text{m}$ at $20 \mu\text{m/s}$, $100 \mu\text{m}$ at 10mm/s (quasi instantaneous step), 5s stops. Colored lines as in (a). Inset: Zoom on one sequence (displacement from the pixel resolution tracking has been removed for clarity). (d) Temporal evolution of the cumulated inter-event slip without (top) and with (bottom) normalization. The non-normalized value is expected to be incremented by $100 \mu\text{m}$ after each inter-event period, and the normalized value is expected to be 0.5 , since the same amount of slip ($100 \mu\text{m}$) occurs during a fast and a slow sliding event. The inter-event slip δ_{IE} is measured between 150ms and 500ms after the rapid event and 50ms before the next rapid event (see Methods), which explains the difference between the expected (0.5) and measured values (0.495) for S .

■ Comment 1.2: (Figure 1) As I am familiar with the work done by the corresponding author, I guess that the diamond shaped points correspond to experiments without center hole. This could be more clear.

Response: The initial data on Fig. 1d were the mean period for empty-hole experiments under a varying normal force (circles). To make our statement clearer, we have added data for classical friction experiments with flat homogeneous solids under varying normal force (squares) and added a legend in the figure.

New Fig. 1d: (d) Mean stick-slip period as a function of the average normal stress carried by the solid-solid sections $\langle \sigma_{yy}^{solid} \rangle$ for experiments with flat solids (black squares) and empty-hole experiments (circles) performed under varying normal force, $750 \text{N} < F_N < 3000 \text{N}$, and granular experiments

(diamonds) performed at $F_N \sim 3000$ N with a varying loading contrast C_σ . The colorbar codes for C_σ for granular experiments.

These data are described in the revised manuscript:

I.111: “In a classical friction experiment where two flat solid blocks are in contact along the entire interface, i.e. without a hole or a granular patch, a reduced normal load leads to smaller shear force drops and therefore to an increased stick-slip frequency. To verify that the observed frequency increase for an increasing loading contrast C_σ is not due to a decrease in the normal loading of the solid-solid sections of the interface, $\langle \sigma_{yy}^{solid} \rangle$, we compare the averaged stick-slip period $\langle \Delta T \rangle$ of no-hole experiments, i.e. classical solid-solid friction experiments with flat homogeneous solids in contact, under varying normal force, $750 \text{ N} < F_N < 3000 \text{ N}$, with our reference empty-hole experiments under the same range of normal force, and with granular experiments under $F_N = 3000 \text{ N}$ with a varying density, hence varying $\langle \sigma_{yy}^{solid} \rangle$ and C_σ (Fig. 1d). First, the stick-slip periods of no-hole and empty-hole experiments are comparable for the same $\langle \sigma_{yy}^{solid} \rangle$, demonstrating that the presence of the hole has no effect on the stick-slip cycle under given loading conditions. Slight differences between the two datasets are due to expected variations in static friction between systems²⁶. Second, the period shortening in the presence of the granular section is greater than that observed for a corresponding normal load reduction in no-hole and empty-hole experiments, demonstrating an effect of the compositional heterogeneity on the stick-slip cycle beyond unloading.”

■ Comment 1.3: The authors investigate here the inter-event duration. It may be interesting to look at the scalar moment release M , in order to evaluate the potency of the seismic ruptures. This can be done thanks to the displacement measurements made in this study.

Response: We can indeed measure the moment M for both inter-event periods and seismic ruptures, that we show below. However, the scope of this paper is to demonstrate the interaction mechanisms between a slipping patch and the neighboring locked frictional interfaces. We agree that it is interesting to extract seismic measurements type from experimental data. However, we feel that it is out of the scope of this specific study and that a subsequent study should be devoted to this subject.

Figure for comment 1.3: Averaged scalar moment release for each experiment as a function of the loading contrast C_σ (blue dots) for the inter-event periods (left) and the rapid slip events (right). Diamonds are a representation were data are binned by range of loading contrast.

- Slow slip and stick-slip cycle

■ Comment 2.1: As in Figure 1, the distinction between solid/solid experiments with (called reference solid/solid experiments) and without (called solid/solid experiments) hole is not clear. For example, I am not sure which kind of experiments is mentioned at lines 92-93 and 97-98.

Response: We agree with the reviewer that our definition of experiments is confusing. We now refer to:

- reference empty-hole experiments
- granular experiments with cylinders filling the hole
- classical friction experiment with flat solids, without hole.

We have modified the text following these new definitions.

■ Comment 2.2: In Figure 2c, I don't understand what are the dark blue and light blue lines. Is S^{gran} of the reference solid/solid experiment is defined from the slip displacement measured on the contour of the hole?

Response: Indeed, in both the empty-hole experiment and the granular experiment, we measure the cumulated slow slip separately for the solid-solid interfaces and for the hole (empty or filled). We have modified the variables names and defined a new consistent color and symbol code. Regarding variables, "patch" is added for the measurements taken over the hole, empty or filled: $\sigma_{yy}^{\text{patch}}$, $\delta_{\text{tot}}^{\text{patch}}(t)$, $\delta_{IE}^{\text{patch}}(t)$, $S^{\text{patch}}(t)$, S_f^{patch} , for respectively the normal stress, the total slip, the cumulated inter-event slip, the normalized cumulated inter-event slip and the converged value of the normalized cumulated inter-event slip. Whether measurements refer to the empty hole of the granular patch is now indicated in the legend and/or captions of the figures with the following color and symbol code: circles correspond to empty-hole experiments and diamond to granular experiments, light blue is for measurements

taken along the solid-solid sections of the interface, dark blue above the empty hole and brown above the hole filled with cylinders.

The quantities are defined in the revised manuscript:

I.133: “Interfacial slip measurements are used to determine the total slip of the solid-solid section of the interface, $\delta_{tot}^{solid}(t)$, and of the empty-hole or granular sections, $\delta_{tot}^{patch}(t)$, and the cumulated inter-event slip $\delta_{IE}^{solid}(t)$ and $\delta_{IE}^{patch}(t)$, defined as the slip experienced by the interface *excluding* slip occurring during rapid events (Fig. 2a,b). These quantities are averaged over the measurement locations for solid-solid sections and empty-hole or granular sections. We define the normalized cumulated inter-event slip $S^{\{l\}}(t)$ as:

$$S^{\{l\}}(t) = \frac{\delta_{IE}^{\{l\}}(t)}{\delta_{tot}^{\{l\}}(t)},$$

where $\{l\}$ refers to solid-solid or patch, the latter being for an empty hole or a hole filled with cylinders.”

The figure captions have been modified:

Fig. 2 caption: “**Fig. 2: Measurements of the interfacial slip.** (a-b) Temporal evolution of the total interfacial slip of the solid-solid sections $\delta_{tot}^{solid}(t)$ (light blue line) and of the patch $\delta_{tot}^{patch}(t)$, when empty (dark blue line in (a)) or filled with cylinders (brown line in (b)), obtained from the particle tracking method for (a) an empty-hole experiment and (b) a granular experiment ($C_\sigma = 1.2$). Measurements are averaged over 4 locations for the solid-solid sections and 3 locations for the hole, empty or filled. (c-d) Temporal evolution of the normalized cumulated inter-event slip $S^{solid}(t)$ and $S^{patch}(t)$, i.e. the interfacial slip excluding the slip occurring during rapid slip events (c) for the empty-hole experiment and (d) for the granular experiment. Each dot corresponds to one measurement performed at the end of an inter-event period. Color code and symbols are the same as in (a-b). Inset: the cumulated inter-event slip $\delta_{IE}^{solid}(t)$ and $\delta_{IE}^{patch}(t)$ without normalization (same colors as in main plots).”

Fig. 3 caption: “**Fig. 3: Slow slip measurements.** (a) Converged values of the normalized cumulated inter-event slip (top) S_f^{patch} for the empty-hole (blue circles) and the granular patch (brown diamonds) and (bottom) S_f^{solid} for the solid-solid sections of the interface, as a function of the loading contrast C_σ . (b) Differential normalized cumulated inter-event slip $S_f^{patch} - S_f^{solid}$ for empty-hole (dark blue circles) and granular (blue diamonds) experiments as a function of the loading contrast C_σ . Inset: mean frequency $\langle 1/\Delta T \rangle$ of the stick-slip cycle as a function of $S_f^{patch} - S_f^{solid}$.”

■ Comment 2.3: At lines 122-123, the authors argue that the mean stick-slip frequency $\langle f_{ss} \rangle$ may be a relevant control parameter for the occurrence of rapid slip events. However, in their case, the energy content of the seismic events looks almost constant at a given LC. How would this observation fare in realistic conditions, when stick-slip events distributes over multiple time and energy scales?

Response: Our sentence was unclear. What we meant is that $\langle f_{ss} \rangle$ is the observable in our experiment, and the cumulative inter-event slip S seems to be a relevant control parameter for

the frequency increase. In some sense, the frequency increase is the smoking gun of what happens locally around the granular patch. In realistic conditions, the mean frequency of the stick-slip cycle is probably not an important parameter, but what matters would be the mechanisms at play during an inter-event period, because it would control the occurrence of the next earthquakes. We have modified the sentence that now reads:

l.157: “We note that the mean stick-slip frequency of the experiments, $\langle 1/\Delta T \rangle$, increases with $S_f^{patch} - S_f^{solid}$, suggesting that this latter quantity, the differential cumulated inter-event slip, is a relevant control parameter for the rapid slip events occurrence.”

- Expansion of the slipping zone with the normal load
- Comment 3.1: For clarity purpose, it would be better to show the nucleation location (defined at lines 144-145) on Figure 4a.

Response: We have added it on Fig. 4a and the new Fig. 4b (refer to our response to comment 3.5).

- Comment 3.2: Can the authors confirm that each F_s/F_n drop observed on Figure 1c correspond to the rupture of the entire frictional interface (as suggested at lines 142-143)? I just want to make sure that the similarities in Figure 1c.iv and Rubinstein et al. (2007, doi:10.1103/PhysRevLett.98.226103)’s Figure 2b cannot be attributed to partial ruptures of the interface.

Response: Indeed, in our experiments, all the ruptures are spanning the entire interface. We have added a sentence in the main text to clarify it:

l.126: “The high-frequency strain measurements shown in Fig. 4 allow us to verify that, in our experiments, each force drop corresponds to a rupture that spans the entire interface, and not to arrested ruptures that may act as precursors to the system-spanning ruptures²⁷.”

- Comment 3.3: I have worked on a similar experimental setup and I know how difficult it can be to accurately set a time of departure from linearity on strain gages data. For a better understanding of Figure 4d, it would be interesting to show each linear/non-linear time intervals selected in the example of Figure 4c.

Response: To perform this measurement, we consider for each strain gauge each loading periods before each rapid stress drop of one experiment and determine per event if the evolution of the shear strain is sublinear or linear. We find that labeling all events on Fig. 5a-b (initially Fig. 4b) make it seriously crowded. In the revised version, we have labeled few events and added lines directly on the plots to better show the deviation from linearity.

The new Fig. 5a-b is now:

Fig. 5 a and b: (a-b) Temporal evolution of the shear strain $\varepsilon_{xy}(x, t)$ measured at 315 Hz for the 10 strain gages at locations x_{SG} , over (a) an empty-hole experiment at $C_\sigma = -1$, and (b) a granular experiment at $C_\sigma = 1.13$. For each event and each strain gage, the evolution of the shear strain during the inter-event period is determined as linear (L) or sublinear (SL). All the events of the empty-hole experiment in (a) are linear (L). Black dotted lines in (b) are a fit of the shear strain over the last points before a stress drop.

■ **Comment 3.4:** In the case where the pre-slip is large (granular experiments, large LC), is it possible to compare the DIC predictions of the sliding region to that inferred from the strain gages?

Response: The experiments presented in this manuscript have been performed with 4 tracking locations along the solid-solid sections of the interface for the interfacial slip measurements. We indeed observe a spatial dependence on the measurement of the inter-event cumulated slip $S_f^{\{i\}}$ as shown on the graphs below. The solid-solid measurements in Figs 2 and 3 in the manuscript correspond to the average of the 4 solid measurements below.

Figure for comment 3.4: Converged values of the normalized cumulated inter-event slip at each tracking location: 4 locations along the solid-solid interfaces, 1 location in the middle of the granular patch.

We have added data in Figs. 4 and 5, where we compute the averaged values of nucleation location (Fig. 4d) and of the slipping patch length (Figs. 5d and 6a) per range of loading. In

Fig. 5d, we plot one against the other and find that these two quantities are in agreement, confirming the validity of our detection method of the central slipping patch.

New Fig. 5d: (d) Comparison of the slipping patch length, $\langle \ell_{slip} \rangle$, corresponding to the averaged length of the slipping patch obtained from the data shown in (c), normalized by the hole length, $\ell_{hole} = 30$ mm, and the nucleation distance from the center $\langle d_{nuc} \rangle$, normalized by the hole half-length $\ell_{hole}/2$. Circle corresponds to empty-hole experiments and diamonds to granular experiments.

We comment this figure in the revised main text:

1.224: “The larger the central width of the distribution, the longer the slipping zone. The slipping patch length, $\langle \ell_{slip} \rangle$, is defined as the averaged spatial extent of the region where the sublinear shear strain evolution is detected for each loading phase, i.e., each inter-event period, per range of C_σ . We first check the consistency of the two previous measurements: rupture nucleation location from high-frequency strain measurements and slipping patch extension from quasi-static loading curves (Fig. 5d). The two lengths $\langle d_{nuc} \rangle$ and $\langle \ell_{slip} \rangle$ are indeed comparable, especially when the slipping patch becomes longer than the initial compositional heterogeneity, i.e., the hole length, $\ell_{hole} = 30$ mm. Furthermore, as expected from the correlation between $\langle d_{nuc} \rangle$ and $\langle \ell_{slip} \rangle$, Fig. 6a shows that the slipping patch length increases with C_σ . This plot indicates that the larger the normal load carried by the granular patch, the longer the slipping patch, that extends beyond the initial compositional heterogeneity and thus into the solid-solid sections of the interface, which otherwise may rupture seismically under different loading conditions.”

■ Comment 3.5: Would it be possible to superimpose (maybe just in the answer) to Figure 4a the estimated sliding region before nucleation, as predicted from the procedure of Figure 4c? How does a nucleation event look for cases at larger LC? Other examples of Figure 4a could be added as supplemental material.

Response: The original Fig. 4 becomes Fig. 4 and Fig. 5 in the revised version. Following the reviewer's suggestion, we now show in Fig. 4 two examples of rupture propagation for a low and a high loading contrast. We prefer not to show the slip length obtained from the procedure in Fig. 5 in the main figure to avoid confusion for the reader between the two types of measurements. Here is the figure with the indication of the slipping length:

Figure for comment 3.5: Slipping length extracted from the sublinear evolution of shear strain for the events shown in Fig. 4a-b of the revised manuscript.

■ Comment 3.6: How would Figure 4a looked if the portion [75mm, 135mm] was creeping before nucleation (~ -0.1 ms)? Can we be sure that rupturing of the [75mm, 135mm] portion of the fault is not a secondary rupture (Shi et al., 2023, doi:10.1038/s41467-023-44086-1)?

Response: Each rapid event is detected via an accelerometer that triggers the high frequency measurements record, before and after the trigger in the time interval $[-10 \text{ ms}, 10 \text{ ms}]$ where 0 is the trigger time. Creep would induce some strain release before the first rupture, something that we do not observe. And the primary rupture is easily detectable as it is the first time where we observe a strong variation of the strain signal. Below are shown the signals of Figs. 4a-b for over the total record time.

Figure for comment 3.6: Temporal evolution of the shear strain for the two rapid vents shown in Fig. 4a, b over a longer duration (10 ms instead of 0.2 or 0.4 ms in the main figure).

- Expansion of the slipping zone with the normal load

■ Comment 4.1: As shown by the authors, the presence of the granular material introduce two effects: (1) early sliding at the fault center, and (2) the creation of a positive stress gradient. I would hypothesize that (1) without (2) would create early dynamic ruptures with decreased energetic content due to lower applied shear force F_s . But (2) promotes stable sliding (decrease of the crack driving force) which permits to dissipate further the energy injected to the system through aseismic slip. Am I interpreting the authors' results correctly?

Response: We detect two possible scenarios for understanding the sliding of the solid regions surrounding the granular patch. One is a local reduction in normal stress, in which case conditions (1) and (2) put forward by the reviewer are necessary. A second hypothesis is creep of the contacts. In this case, only condition (1) is required. We discuss further the perspectives to discriminate between these two effects in the reply to comment 5.1.

■ Comment 4.2: Can the author discuss potential outcomes if the creep patch extends to the fault's ends? (See recent work for some insights on this question: <https://meetingorganizer.copernicus.org/EGU23/EGU23-15559.html>)

Response: Thank you for pointing out this work. In a finite size system with free boundaries, as our experimental setup, the case where the slow slip extends to the interface's end does not seem relevant to us because it is simply a case where the interface is in stable sliding. In fact, we have excluded this type of data (very high loading contrast) to focus on cases where part

of the interface is locked. That said, it might be insightful to study cases where a fault is in a slow slip regime but at a rate depending on the location. We can think about a realization of such a system for future work.

- Discussion

- Comment 5.1: The authors do not mention the positive normal stress gradient observed for large LC (i.e. creeping fault). It looks to me that this effect (which may be related to structural/elastic heterogeneities) may be important to explain stable sliding. Can they comment on this?

Response: In our understanding, a gradient of normal stress per se would not induce stable sliding, but the local value of the normal stress may be low enough that the stick-slip criterion is no longer fulfilled. In fact, there is no reason for the gradient to influence the slip regime (i.e. stick-slip or stable sliding). Once the rupture is propagating, the normal stress gradient affects the velocity profile by modifying the fracture energy and the shear loading (see Svetlizky et al, PRL 2017, ref 14).

The local value of the normal stress, on the other hand, can affect the slip regime. This is one of the reasons we mentioned for the extension of the slow slip patch into the solid-solid interface. In our view, shear-induced dilatancy of the granular material could be responsible for this strong gradient; the normal stress carried by the granular patch increases as the patch slides, locally reducing the normal stress at the edges of the patch. The other possible reason is a weakening of the solid contacts by a creep phenomenon, which would be induced by a very low shear rate imposed by the slow sliding of the granular patch.

Our experiment does not allow us to conclude on the importance of each of these two mechanisms, since the duration of an inter-event is too short to be able to perform slip measurements over this period. One solution would be to find another system where slow sliding occurs locally, without shear-induced dilation. It might happen with a localized patch of lubricant. Indeed, the corresponding author has shown in previous work that the viscosity of a lubricant plays a role in the shear stress level at which frictional rupture initiates (Bayart et al, PRL 2017 and Bayart et al, JGR 2019). Now that we have established in the present study that slow sliding facilitates rupture nucleation, it may be interesting to return to such a system. Another possibility would be to use a system with a local reduction of the normal load, using pistons or machining the block surfaces. However, this is not currently possible with our setup.

We have expanded this discussion in the text to make clearer the possible role of local normal stress reduction. The text now reads:

I.295: “What causes microcontacts to **slide** instead of breaking dynamically? There may be two possible mechanisms involved. The first **mechanism is the creep of the microcontacts loaded by the slipping granular patch at a very low rate, of the order of few $\mu\text{m/s}$ (Fig. 2d). The creep regime has been observed for several experimental systems^{40,41}**. In fact, frictional contacts age with time, strengthening the interface, **while low-rate shear loading induces plastic flow of the contacts⁴²**. The comparable timescales between these two processes may lead to **creep rather than dynamic failure of the microcontacts**. The second **mechanism** is the shear-induced dilatancy of the granular patch. A dense granular material dilates, i.e. expands, when sheared⁴³, which can locally release the normal stress on the surrounding

areas, leading to a transition to stable sliding³⁵. In our experiments, we observe a local reduction in normal stress near the edges of the patch as the normal load imposed on the patch increases, although it never disappears (Figs. 1c). The local reduction in normal stress could explain the transition from stick-slip to steady sliding^{44,45}. This last mechanism emphasizes the importance of considering the mechanical behavior of heterogeneities in models.”

Another effect of the reduction in normal stress near the patch corners is the reduction in normal stress at the rupture nucleation location, which leads to a reduction in the interfacial fracture energy at the rupture tip. This effect plays the same role as the slipping patch elongation in destabilizing the seed crack at lower shear loading. This effect was not presented in the original version of the manuscript, but is now included following a comment from reviewer #2. We have added a figure (Fig. 6b) and a new paragraph in the discussion about the role of the fracture energy in our system.

Fig. 6: Slipping patch length and normal stress leading to rupture destabilization. (a) Evolution of the slipping patch length, $\langle \ell_{slip} \rangle$, corresponding to the averaged value of the events counts in Fig. 5c, normalized by the hole length, ℓ_{hole} , as a function of the loading contrast for the empty-hole experiments (circle) and the granular experiments (diamonds). Error bars corresponds to the standard deviations of the distributions in Fig. 5c. Measurements for each individual event appear in gray, circle or diamond. (b) Averaged values of $\sigma_{yy}^0(x_{nuc})$, the normal stress measured at the nucleation locations x_{nuc} of each rapid event, for each range of C_σ used for the histograms in Fig. 4c and Fig.5c for the empty-hole experiments (blue circle) and the granular experiments (blue diamonds). Measurements of $\sigma_{yy}^0(x_{nuc})$ for each individual event appear in gray, circle or diamond.

1.264: “The local value of the fracture energy G_c also affects the stress τ_c at which a rupture initiates. The fracture energy G_c strongly depends on the local normal load²⁷ and, because of the heterogeneous normal stress distribution along the interface in our experiments, we cannot confidently extract the G_c value from our strain measurements. However, the Griffith criterion must be fulfilled locally at the rupture tip for a rupture to be destabilized. Although G_c cannot be measured, we do measure the normal stress at the nucleation point prior to each event. Since G_c is linearly dependent on normal stress^{27,36}, this measurement gives an indication of the variation trend of the fracture energy at the rupture tip with the loading contrast. Fig. 6b shows the value of σ_{yy} at the nucleation point before the rapid events, $\langle \sigma_{yy}^0(x_{nuc}) \rangle$ where x_{nuc} is the nucleation location, averaged over all the events of the empty-hole experiments (circle) and granular experiments (diamonds) for the four ranges of

loading contrast, C_σ , shown in Figs. 4c and 5c. This plot demonstrates a decreasing trend of $\langle \sigma_{yy}^0(x_{nuc}) \rangle$ with the loading contrast, C_σ . Therefore, the variation of the fracture energy G_c at the rupture nucleation location may play a role in reducing the critical stress for the seed crack destabilization, an effect that is additive to that of the elongation of the slipping patch. The two parameters ℓ_{slip} and G_c cannot be controlled independently of each other in our experiment, due to the presence of the compositional heterogeneity, i.e. the granular patch.”

- Methods

- Comment 6.1: There is no much information on the optical tracking method. If DIC was used, what was the subset size? How is the correlation performed (manual/industrial software)?

Response: We have added more information about the tracking method, cf our reply to comment 1.1.

Additional note: There is no information on data availability. I am advocating for open science and would be glad that the authors put their experimental data in open access (at the very least for the data presented in the manuscript).

Response: The data presented in the manuscript are now available on a data repository. We are working on making all of the experimental data accessible, but this will require a lot more work, as a user manual will be needed.

Minor formatting comments:

- line 66 “Poisson ration” -> “Poisson’s ratio” : done
- line 189 σ_c -> τ_c : done
- line 241 $3 < E$ -> $3 < E$: the text has been modified.

REVIEWER COMMENTS

Reviewer #1 (Remarks to the Author):

I am satisfied with the carefully prepared revised manuscript, the authors' reply to my review, as well as that to the other reviewers. In general, I salute the quality of the review process.

In particular, I believe that the manuscript improved in terms of quantitative results, and I am now confident that the manuscript reflects the results that the authors could have obtained (not less, not more).

The key point of discussion in all the reviews is the question of the nucleation length. The authors have convinced me that the current experiments cannot make a quantitative statement, but do provide qualitative support of a relation $\Delta\sigma \sim \ell^{-1/\nu}$ whatever the exponent may be. In that respect, I find that the authors' formulation is somewhat biased to an exponent given by a Griffith argument. Several theoretical works debate this, and the authors cannot give quantitative support. Therefore, my only gentle suggestion is to be reserved in their statements of a Griffith criterion for nucleation. Specifically, In the added text around l.245 I would suggest the last sentence to be along the lines of "However, we can verify that the macroscopic stress drop as a function of the length of the slipping paths displays a singularity. However, we cannot fit an exponent with accuracy".

Finally,

* in the added text at l.34: " $\dot{\phi} = 0$ " when the fault is slipping at the same velocity as the loading rate" (not "than the loading rate");

* l.140: "where {} refers to the solid-solid (for an empty hole) or patch (for a hole filled with cylinders).

* Fig. 4: I would use ℓ_c instead of d_{nuc} .

* Fig. 4 caption, last sentence: "The circular marker ... and the diamonds" (or "diamond markers").

Reviewer #2 (Remarks to the Author):

I have reviewed both the initial manuscript and the subsequent rebuttal letter, along with the revised manuscript. I wish to express my appreciation for the comprehensive response provided in the rebuttal letter and the substantial modifications made to the manuscript. Specifically, the addition of Figure 6 greatly aided my understanding of aspects that were not clear in the initial version. I commend the extension of the particle tracking methods and the inclusion of detailed explanations in the main text. Furthermore, I value the expanded discussion on the potential implications for natural earthquakes, as well as the thorough consideration of experimental limitations and avenues for further research. The reserved tone adopted regarding the demonstration of the Griffith criterion for this more complex type of rupture is also commendable. At this stage, I have only very minor revisions.

Line 21: Are you referring to 'Nucleation dynamics'?

Line 34: The paragraph on fault coupling is interesting but not well integrated with the previous part of the introduction.

Line 113: "force drops and therefore to an increased stick-slip frequency," could you please provide references?

Lines 202, 204: I am still confused, perhaps it's my misunderstanding, but is the seed crack a velocity-strengthening region as indirectly mentioned in the introduction, or is it a velocity-weakening region? Or maybe this is of no importance?

Line 320: can you reformulate as : amount of slow slip accommodated by the slipping patch?

Line 325: Do you have any references for this?

Reviewer #3 (Remarks to the Author):

Review of NCOMMS-24-00631A:

I'd like to thank the authors for their thorough response to the reviewers' comments. Overall, I think the manuscript gained in clarity during the first round of review. The experiments conducted in this work are novel and should provide quantitative means to address important questions on the nucleation and propagation of frictional ruptures in complex settings. However, I am still not entirely convinced by the explanations proposed to rationalize the outputs of the experiments. My main concerns are that: (1) the initiation/nucleation analysis is entirely based on LEFM, disregarding current state-of-the-art on the matter, and (2) the origin of prolonged slow slip (one key and novel point of the manuscript) is still a matter of debate.

Point 1 – Initiation/nucleation analysis based on LEFM:

In the abstract, the author put forth “fracture tools in elucidating the dynamics of complex faults”. These tools boil down to Griffith's criterion. While it has been known since Andrews' seminal work (JGR, 1976, doi:10.1029/JB081i032p05679) that frictional cracks can become unstable once their size exceeds a critical value set by Griffith's criterion, this nucleation criterion is not the only one found in the literature. Other criteria may involve the rate W of frictional weakening with slip – independently of G_c – (Campillo and Ionescu, JGR, 1997, doi:10.1029/97JB01508), the parameters a & b for rate-and-state friction laws (Rubin and Ampuero, JGR, 2005, doi:10.1029/2005JB003686), or a criterion based on a global energetic stability analysis – and not rate-of-energy-based as Griffith's criterion – (Rice and Uenishi, Int J Frac, 2010, doi:10.1007/s10704-010-9478-5). The latter is strikingly similar to the finite fracture mechanics framework mentioned by Referee 2. Numerical simulations based on slip-weakening (Castellano et al., JMPS, 2023, doi:10.1016/j.jmps.2022.105193) or rate-and-state (Rubin and Ampuero, JGR, 2005, doi:10.1029/2005JB003686) friction laws confirmed a transition between Andrews' theory and Campillo and Ionescu's prediction, depending on the initial background stress acting on the interface. Despite recent experimental progress (Gvirtzman and Fineberg, Nature Phys, 2021, doi:10.1038/s41567-021-01299-9), the identification of a comprehensive nucleation criterion is still an open question. Note however, that all the models mentioned above

feature a decrease of the nucleation length with increasing normal stress, a feature that the authors highlight in their new Figure 6.

As the authors pointed out several times in their response to the reviewers, they may not have the resolution to compute G or G_c in their experiments. As a result, they should not be able to conclude on which criterion is suitable for crack initiation. Yet, they state (lines 247-248) that “the initiation criterion of a crack, the Griffith criterion, is $G_s=G_c$ ”.

Moreover, Andrews’ theory assumes that the shear stress has reached its residual value in the wake of the crack tip. Yet, one can observe in Figure 4a that the rupture is associated with non-negligible weakening within the already-slipping patch (see Figure answering my comment 3.5 for an estimate of the slipping patch size). This is not the case in Figure 4b. This suggests that these rich experiments may alternatively belong to the “yielding” regime or in the “fracture” regime, as identified by Castellano et al. (JMPS, 2023, doi:10.1016/j.jmps.2022.105193). This could be explained by the fact that strong loading contrast promote larger value of K_{II} for small cracks (see point below), so that enough slip is achieved within the slipping patch for the shear stress to achieve its residual value.

In conclusion, I do not think that the paper presents a clear view on the established literature regarding rupture initiation/nucleation. Moreover, I am not convinced that the authors possess the proper dataset to support Griffith’s criterion as THE initiation/nucleation criterion.

Point 2 – Origin of prolonged slow slip:

In the revised manuscript, most of the discussion on the spatial extent of the slow sliding region is focused on the fracture energy G_c . However, the impact of normal stress heterogeneities can be more prevalent as positive gradient of normal stress in the propagation direction can lead as negative stress drops, which are often associated with stable sliding or crack arrest (Garagash and Germanovich, JGR, 2012, doi:10.1029/2012JB009209). I think it would be very useful to have fracture mechanics insights of the impact of the loading contrast on some energy release rate profiles, even if the spatial resolution of the strain gages do not allow for quantitative agreement on Griffith’s criterion.

Other comments:

I have other comments regarding the measurements of G , G_c , and ℓ_{nuc} .

- Comment 1: The authors state that they do not possess the spatial resolution required to estimate G and G_c . However, previous works performed by the corresponding author showed estimates of G and G_c with an array of 14 strain gages (instead of 10). How many strain gages would be needed to achieve quantitative predictions? Could static (for stress drop and G) and dynamic (for G_c and rupture position) strain gages be combined to do so?
- Comment 2: The authors state that they cannot measure G_c , but at the same time they write line 270 that “ G_c is linearly dependent on normal stress”. As experiments with negative loading contrast show dynamic rupture along the whole solid-solid interface, it should be possible to fit K_{II} from the strain singularity at some locations, and measure G_c using Griffith’s criterion. This would give some value of G_c for various normal stress, and then one could build on the previously-evoked linearity of G_c with normal stress to infer it for higher values of loading contrast (at least at some locations).
- Comment 3: The authors argue that they cannot estimate K_{II} (and so G) due to a lack of precision on the residual stress τ_r . The normal stress profiles shown in Figure 1b look resolved enough to compute rough estimates of G from the shear stress. This is especially true if the residual friction is an interface property, as suggested from previous works done the corresponding author (Bayart et al., Nature Phys, 2016, doi:10.1038/nphys3539). However, recent experimental work performed on a similar setup (Rubino et al., Nature Comm, 2017) show that this might not be the case when thermal processes are involved. In these experiments where normal stress is varying with loading contrast, is the residual shear stress still nearly proportional to the local normal stress?
- Comment 4: I do not fully agree with the red line shown in Figures 4a&b. The passage of a rupture is usually associated with a decrease of shear stress (see Berman et al., PRL, 2020, doi:10.1103/PhysRevLett.125.125503), while arrival times are sometimes located in the ascending phase of shear stress, i.e. when the stress concentration ahead of the crack tip starts loading the material point behind the strain gage.
- Comment 5: The authors discuss the implications of their experiments on the understanding of tremors (line 319-321). In the case of anthropogenic fluid injection tremors are often observed both within and at the edges of what may be a slow slipping region (Sáez and Lecampion, Phil Trans Roy Soc A, 2023, doi:10.1098/rspa.2022.0810). Dynamic events associated with tremoring activity is often considered as contained ruptures

(Veedu and Barbot, Nature, 2016, doi:10.1038/nature17190). Here, these events correspond to ruptures of the entire fault interface. How do the authors explain this difference? How is it possible to connect their lab measurements to field observations?

REVIEWER COMMENTS

Reviewer #1 (Remarks to the Author):

I am satisfied with the carefully prepared revised manuscript, the authors' reply to my review, as well as that to the other reviewers. In general, I salute the quality of the review process.

In particular, I believe that the manuscript improved in terms of quantitative results, and I am now confident that the manuscript reflects the results that the authors could have obtained (not less, not more).

The key point of discussion in all the reviews is the question of the nucleation length. The authors have convinced me that the current experiments cannot make a quantitative statement, but do provide qualitative support of a relation $\Delta \sigma \sim \ell^{-1/\nu}$ whatever the exponent may be. In that respect, I find that the authors' formulation is somewhat biased to an exponent given by a Griffith argument. Several theoretical works debate this, and the authors cannot give quantitative support. Therefore, my only gentle suggestion is to be reserved in their statements of a Griffith criterion for nucleation. Specifically, In the added text around l.245 I would suggest the last sentence to be along the lines of "However, we can verify that the macroscopic stress drop as a function of the length of the slipping paths displays a singularity. However, we cannot fit an exponent with accuracy".

Response: We thank the referee for his positive feedback on our manuscript. Based on his comment and the comments of the other referees, we have modified the discussion of the Griffith criterion to make it clearer that we are using this criterion as an illustrative case of a rupture initiation criterion and that the aim of the study is not to validate one criterion over the other. As a result, the paragraph that previously began on l.245 has been modified, as well as the conclusion:

l.245 to 267: From "Our interpretation is that the central slipping zone acts as a nucleation center, i.e., an initial rupture destabilized into rapid ruptures along the interface..." to "... Although our experimental system is not designed to validate a criterion for rupture initiation over another, we can verify that the response of the system to the presence of a slipping patch follows the same trend as in the presence of an initial rupture."

l.342 to 350: "In summary, we have shown that a slowly slipping area within a frictional interface acts as a nucleation center,..." to "Furthermore, the use of a fracture mechanics-based description may allow the use of off-fault measurements to predict the destabilization of a slowly slipping patch."

Finally,

* in the added text at l.34: " $\phi = 0$ when the fault is slipping at the same velocity as the loading rate" (not "than the loading rate");

Response: Following referee #2 comment, this sentence has been revised:

l. 34: "To describe the slip behavior of a fault, the coupling compares the slip velocity of the fault to the loading velocity. The fault is uncoupled when it slides at the loading velocity and coupled when it is locked during the inter-seismic period."

* l.140: "where {1} refers to the solid-solid (for an empty hole) or patch (for a hole filled with cylinders).

Done

* Fig. 4: I would use ℓ_c instead of d_{nuc} .

Response: We find that ℓ_c can be confused with some critical length (for nucleation for example) and we prefer to keep a name that refers directly to nucleation.

* Fig. 4 caption, last sentence: "The circular marker ... and the diamonds" (or "diamond markers").

Done

Reviewer #2 (Remarks to the Author):

I have reviewed both the initial manuscript and the subsequent rebuttal letter, along with the revised manuscript. I wish to express my appreciation for the comprehensive response provided in the rebuttal letter and the substantial modifications made to the manuscript. Specifically, the addition of Figure 6 greatly aided my understanding of aspects that were not clear in the initial version. I commend the extension of the particle tracking methods and the inclusion of detailed explanations in the main text. Furthermore, I value the expanded discussion on the potential implications for natural earthquakes, as well as the thorough consideration of experimental limitations and avenues for further research. The reserved tone adopted regarding the demonstration of the Griffith criterion for this more complex type of rupture is also commendable. At this stage, I have only very minor revisions.

Line 21: Are you referring to 'Nucleation dynamics'?

Response: We have modified this sentence to better convey the message of our study. It now directly refers to the nucleation dynamics:

l.21: "Our results reveal that the presence of a slow-slip region modifies the dynamics of rupture nucleation. The identification of such a mechanism paves the way for the construction of novel models accounting for the slow-slip region evolution under varying conditions, thereby improving fault monitoring and seismic hazard mitigation."

Line 34: The paragraph on fault coupling is interesting but not well integrated with the previous part of the introduction.

Response: We have modified this sentence to better justify the introduction of the coupling term:

l. 34: "...without being earthquake precursors⁷. To describe the slip behavior of a fault, the slip velocity of the fault and the loading velocity are compared through the coupling term. The fault is uncoupled when it slides at the loading velocity and coupled when it is locked during the inter-seismic period."

Line 113: "force drops and therefore to an increased stick-slip frequency," could you please provide references?

Response: This effect comes directly from the resolution of a spring-mass system with a dynamic friction coefficient smaller than the static one. This system is an oscillator with a pulsation $\omega_0 = \sqrt{k/m}$, where k is the spring stiffness and m is the mass corresponding to the normal force in our system.

Lines 202, 204: I am still confused, perhaps it's my misunderstanding, but is the seed crack a velocity-strengthening region as indirectly mentioned in the introduction, or is it a velocity-weakening region? Or maybe this is of no importance?

Response: The initial rupture is composed of the granular patch and the neighboring solid-solid sections. We don't know the frictional properties of the granular patch, we only observe that it slides when sheared, which could be due to a velocity-strengthening behavior. The solid-solid interface sections have a velocity- or slip-weakening behavior because seismic ruptures can propagate through them. But the details of the friction laws are not important in our analysis. The correct law would affect the residual stress value or its evolution with inertial slip.

Line 320: can you reformulate as : amount of slow slip accommodated by the slipping patch?

Response: Done

Line 325: Do you have any references for this?

Response: This sentence is a statement based on our results. We are not aware of field observations reporting the extension of a slow slip region over time.

Reviewer #3 (Remarks to the Author):

Review of NCOMMS-24-00631A:

I'd like to thank the authors for their thorough response to the reviewers' comments. Overall, I think the manuscript gained in clarity during the first round of review. The experiments conducted in this work are novel and should provide quantitative means to address important questions on the nucleation and propagation of frictional ruptures in complex settings. However, I am still not entirely convinced by the explanations proposed to rationalize the outputs of the experiments. My main concerns are that: (1) the initiation/nucleation analysis is entirely based on LEFM, disregarding current state-of-the-art on the matter, and (2) the origin of prolonged slow slip (one key and novel point of the manuscript) is still a matter of debate.

Response: We thank the referee for their positive feedback about the improvement of the manuscript following the first round of review. Concerning the new revision, we think that all of their comments on the revised version stem from the fact that the message we are trying to convey is not clear enough in the current version of the manuscript. The goal of our study *is not* to demonstrate the validity of one correct initiation criterion over another. The interest of our study is to shed light on the interaction mechanism between a slowly slipping region and locked regions of a frictional interface. The mechanism highlighted is that the slowly slipping patch acts as an initial rupture that destabilizes into dynamic ruptures when the critical shear stress is reached, and that this patch extends beyond the compositional heterogeneity when normal loading is increased. This results in an increased frequency of rapid slip events, i.e., an increased stick-slip frequency.

The experimental system we are considering is complex, by its composition and by the stress distribution resulting from the far-field loading, and this complexity prevents us from performing fracture mechanics analyzes as the corresponding authors has done in previous works. In this context, we use Griffith's criterion as an illustration of an initiation criterion, and we agree with the reviewer that the discussion as presented in the manuscript does not sufficiently reflect the fact that this is one criterion among others (see our response to Point 1). We have carefully revised the manuscript to better convey our message and we answer the comments of the referee below.

Referee: Point 1 – Initiation/nucleation analysis based on LEFM:

In the abstract, the author put forth “fracture tools in elucidating the dynamics of complex faults”. These tools boil down to Griffith’s criterion. While it has been known since Andrews’ seminal work (JGR, 1976, doi:10.1029/JB081i032p05679) that frictional cracks can become unstable once their size exceeds a critical value set by Griffith’s criterion, this nucleation criterion is not the only one found in the literature. Other criteria may involve the rate W of frictional weakening with slip – independently of G_c – (Campillo and Ionescu, JGR, 1997, doi:10.1029/97JB01508), the parameters a & b for rate-and-state friction laws (Rubin and Ampuero, JGR, 2005, doi:10.1029/2005JB003686), or a criterion based on a global energetic stability analysis – and not rate-of-energy-based as Griffith’s criterion – (Rice and Uenishi, Int J Frac, 2010, doi:10.1007/s10704-010-9478-5). The latter is strikingly similar to the finite fracture mechanics framework mentioned by Referee 2. Numerical simulations based on slip-weakening (Castellano et al., JMPS, 2023, doi:10.1016/j.jmps.2022.105193) or rate-and-state (Rubin and Ampuero, JGR, 2005, doi:10.1029/2005JB003686) friction laws confirmed a transition between Andrews’ theory and Campillo and Ionescu’s prediction, depending on the initial background stress acting on the interface. Despite recent experimental progress (Gvirtzmann and Fineberg, Nature Phys, 2021, doi:10.1038/s41567-021-01299-9), the identification of a comprehensive nucleation criterion is still an open question. Note however, that all the models mentioned above feature a decrease of the nucleation length with increasing normal stress, a feature that the authors highlight in their new Figure 6.

As the authors pointed out several times in their response to the reviewers, they may not have the resolution to compute G or G_c in their experiments. As a result, they should not be able to conclude on which criterion is suitable for crack initiation. Yet, they state (lines 247-248) that “the initiation criterion of a crack, the Griffith criterion, is $G_s=G_c$ ”.

Moreover, Andrews’ theory assumes that the shear stress has reached its residual value in the wake of the crack tip. Yet, one can observe in Figure 4a that the rupture is associated with non-negligible weakening within the already-slipping patch (see Figure answering my comment 3.5 for an estimate of the slipping patch size). This is not the case in Figure 4b. This suggests that these rich experiments may alternatively belong to the “yielding” regime or in the “fracture” regime, as identified by Castellano et al. (JMPS, 2023, doi:10.1016/j.jmps.2022.105193). This could be explained by the fact that strong loading contrast promote larger value of K_{II} for small cracks (see point below), so that enough slip is achieved within the slipping patch for the shear stress to achieve its residual value.

In conclusion, I do not think that the paper presents a clear view on the established literature regarding rupture initiation/nucleation. Moreover, I am not convinced that the authors possess the proper dataset to support Griffith’s criterion as THE initiation/nucleation criterion.

Response: We thank the referee for this important comment. We recognize that the message we intend to convey is not sufficiently clear in the current version of the manuscript. As written before, the goal of our study is not to demonstrate the validity of one correct initiation criterion over another. The use of the fracture mechanics framework allows us to illustrate the role of the slipping patch in the destabilization of the rest of the interface, but we are aware that other criteria exist, and that they all predict the same trend for the evolution of the nucleation length. Actually, one could argue that the search for a single valid initiation criterion is probably unnecessary, since most criteria describe the same physics, but within different theoretical frameworks. The difference between criteria is the physical quantities they deal with. For some criteria, knowledge of the stresses *at* the interface is mandatory (but difficult to measure), while for others a characterization of the large-scale frictional properties is required (rate-and-state parameters *a* and *b*). In this study, we analyze the destabilization of the slowly slipping patch with the tools of fracture mechanics, which deal with the interfacial fracture energy, and we think that they are indeed useful to characterize the interplay mechanism with the off-fault measurements at hand. With sufficient measurement resolution, this tool could even become quantitative and predictive by determining the different terms of the Mushkelishvili integral, put forward in the previous point-by-point response to the referees. We have made substantial revisions to the manuscript to make more general statements about initiation criteria (see below).

Based on the experimental limitations of our setup, what we are able to do is verify that the frictional system responds to the presence of a slipping patch as it would in the presence of a model initial rupture. All initiation criteria include three ingredients: loading, rupture length and some characteristic quantity of the interface. Our Fig. 6 is intended to show the evolution of the rupture length and the evolution of the interface characteristic – the fracture energy in the chosen framework of fracture mechanics – with the loading applied to the patch. In fact, it shows that these 2 quantities evolve according to the 3rd one, i.e. a decrease of the shear resistance with the loading contrast.

Regarding the comment about lines 247-248 in the manuscript, we were only describing the results of pure fracture mechanics. At this stage, we do not apply the Griffith criterion to the slowly-slipping patch present along the interface in our experiment. We have now revised this paragraph to start with more general statements about the nucleation criterion (see below).

Regarding the comment on the weakening of the granular patch after the passage of the first rupture, we thank the reviewer for pointing out the possible existence of two rupture regimes. This question would benefit from further study in the case of an entirely granular frictional interface. In this study, we do not investigate the establishment of stable sliding within the granular patch, but only its effect on the initiation of dynamic ruptures. The Andrew's theory considers the residual stress reached after the passage of the 1st rupture. Any further weakening is due to rupture reflections or inertial sliding (Shi et al, Nature Comm. 2024, doi: 10.1038/s41467-023-44086-1).

We have revised the manuscript to introduce the nucleation criterion in a more general way, to clarify that we use the Griffith criterion as an illustrative case of the initiation criterion, and to present the advantage of using fracture mechanics for such a system (off-fault measurements).

1.16 (abstract): “By measuring the response of the fault to shear, we show that the role of the compositional heterogeneity is to serve as **an initial rupture**, reducing the macroscopic fault shear resistance.”

1.21(abstract): “Our results reveal that the presence of a slow-slip region modifies the dynamics of rupture nucleation. The identification of such a mechanism paves the way for the construction of novel models accounting for the slow-slip region evolution under varying conditions, thereby improving fault monitoring and seismic hazard mitigation.”

1.56: “Our experiment demonstrates a case where slow slip is responsible for the initiation of rapid slip events by acting as an initial rupture, which eventually propagates and destabilizes the interface.”

1.198: We replaced “We interpret this result as the fact that a slowly slipping granular section induces the formation of a slipping seed crack, destabilized following a fracture initiation criterion. A seed crack is a pre-existing crack, i.e. a line along which stresses have been released, that destabilizes once the shear stress reaches a threshold value, given by the Griffith criterion in elastic fracture mechanics.”

by

“We interpret this result as the fact that a slowly slipping granular section induces the formation of an initial rupture – a line along which stresses have been released – which destabilizes into a rapid rupture once a critical shear stress is reached³⁹⁻⁴¹.”

The discussion about the trend of the initiation criterion has been generalized:

From 1.245 to 1. 275: “Our interpretation is that the central slipping zone acts as a nucleation center, i.e., an initial rupture destabilized into rapid ruptures along the interface [...] resulting in the initiation of rapid slip events after a shorter loading time and thus to an increased stick-slip frequency.”

Finally, we have revised the conclusion to better convey the message of our study:

From 1.342 to 1.350: “In summary, we have shown that a slowly slipping area within a frictional interface acts as a nucleation center, [...] Furthermore, the use of a fracture mechanics-based description may allow the use of off-fault measurements to predict the destabilization of a slowly slipping patch.”

Referee: Point 2 – Origin of prolonged slow slip:

In the revised manuscript, most of the discussion on the spatial extent of the slow sliding region is focused on the fracture energy G_c . However, the impact of normal stress heterogeneities can be more prevalent as positive gradient of normal stress in the propagation direction can lead as negative stress drops, which are often associated with stable sliding or crack arrest (Garagash and Germanovich, JGR, 2012, doi:10.1029/2012JB009209). I think it would be very useful to have fracture mechanics insights of the impact of the loading contrast on some energy release rate profiles, even if the spatial resolution of the strain gages do not allow for quantitative agreement on Griffith’s criterion.

Response: We respectfully disagree with the referee on the first statement. The discussion of the spatial extent of the slipping patch occurs between lines 308 and 321 (in the new version of the manuscript). In this paragraph, we present a discussion of the potential reasons why the solid-solid interface may slip in the vicinity of the granular patch, leading to the existence of an extended slow-slip patch. The reasons given are a reduction in the normal load in the

vicinity of the patch due to the shear-induced dilatancy of the sheared granular material, or the creep of microcontacts driven at a very low rate. On the other hand, the discussion of the role of G_c from 1.276 to 1.293 concerns the mechanism of destabilization of the slow-slip patch into rapid rupture. In the context of fracture mechanics, l and the value of G_c at the nucleation location have an effect on the critical stress for destabilization. The results of our experiments show that these two quantities evolve according to the normal load carried by the granular patch, following the expected trend. This observation allows us to conclude that the slow-slip patch plays the role of a rupture that destabilizes when the critical shear stress is reached.

Regarding the normal stress gradient, it can indeed induce rupture arrest if the shear loading becomes too small compared to the fracture energy corresponding to the local value of the normal stress. However, this process occurs away from the nucleation location. On the other hand, a local reduction in normal stress can lead to stable sliding because the stick-slip transition is crossed. We mentioned this possibility in the discussion about the origin of slow slip (1.314-321). The complex geometry considered in this study prevents a complete fracture analysis of the problem, and the suggestion made by the referee is beyond the scope of this study.

Referee: Other comments:

I have other comments regarding the measurements of G , G_c , and ℓ_{nuc} .

- Comment 1: The authors state that they do not possess the spatial resolution required to estimate G and G_c . However, previous works performed by the corresponding author showed estimates of G and G_c with an array of 14 strain gages (instead of 10). How many strain gages would be needed to achieve quantitative predictions? Could static (for stress drop and G) and dynamic (for G_c and rupture position) strain gages be combined to do so?

Response: Previous work done by the corresponding author was dedicated to the study of the fracture properties of an interfacial rupture. In particular, for technical reasons, the determination of G_c requires a homogeneous normal stress distribution, strain measurements and contact area measurements (to ensure the block alignment and therefore the correct measurement of a 2D strain-field). As we do not have homogeneous normal stress distribution and no contact area measurements, we are not able to determine G_c , whatever the number of strain gauges.

The static measurements do not allow to determine G . Indeed, the value of G is related to the dynamic stress drop induced by the rupture first passage. A rapid slip event consists in a first stress drop induced by the passage of the rupture, which has been shown to be a shear crack for homogeneous systems, followed by a larger stress drop, due to rupture reflections (see Shi et al, Nature Comm. 2024, doi: 10.1038/s41467-023-44086-1) and subsequent inertial sliding. Eventually, it reaches a minimal shear stress value before reloading (see Fig.2d in Passelègue et al, J. Geophys. Res. Solid Earth (2016), doi:10.1002/2015JB012694). The static stress drop would provide a measurement of this maximal stress drop, which does not relate to the fracture energy.

Referee: Comment 2: The authors state that they cannot measure G_c , but at the same time they write line 270 that “ G_c is linearly dependent on normal stress”. As experiments with negative loading contrast show dynamic rupture along the whole solid-solid interface, it should be possible to fit KII from the strain singularity at some locations, and measure G_c using Griffith’s criterion. This would give some value of G_c for various normal stress, and

then one could build on the previously-evoked linearity of G_c with normal stress to infer it for higher values of loading contrast (at least at some locations).

Response: Although the strain curves are very similar to those expected from LEFM (see Fig. 4), the fit to the data is poor. We believe this is due to the 2D strain field assumption. To make this assumption, the rupture front must be perpendicular to the direction of the interface and the blocks must be thin. We cannot check the straightness of the front because contact area measurements are not performed, and our blocks are moderately thin, 10 mm thick instead of 5.5 mm in the experiments performed by the corresponding author dedicated to the study of fracture mechanics. Another hypothesis to explain the poor fit is the possible emergence of an unconventional singularity due to the heterogeneous residual stress distribution (see Barras, F. et al., EPSL 2020; Brener et al, JMPS 2021; Paglialunga et al, EPSL 2024). Eventually, the determination of G_c from our data leads to large error bars, hiding any variation of its value.

We also would like to stress that a validation of the Griffith criterion necessitates a knowledge of both G_c and G . For the reasons stated in our response to comments 1 and 3, G cannot be determined in this experiment. Therefore, we believe that attempting to provide the value of G_c with large error bars is not useful in this study, and that the evolution of the normal stress at the rupture nucleation location is a good proxy to evaluate the evolution of G_c with the loading contrast.

We mention the existence of unconventional singularity for frictional crack in the revised manuscript:

l.165: “They are classical shear cracks for a homogeneous system of two flat solids in contact^{30,35}. Sometimes linear fracture mechanics must be modified to account for the inhomogeneous residual stress³⁶⁻³⁸.”

36 Barras, F. et al. The emergence of crack-like behavior of frictional rupture: Edge singularity and energy balance. *Earth Planet. Sci. Lett.* 531, 115978 (2020).

37 Brener, E. A. & Bouchbinder, E. Theory of unconventional singularities of frictional shear cracks. *J. Mech. Phys. Solids* 153, 104466 (2021).

38 Paglialunga, F., Passelègue, F., Lebihain, M. & Violay, M. Frictional weakening leads to unconventional singularities during dynamic rupture propagation. *Earth Planet. Sci. Lett.* 626, 118550 (2024).

Referee: Comment 3: The authors argue that they cannot estimate K_{II} (and so G) due to a lack of precision on the residual stress τ_r . The normal stress profiles shown in Figure 1b look resolved enough to compute rough estimates of G from the shear stress. This is especially true if the residual friction is an interface property, as suggested from previous works done the corresponding author (Bayart et al., *Nature Phys*, 2016, doi:10.1038/nphys3539). However, recent experimental work performed on a similar setup (Rubino et al., *Nature Comm*, 2017) show that this might not be the case when thermal processes are involved. In these experiments where normal stress is varying with loading contrast, is the residual shear stress still nearly proportional to the local normal stress?

Response: The computation of G in our geometry consists of computing the Mushkelishvili integral (see our previous point-by-point response to the referees). This requires spatial resolution to accurately determine the residual stress *within the slipping patch* (granular part

and neighboring slipping solid-solid parts). Because we use a single strain gauge over the granular section, and that the subsequent strain gauges are located 15 mm away from the patch corners, the computation of the integral cannot be achieved. The best way to do this would be to perform DIC on a disordered pattern to obtain high resolution static strain measurements, but this is beyond what can be achieved in the current study.

We clarified the need for spatial resolution on the residual stress measurement in the revised manuscript:

l. 262: “In particular, knowledge of the residual stress distribution *within* the slowly slipping patch is required to evaluate the stress intensity factor at the tip of the initial rupture, while in our experiments, only one strain gauge is located above this region⁴⁴.”

44 Freund, L. B. *Dynamic Fracture Mechanics*. (Cambridge, 1990).

Referee: Comment 4: I do not fully agree with the red line shown in Figures 4a&b. The passage of a rupture is usually associated with a decrease of shear stress (see Berman et al., PRL, 2020, doi:10.1103/PhysRevLett.125.125503), while arrival times are sometimes located in the ascending phase of shear stress, i.e. when the stress concentration ahead of the crack tip starts loading the material point behind the strain gage.

Response: The red lines are thought to be a guide for the eye. Indeed, the rupture velocity is not expected to be constant during propagation, c_f depends on the rupture length, on the local shear loading and on the local normal stress (via the G_c value) (see Svetlizky et al, PRL 2017, DOI: 10.1103/PhysRevLett.118.125501).

We have modified the caption of Fig. 4 to make this clearer:

“Black dotted lines indicate the detection of the initiation of the strain variation for each strain gauge signal and red lines are a guide for the eyes to indicate the approximate rupture passage locations.”

Referee: Comment 5: The authors discuss the implications of their experiments on the understanding of tremors (line 319-321). In the case of anthropogenic fluid injection tremors are often observed both within and at the edges of what may be a slow slipping region (Sáez and Lecampion, Phil Trans Roy Soc A, 2023, doi:10.1098/rspa.2022.0810). Dynamic events associated with tremoring activity is often considered as contained ruptures (Veedu and Barbot, Nature, 2016, doi:10.1038/nature17190). Here, these events correspond to ruptures of the entire fault interface. How do the authors explain this difference? How is it possible to connect their lab measurements to field observations?

Response: We believe that the observed ruptures of the entire fault are due to the finite size of our system. For a longer fault, the shear stress distribution, compared to the normal stress distribution, would be more heterogeneous, leading to arrested ruptures (see Bayart et al, JGR 2018, doi.org/10.1002/2018JB015509). Our suggestion for gaining insight into the physics of tremors would be to perform acoustic measurements and relate the frequency content to the properties of the sliding patch. The emitted waves are mostly due to the acceleration of the rupture during the nucleation process, so the emergence of unconfined ruptures in experiments might not affect the results. It would also be interesting to find a way to introduce local slow slip along the interface other than a granular patch. This would allow the effect of

slow slip to be measured without introducing a normal stress gradient due to granular shear induced dilatancy.

REVIEWERS' COMMENTS

Reviewer #3 (Remarks to the Author):

I thank the authors for their careful answers to the questions I've raised. I am satisfied with the revised manuscript, and I am happy to recommend its publication in Nature Communications.